# Training deep neural density estimators to identify mechanistic models of neural dynamics

Pedro J Gonçalves[1,2]†*, Jan-Matthis Lueckmann[1,2]†*, Michael Deistler[1,3]†*, Marcel Nonnenmacher[1,2,4], Kaan Öcal[2,5], Giacomo Bassetto[1,2], Chaitanya Chintaluri[6,7], William F Podlaski[6], Sara A Haddad[8], Tim P Vogels[6,7], David S Greenberg[1,4], Jakob H Macke[1,2,3,9]*

[1]Computational Neuroengineering, Department of Electrical and Computer Engineering, Technical University of Munich, Munich, Germany; [2]Max Planck Research Group Neural Systems Analysis, Center of Advanced European Studies and Research (caesar), Bonn, Germany; [3]Machine Learning in Science, Excellence Cluster Machine Learning, Tübingen University, Tübingen, Germany; [4]Model-Driven Machine Learning, Institute of Coastal Research, Helmholtz Centre Geesthacht, Geesthacht, Germany; [5]Mathematical Institute, University of Bonn, Bonn, Germany; [6]Centre for Neural Circuits and Behaviour, University of Oxford, Oxford, United Kingdom; [7]Institute of Science and Technology Austria, Klosterneuburg, Austria; [8]Max Planck Institute for Brain Research, Frankfurt, Germany; [9]Max Planck Institute for Intelligent Systems, Tübingen, Germany

*For correspondence:
pedro.goncalves@caesar.de (PJG);
jan-matthis.lueckmann@tum.de (J-ML);
michael.deistler@tum.de (MD);
Jakob.Macke@gmail.com (JHM)

†These authors contributed equally to this work

Competing interests: The authors declare that no competing interests exist.

**Abstract** Mechanistic modeling in neuroscience aims to explain observed phenomena in terms of underlying causes. However, determining which model parameters agree with complex and stochastic neural data presents a significant challenge. We address this challenge with a machine learning tool which uses deep neural density estimators—trained using model simulations—to carry out Bayesian inference and retrieve the full space of parameters compatible with raw data or selected data features. Our method is scalable in parameters and data features and can rapidly analyze new data after initial training. We demonstrate the power and flexibility of our approach on receptive fields, ion channels, and Hodgkin–Huxley models. We also characterize the space of circuit configurations giving rise to rhythmic activity in the crustacean stomatogastric ganglion, and use these results to derive hypotheses for underlying compensation mechanisms. Our approach will help close the gap between data-driven and theory-driven models of neural dynamics.

## Introduction

New experimental technologies allow us to observe neurons, networks, brain regions, and entire systems at unprecedented scale and resolution, but using these data to understand how behavior arises from neural processes remains a challenge. To test our understanding of a phenomenon, we often take to rebuilding it in the form of a computational model that incorporates the mechanisms we believe to be at play, based on scientific knowledge, intuition, and hypotheses about the components of a system and the laws governing their relationships. The goal of such mechanistic models is to investigate whether a proposed mechanism can explain experimental data, uncover details that may have been missed, inspire new experiments, and eventually provide insights into the inner workings of an observed neural or behavioral phenomenon (*Herz et al., 2006*; *Gerstner et al., 2012*; *O'Leary et al., 2015*; *Baker et al., 2018*). Examples

**eLife digest** Computational neuroscientists use mathematical models built on observational data to investigate what's happening in the brain. Models can simulate brain activity from the behavior of a single neuron right through to the patterns of collective activity in whole neural networks. Collecting the experimental data is the first step, then the challenge becomes deciding which computer models best represent the data and can explain the underlying causes of how the brain behaves.

Researchers usually find the right model for their data through trial and error. This involves tweaking a model's parameters until the model can reproduce the data of interest. But this process is laborious and not systematic. Moreover, with the ever-increasing complexity of both data and computer models in neuroscience, the old-school approach of building models is starting to show its limitations.

Now, Gonçalves, Lueckmann, Deistler et al. have designed an algorithm that makes it easier for researchers to fit mathematical models to experimental data. First, the algorithm trains an artificial neural network to predict which models are compatible with simulated data. After initial training, the method can rapidly be applied to either raw experimental data or selected data features. The algorithm then returns the models that generate the best match.

This newly developed machine learning tool was able to automatically identify models which can replicate the observed data from a diverse set of neuroscience problems. Importantly, further experiments showed that this new approach can be scaled up to complex mechanisms, such as how a neural network in crabs maintains its rhythm of activity. This tool could be applied to a wide range of computational investigations in neuroscience and other fields of biology, which may help bridge the gap between 'data-driven' and 'theory-driven' approaches.

for such a symbiotic relationship between model and experiments range from the now classical work of *Hodgkin and Huxley, 1952*, to population models investigating rules of connectivity, plasticity and network dynamics (*van Vreeswijk and Sompolinsky, 1996*; *Prinz et al., 2004*; *Vogels et al., 2005*; *Potjans and Diesmann, 2014*; *Litwin-Kumar and Doiron, 2012*), network models of inter-area interactions (*Sporns, 2014*; *Bassett et al., 2018*), and models of decision making (*Gold and Shadlen, 2007*; *Wang, 2008*).

A crucial step in building a model is adjusting its free parameters to be consistent with experimental observations. This is essential both for investigating whether the model agrees with reality and for gaining insight into processes which cannot be measured experimentally. For some models in neuroscience, it is possible to identify the relevant parameter regimes from careful mathematical analysis of the model equations. But as the complexity of both neural data and neural models increases, it becomes very difficult to find well-fitting parameters by inspection, and *automated* identification of data-consistent parameters is required.

Furthermore, to understand how a model quantitatively explains data, it is necessary to find not only the *best*, but *all* parameter settings consistent with experimental observations. This is especially important when modeling neural data, where highly variable observations can lead to broad ranges of data-consistent parameters. Moreover, many models in biology are inherently robust to some perturbations of parameters, but highly sensitive to others (*Gutenkunst et al., 2007*; *O'Leary et al., 2015*), for example because of processes such as homeostatic regulation. For these systems, identifying the full range of data-consistent parameters can reveal how multiple distinct parameter settings give rise to the same model behavior (*Foster et al., 1993*; *Prinz et al., 2004*; *Achard and De Schutter, 2006*; *Alonso and Marder, 2019*). Yet, despite the clear benefits of mechanistic models in providing scientific insight, identifying their parameters given data remains a challenging open problem that demands new algorithmic strategies.

The gold standard for automated parameter identification is *statistical inference*, which uses the likelihood $p(\mathbf{x}|\theta)$ to quantify the match between parameters $\theta$ and data $\mathbf{x}$. Likelihoods can be efficiently computed for purely statistical models commonly used in neuroscience (*Truccolo et al., 2005*; *Schneidman et al., 2006*; *Pillow et al., 2008*; *Yu et al., 2009*; *Macke et al., 2011*; *Cunningham and Yu, 2014*; *Pandarinath et al., 2018*), but are computationally intractable for most

mechanistic models. Mechanistic models are designed to reflect knowledge about biological mechanisms, and not necessarily to be amenable to efficient inference: many mechanistic models are defined implicitly through stochastic computer simulations (e.g. a simulation of a network of spiking neurons), and likelihood calculation would require the ability to integrate over all potential paths through the simulator code. Similarly, a common goal of mechanistic modeling is to capture selected summary features of the data (e.g. a certain firing rate, bursting behavior, etc...), *not* the full dataset in all its details. The same feature (such as a particular average firing rate) can be produced by infinitely many realizations of the simulated process (such as a time-series of membrane potential). This makes it impractical to compute likelihoods, as one would have to average over all possible realizations which produce the same output.

Since the toolkit of (likelihood-based) statistical inference is inaccessible for mechanistic models, parameters are typically tuned ad-hoc (often through laborious, and subjective, trial-and-error), or by computationally expensive parameter search: a large set of models is generated, and grid search (*Prinz et al., 2003*; *Tomm et al., 2011*; *Stringer et al., 2016*) or a genetic algorithm (*Druckmann et al., 2007*; *Hay et al., 2011*; *Rossant et al., 2011*; *Van Geit et al., 2016*) is used to filter out simulations which do not match the data. However, these approaches require the user to define a heuristic rejection criterion on which simulations to keep (which can be challenging when observations have many dimensions or multiple units of measurement), and typically end up discarding most simulations. Furthermore, they lack the advantages of statistical inference, which provides principled approaches for handling variability, quantifying uncertainty, incorporating prior knowledge and integrating multiple data sources. Approximate Bayesian Computation (ABC) (*Beaumont et al., 2002*; *Marjoram et al., 2003*; *Sisson et al., 2007*) is a parameter-search technique which aims to perform statistical inference, but still requires definition of a rejection criterion and struggles in high-dimensional problems. Thus, computational neuroscientists face a dilemma: either create carefully designed, highly interpretable mechanistic models (but rely on ad-hoc parameter tuning), or resort to purely statistical models offering sophisticated parameter inference but limited mechanistic insight.

Here, we propose a new approach using machine learning to combine the advantages of mechanistic and statistical modeling. We present SNPE (Sequential Neural Posterior Estimation), a tool that makes it possible to perform Bayesian inference on mechanistic models in neuroscience without requiring access to likelihoods. SNPE identifies all mechanistic model parameters consistent with observed experimental data (or summary features). It builds on recent advances in simulation-based Bayesian inference (*Papamakarios and Murray, 2016*; *Lueckmann et al., 2017*; *Greenberg et al., 2019*; *Cranmer et al., 2020*): given observed experimental data (or summary features) $\mathbf{x}_o$, and a mechanistic model with parameters $\theta$, it expresses both prior knowledge and the range of data-compatible parameters through probability distributions. SNPE returns a posterior distribution $p(\theta|\mathbf{x}_o)$ which is high for parameters $\theta$ consistent with both the data $\mathbf{x}_o$ and prior knowledge, but approaches zero for $\theta$ inconsistent with either (*Figure 1*).

Similar to parameter search methods, SNPE uses simulations instead of likelihood calculations, but instead of filtering out simulations, it uses *all* simulations to train a multilayer artificial network to identify admissible parameters (*Figure 1*). By incorporating modern deep neural networks for conditional density estimation (*Rezende and Mohamed, 2015*; *Papamakarios et al., 2017*), it can capture the full *distribution* of parameters consistent with the data, even when this distribution has multiple peaks or lies on curved manifolds. Critically, SNPE decouples the design of the model and design of the inference approach, giving the investigator maximal flexibility to design and modify mechanistic models. Our method makes minimal assumptions about the model or its implementation, and can for example also be applied to non-differentiable models, such as networks of spiking neurons. Its only requirement is that one can run model simulations for different parameters, and collect the resulting synthetic data or summary features of interest.

While the theoretical foundations of SNPE were originally developed and tested using simple inference problems on small models (*Papamakarios and Murray, 2016*; *Lueckmann et al., 2017*; *Greenberg et al., 2019*), here we show that SNPE can scale to complex mechanistic models in neuroscience, provide an accessible and powerful implementation, and develop validation and visualization techniques for exploring the derived posteriors. We illustrate SNPE using mechanistic models expressing key neuroscientific concepts: beginning with a simple neural encoding problem with a known solution, we progress to more complex data types, large datasets and many-parameter

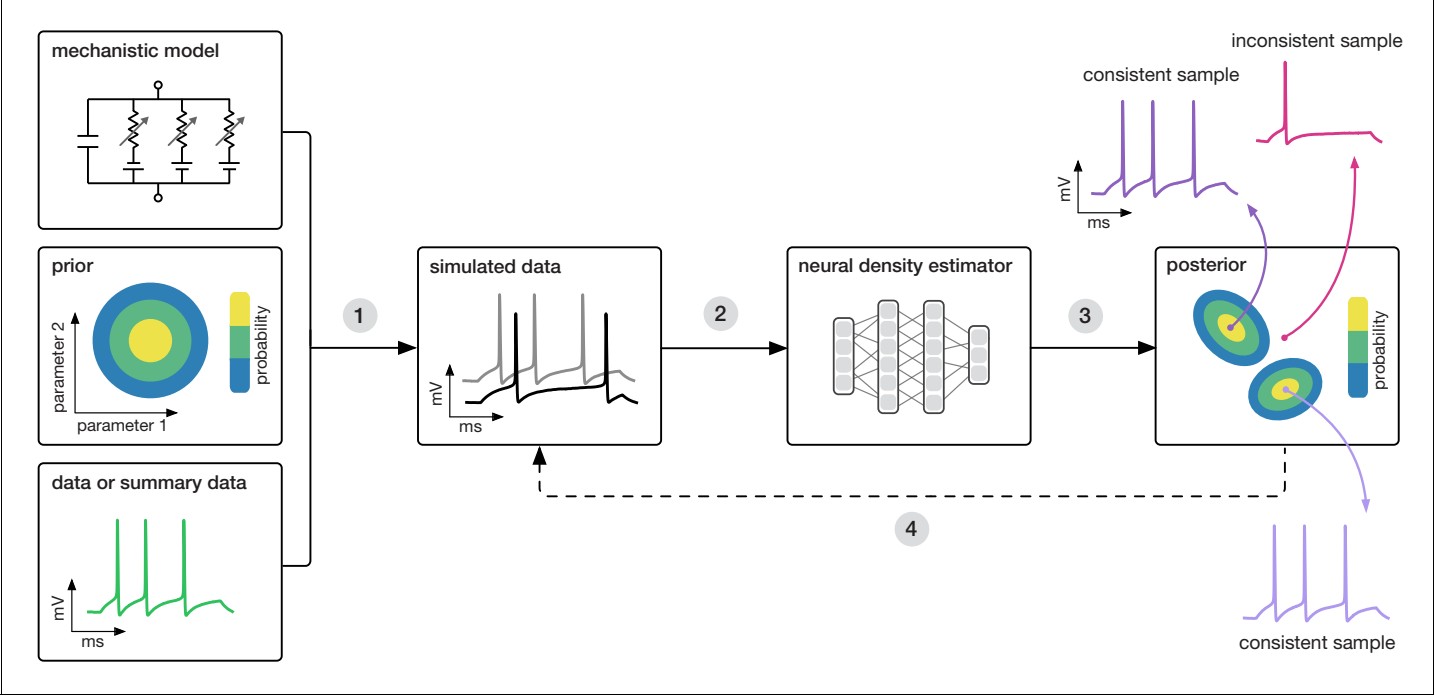

**Figure 1.** Goal: algorithmically identify mechanistic models which are consistent with data. Our algorithm (SNPE) takes three inputs: a candidate mechanistic model, prior knowledge or constraints on model parameters, and data (or summary statistics). SNPE proceeds by (1) sampling parameters from the prior and simulating synthetic datasets from these parameters, and (2) using a deep density estimation neural network to learn the (probabilistic) association between data (or data features) and underlying parameters, that is to learn statistical inference from simulated data. (3) This density estimation network is then applied to empirical data to derive the full space of parameters consistent with the data and the prior, that is, the posterior distribution. High posterior probability is assigned to parameters which are consistent with both the data and the prior, low probability to inconsistent parameters. (4) If needed, an initial estimate of the posterior can be used to adaptively guide further simulations to produce data-consistent results.

models inaccessible to previous methods. We estimate visual receptive fields using many data features, demonstrate rapid inference of ion channel properties from high-throughput voltage-clamp protocols, and show how Hodgkin–Huxley models are more tightly constrained by increasing numbers of data features. Finally, we showcase the power of SNPE by using it to identify the parameters of a network model which can explain an experimentally observed pyloric rhythm in the stomatogastric ganglion (*Prinz et al., 2004*)–in contrast to previous approaches, SNPE allows us to search over the full space of both single-neuron and synaptic parameters, allowing us to study the geometry of the parameter space, as well as to provide new hypotheses for which compensation mechanisms might be at play.

## Results

### Training neural networks to perform Bayesian inference without likelihood evaluations

SNPE performs Bayesian inference on mechanistic models using only model-simulations, without requiring likelihood evaluations. It requires three inputs: a model (i.e. computer code to simulate data from parameters), prior knowledge or constraints on parameters, and data (outputs from the model or the real system it describes, *Figure 1*). SNPE runs simulations for a range of parameter values, and trains an artificial neural network to map any simulation result onto a range of possible parameters. Importantly, a network trained to maximize log-probability (of parameters given simulation results) will learn to approximate the posterior distribution as given by Bayes rule (*Papamakarios and Murray, 2016*) (see Materials and methods for details, *Figure 1*). After training on *simulated* data with known model parameters, SNPE can perform Bayesian inference of unknown

parameters for *empirical* data. This approach to Bayesian inference never requires evaluating likelihoods. SNPE's efficiency can be further improved by using the running estimate of the posterior distribution to guide further simulations toward data-compatible regions of the parameter space (*Papamakarios and Murray, 2016*; *Lueckmann et al., 2017*; *Greenberg et al., 2019*). Below, we apply SNPE to a range of stochastic models in neuroscience.

## Estimating stimulus-selectivity in linear-nonlinear encoding models

We first illustrate SNPE on linear-nonlinear (LN) encoding models, a special case of generalized linear models (GLMs). These are simple, commonly used phenomenological models for which likelihood-based parameter estimation is feasible (*Brown et al., 1998*; *Paninski, 2004*; *Pillow, 2007*; *Gerwinn et al., 2010*; *Polson et al., 2013*; *Pillow and Scott, 2012*), and which can be used to validate the accuracy of our approach, before applying SNPE to more complex models for which the likelihood is unavailable. We will show that SNPE returns the correct posterior distribution over parameters, that it can cope with high-dimensional observation data, that it can recover multiple solutions to parameter inference problems, and that it is substantially more simulation efficient than conventional rejection-based ABC methods.

An LN model describes how a neuron's firing rate is modulated by a sensory stimulus through a linear filter $\theta$, often referred to as the *receptive field* (*Pillow et al., 2005*; *Chichilnisky, 2001*). We first considered a model of a retinal ganglion cell (RGC) driven by full-field flicker (*Figure 2a*). A statistic that is often used to characterize such a neuron is the *spike-triggered average* (STA) (*Figure 2a*, right). We therefore used the STA, as well as the firing rate of the neuron, as input $\mathbf{x}_o$ to SNPE. (Note that, in the limit of infinite data, and for white noise stimuli, the STA will converge to the receptive field [*Paninski, 2004*]–for finite, and non-white data, the two will in general be different.) Starting with random receptive fields $\theta$, we generated synthetic spike trains and calculated STAs from them (*Figure 2b*). We then trained a neural conditional density estimator to recover the receptive fields from the STAs and firing rates (*Figure 2c*). This allowed us to estimate the posterior distribution over receptive fields, that is to estimate which receptive fields are consistent with the data (and prior) (*Figure 2c*). For LN models, likelihood-based inference is possible, allowing us to validate the SNPE posterior by comparing it to a reference posterior obtained via Markov Chain Monte Carlo (MCMC) sampling (*Polson et al., 2013*; *Pillow and Scott, 2012*). We found that SNPE accurately estimates the posterior distribution (*Appendix 1—figure 1* and *Appendix 1—figure 2*), and substantially outperforms Sequential Monte Carlo (SMC) ABC methods (*Sisson et al., 2007*; *Beaumont et al., 2009*; *Figure 2d*). If SNPE works correctly, its posterior mean filter will match that of the reference posterior – however, it is not to be expected that either of them precisely matches the ground-truth filter (*Figure 2c* and *Appendix 1—figure 1*): In the presence of finite sampling and stochasticity, multiple different filters could have plausibly given rise to the observed data. A properly inferred posterior will reflect this uncertainty, and include the true filters as one of many plausible explanations of the data (but not necessarily as the 'mean' of all plausible explanations) (*Appendix 1—figure 2*). Increasing the number of Bernoulli samples in the observed data leads to progressively tighter posteriors, with posterior samples closer to the true filter (*Appendix 1—figure 3*). Furthermore, SNPE closely agrees with the MCMC reference solution in all these cases, further emphasizing the correctness of the posteriors inferred with SNPE.

As a more challenging problem, we inferred the receptive field of a neuron in primary visual cortex (V1) (*Niell and Stryker, 2008*; *Dyballa et al., 2018*). Using a model composed of a bias (related to the spontaneous firing rate) and a Gabor function with eight parameters (*Jones and Palmer, 1987*) describing the receptive field's location, shape and strength, we simulated responses to 5 min random noise movies of $41 \times 41$ pixels, such that the STA is high-dimensional, with a total of 1681 dimensions (*Figure 2e*). This problem admits multiple solutions (as e.g. rotating the receptive field by 180°). As a result, the posterior distribution has multiple peaks ('modes'). Starting from a simulation result $\mathbf{x}_o$ with known parameters, we used SNPE to estimate the posterior distribution $p(\theta|\mathbf{x}_o)$. To deal with the high-dimensional data $\mathbf{x}_o$ in this problem, we used a convolutional neural network (CNN), as this architecture excels at learning relevant features from image data (*Krizhevsky et al., 2012*; *Simonyan and Zisserman, 2015*). To deal with the multiple peaks in the posterior, we fed the CNN's output into a mixture density network (MDN) (*Bishop, 1994*), which can learn to assign probability distributions with multiple peaks as a function of its inputs (details in Materials and methods). Using this strategy, SNPE was able to infer a posterior distribution that tightly enclosed the ground

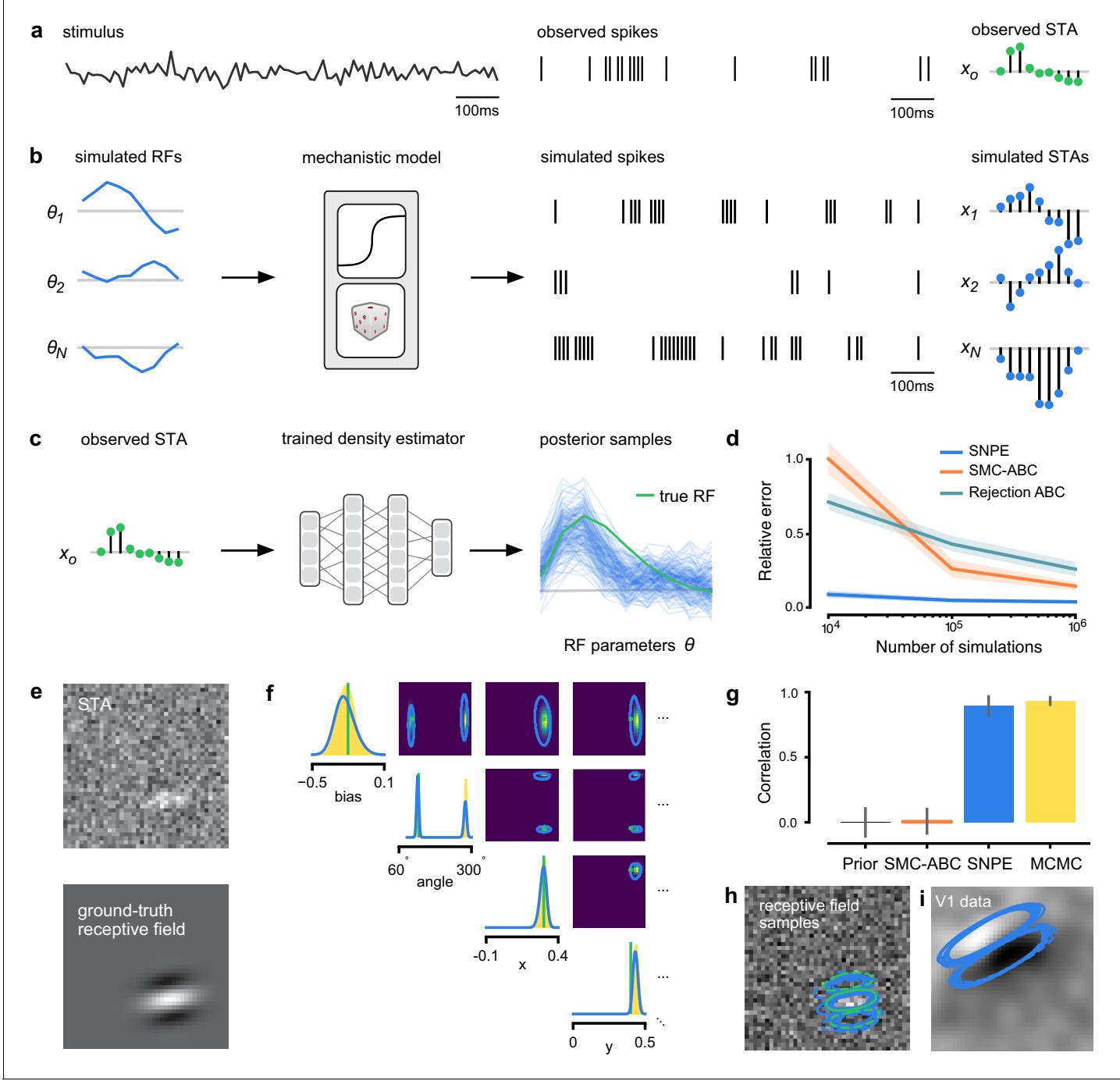

**Figure 2.** Estimating receptive fields in linear-nonlinear models of single neurons with statistical inference. (**a**) Schematic of a time-varying stimulus, associated observed spike train and resulting spike-triggered average (STA) (**b**) Sequential Neural Posterior Estimation (SNPE) proceeds by first randomly generating simulated receptive fields θ, and using the mechanistic model (here an LN model) to generate simulated spike trains and simulated STAs. (**c**) These simulated STAs and receptive fields are then used to train a deep neural density estimator to identify the distribution of receptive fields consistent with a given observed STA $\mathbf{x}_o$. (**d**) Relative error in posterior estimation of SNPE and alternative methods (mean and 95% CI; 0 corresponds to perfect estimation, one to prior-level, details in Materials and methods). (**e**) Example of spatial receptive field. We simulated responses and an STA of an LN-model with oriented receptive field. (**f**) We used SNPE to recover the distribution of receptive-field parameters. Univariate and pairwise marginals for four parameters of the spatial filter (MCMC, yellow histograms; SNPE, blue lines; ground truth, green; full posterior in *Appendix 1—figure 4*). Non-identifiabilities of the Gabor parameterization lead to multimodal posteriors. (**g**) Average correlation (± SD) between ground-truth receptive field and receptive field samples from posteriors inferred with SMC-ABC, SNPE, and MCMC (which provides an upper

*Figure 2 continued on next page*

*Figure 2 continued*

bound given the inherent stochasticity of the data). (h) Posterior samples from SNPE posterior (SNPE, blue) compared to ground-truth receptive field (green; see panel (e)), overlaid on STA. (i) Posterior samples for V1 data; full posterior in *Appendix 1—figure 6*.

truth simulation parameters which generated the original simulated data $\mathbf{x}_o$, and matched a reference MCMC posterior (*Figure 2f*, posterior over all parameters in *Appendix 1—figure 4*). For this challenging estimation problem with high-dimensional summary features, an SMC-ABC algorithm with the same simulation-budget failed to identify the correct receptive fields (*Figure 2g*) and posterior distributions (*Appendix 1—figure 5*). We also applied this approach to electrophysiological data from a V1 cell (*Dyballa et al., 2018*), identifying a sine-shaped Gabor receptive field consistent with the original spike-triggered average (*Figure 2i*; posterior distribution in *Appendix 1—figure 6*).

## Functional diversity of ion channels: efficient high-throughput inference

We next show how SNPE can be efficiently applied to estimation problems in which we want to identify a large number of models for different observations in a database. We considered a flexible model of ion channels (*Destexhe and Huguenard, 2000*), which we here refer to as the *Omnimodel*. This model uses eight parameters to describe how the dynamics of currents through non-inactivating potassium channels depend on membrane voltage (*Figure 3a*). For various choices of its parameters $\theta$, it can capture 350 specific models in publications describing this channel type, cataloged in the IonChannelGenealogy (ICG) database (*Podlaski et al., 2017*). We aimed to identify these ion channel parameters $\theta$ for each ICG model, based on 11 features of the model's response to a sequence of five noisy voltage clamp protocols, resulting in a total of 55 different characteristic features per model (*Figure 3b*, see Materials and methods for details).

Because this model's output is a typical format for functional characterization of ion channels both in simulations (*Podlaski et al., 2017*) and in high-throughput electrophysiological experiments (*Dunlop et al., 2008*; *Suk et al., 2019*; *Ranjan et al., 2019*), the ability to rapidly infer different parameters for many separate experiments is advantageous. Existing fitting approaches based on numerical optimization (*Destexhe and Huguenard, 2000*; *Ranjan et al., 2019*) must repeat all computations anew for a new experiment or data point (*Figure 3c*). However, for SNPE the only heavy computational tasks are carrying out simulations to generate training data, and training the neural network. We therefore reasoned that by training a network once using a large number of simulations, we could subsequently carry out rapid 'amortized' parameter inference on new data using a single pass through the network (*Figure 3d*; *Speiser et al., 2017*; *Webb et al., 2018*). To test this idea, we used SNPE to train a neural network to infer the posterior from any data $\mathbf{x}$. To generate training data, we carried out 1 million Omnimodel simulations, with parameters randomly chosen across ranges large enough to capture the models in the ICG database (*Podlaski et al., 2017*). SNPE was run using a single round, that is, it learned to perform inference for all data from the prior (rather than a specific observed datum). Generating these simulations took around 1000 CPU-hours and training the network 150 CPU-hours, but afterwards a full posterior distribution could be inferred for new data in less than 10 ms.

As a first test, SNPE was run on simulation data, generated by a previously published model of a non-inactivating potassium channel (*McTavish et al., 2012*; *Figure 3b*). Simulations of the Omnimodel using parameter sets sampled from the obtained posterior distribution (*Figure 3e*) closely resembled the input data on which the SNPE-based inference had been carried out, while simulations using 'outlier' parameter sets with low probability under the posterior generated current responses that were markedly different from the data $\mathbf{x}_o$ (*Figure 3f*). Taking advantage of SNPE's capability for rapid amortized inference, we further evaluated its performance on all 350 non-inactivating potassium channel models in ICG. In each case, we carried out a simulation to generate initial data from the original ICG model, used SNPE to calculate the posterior given the Omnimodel, and then generated a new simulation $\mathbf{x}$ using parameters sampled from the posterior (*Figure 3f*). This resulted in high correlation between the original ICG model response and the Omnimodel response, in every case (>0.98 for more than 90% of models, see *Appendix 1—figure 7*). However, this approach was not able to capture all traces perfectly, as for example it failed to capture the shape of the onset of the bottom right model in *Figure 3g*. Additional analysis of this example revealed

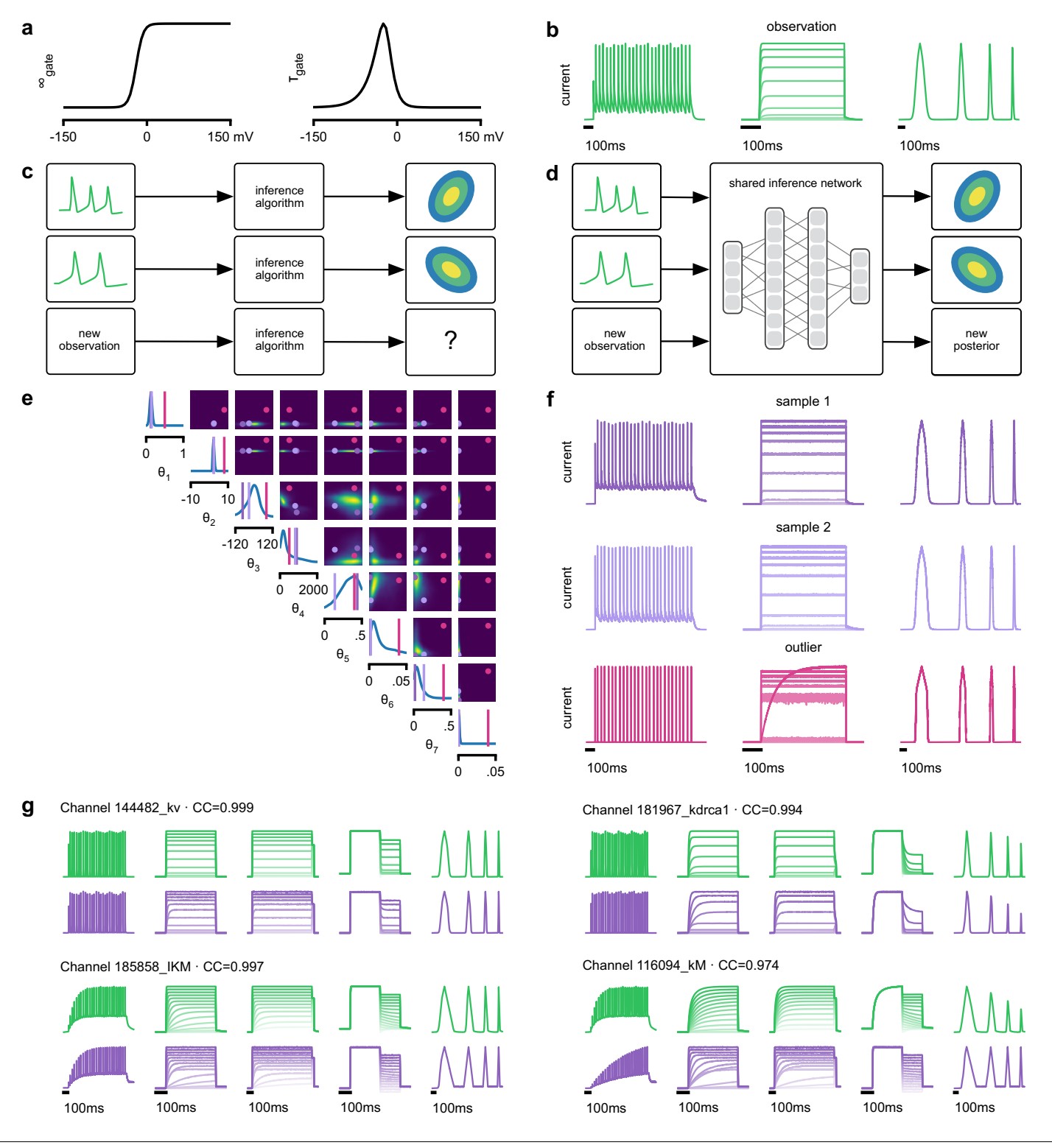

**Figure 3.** Inference on a database of ion-channel models. (**a**) We perform inference over the parameters of non-inactivating potassium channel models. Channel kinetics are described by steady-state activation curves, $\infty_{\text{gate}}$, and time-constant curves, $\tau_{\text{gate}}$. (**b**) Observation generated from a channel model from ICG database: normalized current responses to three (out of five) voltage-clamp protocols (action potentials, activation, and ramping). Details in *Podlaski et al., 2017*. (**c**) Classical approach to parameter identification: inference is optimized on each datum separately, requiring new computations for each new datum. (**d**) Amortized inference: an inference network is learned which can be applied to multiple data, enabling rapid inference on new data. (**e**) Posterior distribution over eight model parameters, $\theta_1$ to $\theta_8$. Ground truth parameters in green, high-probability parameters
*Figure 3 continued on next page*

*Figure 3 continued*

in purple, low-probability parameters in magenta. (f) Traces obtained by sampling from the posterior in (e). Purple: traces sampled from posterior, that is, with high posterior probability. Magenta: trace from parameters with low probability. (g) Observations (green) and traces generated by posterior samples (purple) for four models from the database.

that this example is not a failure of SNPE, but rather a limitation of the Omnimodel: in particular, directly fitting the steady-state activation and time-constant curves on this specific example yielded no further quantitative or qualitative improvement, suggesting that the limitation is in the model, not the fit. Thus, SNPE can be used to reveal limitations of candidate models and aid the development of more verisimilar mechanistic models.

Calculating the posterior for all 350 ICG models took only a few seconds, and was fully automated, that is, did not require user interactions. These results show how SNPE allows fast and accurate identification of biophysical model parameters on new data, and how SNPE can be deployed for applications requiring rapid automated inference, such as high-throughput screening-assays, closed-loop paradigms (e.g. for adaptive experimental manipulations or stimulus-selection [*Kleinegesse and Gutmann, 2019*]), or interactive software tools.

## Hodgkin–Huxley model: stronger constraints from additional data features

The Hodgkin–Huxley (HH) model (*Hodgkin and Huxley, 1952*) of action potential generation through ion channel dynamics is a highly influential mechanistic model in neuroscience. A number of algorithms have been proposed for fitting HH models to electrophysiological data (*Prinz et al., 2003*; *Huys et al., 2006*; *Pospischil et al., 2008*; *Rossant et al., 2011*; *Meliza et al., 2014*; *Van Geit et al., 2016*; *Ben-Shalom et al., 2019*), but (with the exception of *Daly et al., 2015*) these approaches do not attempt to estimate the full posterior. Given the central importance of the HH model in neuroscience, we sought to test how SNPE would cope with this challenging non-linear model.

As previous approaches for HH models concentrated on reproducing specified features (e.g. the number of spikes, [*Pospischil et al., 2008*]), we also sought to determine how various features provide different constraints. We considered the problem of inferring eight biophysical parameters in a HH single-compartment model, describing voltage-dependent sodium and potassium conductances and other intrinsic membrane properties, including neural noise, making the model stochastic by nature (*Figure 4a*, left). We simulated the neuron's voltage response to the injection of a square wave of depolarizing current, and defined the model output $\mathbf{x}$ used for inference as the number of evoked action potentials along with six additional features of the voltage response (*Figure 4a*, right, details in Materials and methods). We first applied SNPE to observed data $\mathbf{x}_o$ created by simulation from the model, calculating the posterior distribution using all seven features in the observed data (*Figure 4b*). The posterior contained the ground truth parameters in a high probability-region, as in previous applications, indicating the consistency of parameter identification. The variance of the posterior was narrower for some parameters than for others, indicating that the seven data features constrain some parameters strongly (such as the potassium conductance), but others only weakly (such as the adaptation time constant). Additional simulations with parameters sampled from the posterior closely resembled the observed data $\mathbf{x}_o$, in terms of both the raw membrane voltage over time and the seven data features (*Figure 4c*, purple and green). Parameters with low posterior probability (outliers) generated simulations that markedly differed from $\mathbf{x}_o$ (*Figure 4c*, magenta).

Genetic algorithms are commonly used to fit parameters of deterministic biophysical models (*Druckmann et al., 2007*; *Hay et al., 2011*; *Van Geit et al., 2016*; *Gouwens et al., 2018*). While genetic algorithms can also return multiple data-compatible parameters, they do not perform inference (i.e. find the posterior distribution), and their outputs depend strongly on user-defined goodness-of-fit criteria. When comparing a state-of-the-art genetic algorithm (Indicator Based Evolutionary Algorithm, IBEA, [*Bleuler et al., 2003*; *Zitzler and Künzli, 2004*; *Van Geit et al., 2016*]) to SNPE, we found that the parameter-settings favored by IBEA produced simulations whose summary features were as similar to the observed data as those obtained by SNPE high-probability samples (*Appendix 1—figure 10*). However, high-scoring IBEA parameters were concentrated in

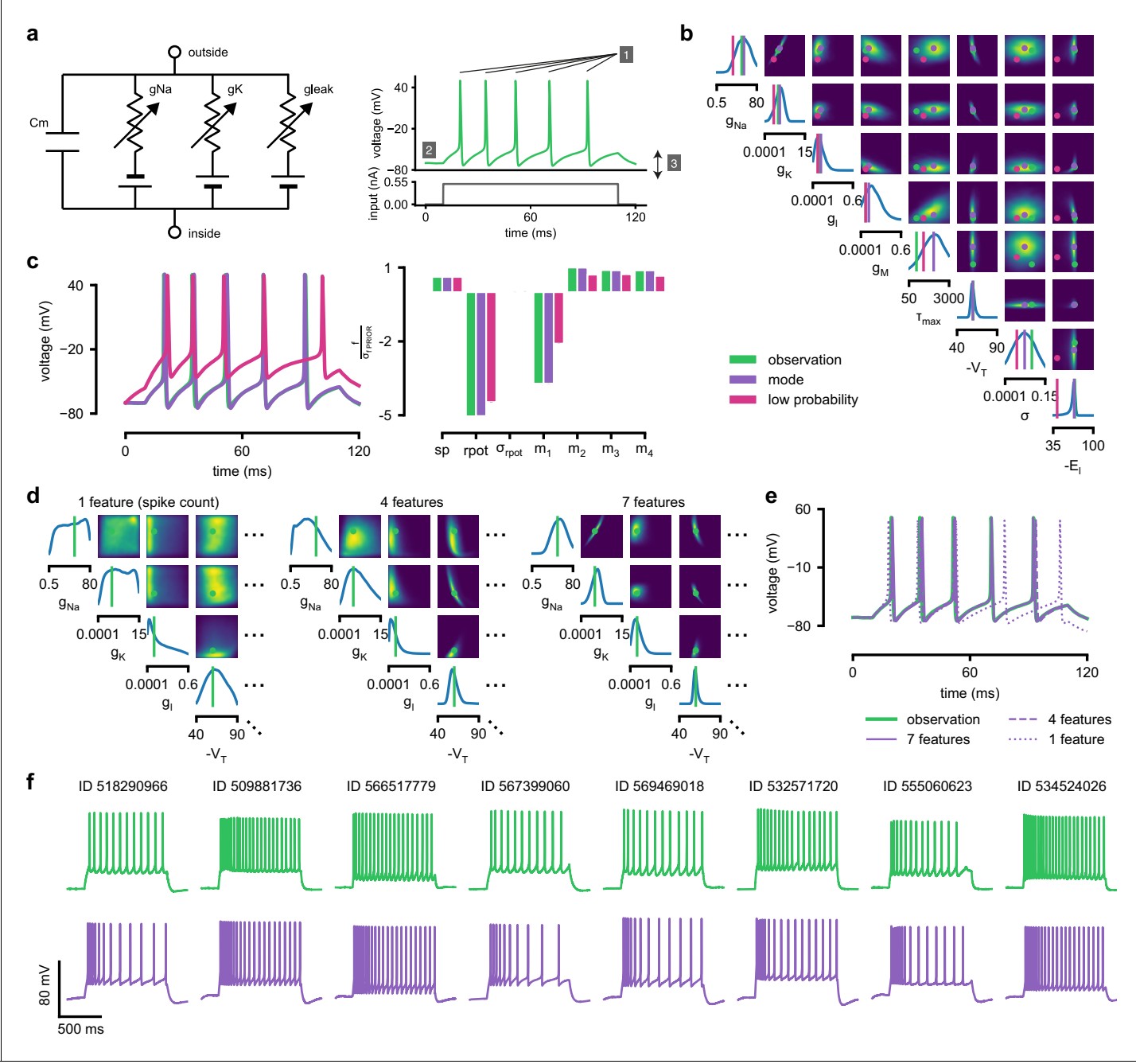

**Figure 4.** Inference for single compartment Hodgkin–Huxley model. (**a**) Circuit diagram describing the Hodgkin–Huxley model (left), and simulated voltage-trace given a current input (right). Three out of 7 voltage features are depicted: (1) number of spikes, (2) mean resting potential, and (3) standard deviation of the pre-stimulus resting potential. (**b**) Inferred posterior for 8 parameters given seven voltage features. Ground truth parameters in green, high-probability parameters in purple, low-probability parameters in magenta. (**c**) Traces (left) and associated features *f* (right) for the desired output (observation), the mode of the inferred posterior, and a sample with low posterior probability. The voltage features are: number of spikes *sp*, mean resting potential *rpot*, standard deviation of the resting potential $\sigma_{rpot}$, and the first four voltage moments, mean $m_1$, standard deviation $m_2$, skewness $m_3$ and kurtosis $m_4$. Each value plotted is the mean feature ± standard deviation across 100 simulations with the same parameter set. Each feature is normalized by $\sigma_{fPRIOR}$, the standard deviation of the respective feature of simulations sampled from the prior. (**d**) Partial view of the inferred posteriors (4 out of 8 parameters) given 1, 4 and 7 features (full posteriors over eight parameters in *Appendix 1—figure 8*). (**e**) Traces for posterior modes given 1, 4 and 7 features. Increasing the number of features leads to posterior traces that are closer to the observed data. (**f**) Observations from Allen Cell Types Database (green) and corresponding mode samples (purple). Posteriors in *Appendix 1—figure 9*.

small regions of the posterior, that is, IBEA did not identify the full space of data-compatible models.

To investigate how individual data features constrain parameters, we compared SNPE-estimated posteriors based (1) solely on the spike count, (2) on the spike count and three voltage-features, or (3) on all 7 features of $\mathbf{x}_o$. As more features were taken into account, the posterior became narrower and centered more closely on the ground truth parameters (*Figure 4d*, *Appendix 1—figure 8*). Posterior simulations matched the observed data only in those features that had been used for inference (e.g. applying SNPE to spike counts alone identified parameters that generated the correct number of spikes, but for which spike timing and subthreshold voltage time course were off, *Figure 4e*). For some parameters, such as the potassium conductance, providing more data features brought the peak of the posterior (the *posterior mode*) closer to the ground truth and also decreased uncertainty. For other parameters, such as $V_T$, a parameter adjusting the spike threshold (*Pospischil et al., 2008*), the peak of the posterior was already close to the correct value with spike counts alone, but adding additional features reduced uncertainty. While SNPE can be used to study the effect of additional data features in reducing parameter uncertainty, this would not be the case for methods that only return a single best-guess estimate of parameters. These results show that SNPE can reveal how information from multiple data features imposes collective constraints on channel and membrane properties in the HH model.

We also inferred HH parameters for eight in vitro recordings from the Allen Cell Types database using the same current-clamp stimulation protocol as in our model (*Allen Institute for Brain Science, 2016*; *Teeter et al., 2018*; *Figure 4f*, *Appendix 1—figure 9*). In each case, simulations based on the SNPE-inferred posterior closely resembled the original data (*Figure 4f*). We note that while inferred parameters differed across recordings, some parameters (the spike threshold, the density of sodium channels, the membrane reversal potential and the density of potassium channels) were consistently more strongly constrained than others (the intrinsic neural noise, the adaptation time constant, the density of slow voltage-dependent channels and the leak conductance) (*Appendix 1—figure 9*). Overall, these results suggest that the electrophysiological responses measured by this current-clamp protocol can be approximated by a single-compartment HH model, and that SNPE can identify the admissible parameters.

## Crustacean stomatogastric ganglion: sensitivity to perturbations

We next aimed to demonstrate how the full posterior distribution obtained with SNPE can lead to novel scientific insights. To do so, we used the pyloric network of the stomatogastric ganglion (STG) of the crab *Cancer borealis*, a well-characterized neural circuit producing rhythmic activity. In this circuit, similar network activity can arise from vastly different sets of membrane and synaptic conductances (*Prinz et al., 2004*). We first investigated whether data-consistent sets of membrane and synaptic conductances are connected in parameter space, as has been demonstrated for single neurons (*Taylor et al., 2006*), and, second, which compensation mechanisms between parameters of this circuit allow the neural system to maintain its activity despite parameter variations. While this model has been studied extensively, answering these questions requires characterizing higher dimensional parameter spaces than those accessed previously. We demonstrate how SNPE can be used to identify the posterior distribution over both membrane and synaptic conductances of the STG (31 parameters total) and how the full posterior distribution can be used to study the above questions at the circuit level.

For some biological systems, multiple parameter sets give rise to the same system behavior (*Prinz et al., 2004*; *Marder and Goaillard, 2006*; *Gutierrez et al., 2013*; *Fisher et al., 2013*; *Marder et al., 2015*; *Alonso and Marder, 2019*). In particular, neural systems can be robust to specific perturbations of parameters (*O'Leary et al., 2014*; *Marder et al., 2015*; *O'Leary and Marder, 2016*), yet highly sensitive to others, properties referred to as *sloppiness* and *stiffness* (*Goldman et al., 2001*; *Gutenkunst et al., 2007*; *Machta et al., 2013*; *O'Leary et al., 2015*). We studied how perturbations affect model output using a model (*Prinz et al., 2004*) and data (*Haddad and Marder, 2018*) of the pyloric rhythm in the crustacean stomatogastric ganglion (STG). This model describes a triphasic motor pattern generated by a well-characterized circuit (*Figure 5a*). The circuit consists of two electrically coupled pacemaker neurons (anterior burster and pyloric dilator, AB/PD), modeled as a single neuron, as well as two types of follower neurons (lateral pyloric (LP) and pyloric (PY)), all connected through inhibitory synapses (details in Materials and methods). Eight

membrane conductances are included for each modeled neuron, along with seven synaptic conductances, for a total of 31 parameters. This model has been used to demonstrate that virtually indistinguishable activity can arise from vastly different membrane and synaptic conductances in the STG (*Prinz et al., 2004*; *Alonso and Marder, 2019*). Here, we build on these studies and extend the model to include intrinsic neural noise on each neuron (see Materials and methods).

We applied SNPE to an extracellular recording from the STG of the crab *Cancer borealis* (*Haddad and Marder, 2018*) which exhibited pyloric activity (*Figure 5b*), and inferred the posterior distribution over all 31 parameters based on 18 salient features of the voltage traces, including cycle period, phase delays, phase gaps, and burst durations (features in *Figure 5B*, posterior in *Figure 5c*, posterior over all parameters in *Appendix 1—figure 11*, details in Materials and methods). Consistent with previous reports, the posterior distribution has high probability over extended value ranges for many membrane and synaptic conductances. To verify that parameter settings across these extended ranges are indeed capable of generating the experimentally observed network activity, we sampled two sets of membrane and synaptic conductances from the posterior distribution. These two samples have widely disparate parameters from each other (*Figure 5c*, purple dots, details in Materials and methods), but both exhibit activity highly similar to the experimental observation (*Figure 5d*, top left and top right).

We then investigated the geometry of the parameter space producing these rhythms (*Achard and De Schutter, 2006*; *Alonso and Marder, 2019*). First, we wanted to identify directions of sloppiness, and we were interested in whether parameter settings producing pyloric rhythms form a single connected region, as has been shown for single neurons (*Taylor et al., 2006*), or whether they lie on separate 'islands'. Starting from the two parameter settings showing similar activity above, we examined whether they were connected by searching for a path

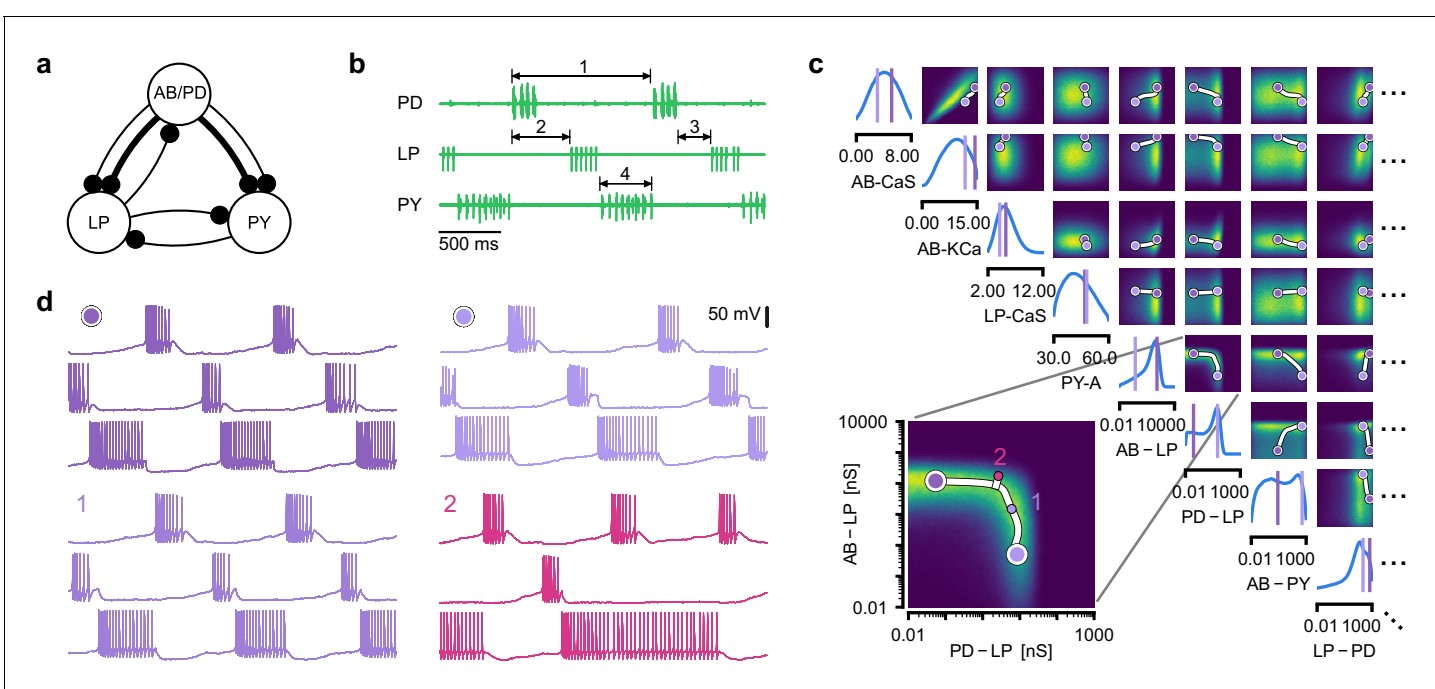

**Figure 5.** Identifying network models underlying an experimentally observed pyloric rhythm in the crustacean stomatogastric ganglion. (a) Simplified circuit diagram of the pyloric network from the stomatogastric ganglion. Thin connections are fast glutamatergic, thick connections are slow cholinergic. (b) Extracellular recordings from nerves of pyloric motor neurons of the crab *Cancer borealis* (*Haddad and Marder, 2018*). Numbers indicate some of the used summary features, namely cycle period (1), phase delays (2), phase gaps (3), and burst durations (4) (see Materials and methods for details). (c) Posterior over 24 membrane and seven synaptic conductances given the experimental observation shown in panel b (eight parameters shown, full posterior in *Appendix 1—figure 11*). Two high-probability parameter sets in purple. Inset: magnified marginal posterior for the synaptic strengths AB to LP neuron vs. PD to LP neuron. (d) Identifying directions of sloppiness and stiffness. Two samples from the posterior both show similar network activity as the experimental observation (top left and top right), but have very different parameters (purple dots in panel c). Along the high-probability path between these samples, network activity is preserved (trace 1). When perturbing the parameters orthogonally off the path, network activity changes abruptly and becomes non-pyloric (trace 2).

through parameter space along which pyloric activity was maintained. To do this, we algorithmically identified a path lying only in regions of high posterior probability (*Figure 5c*, white, details in Materials and methods). Along the path, network output was tightly preserved, despite a substantial variation of the parameters (voltage trace 1 in *Figure 5d*, *Appendix 1—figure 12*). Second, we inspected directions of stiffness by perturbing parameters off the path. We applied perturbations that yield maximal drops in posterior probability (see Materials and methods for details), and found that the network quickly produced non-pyloric activity (voltage trace 2, *Figure 5d*; *Goldman et al., 2001*). Note that, while parameter set 2 seems to lie in regions of high probability when inspecting pairwise marginals, it in fact has low probability under the full posterior distribution (*Appendix 1—figure 13*). In identifying these paths and perturbations, we exploited the fact that SNPE provides a differentiable estimate of the posterior, as opposed to parameter search methods which provide only discrete samples.

Overall, these results show that the pyloric network can be robust to specific perturbations in parameter space, but sensitive to others, and that one can interpolate between disparate solutions while preserving network activity. This analysis demonstrates the flexibility of SNPE in capturing complex posterior distributions, and shows how the differentiable posterior can be used to study directions of sloppiness and stiffness.

## Predicting compensation mechanisms from posterior distributions

Experimental and computational studies have shown that stable neural activity can be maintained despite variable circuit parameters (*Prinz et al., 2004*; *Marder and Taylor, 2011*; *O'Leary, 2018*). This behavior can emerge from two sources (*Marder and Taylor, 2011*): either, the variation of a certain parameter barely influences network activity at all, or alternatively, variations of several parameters influence network activity, but their effects compensate for one another. Here, we investigated these possibilities by using the posterior distribution over membrane and synaptic conductances of the STG.

We began by drawing samples from the posterior and inspecting their pairwise histograms (i.e. the pairwise marginals, *Figure 6a*, posterior over all parameters in *Appendix 1—figure 11*). Consistent with previously reported results (*Taylor et al., 2009*), many parameters seem only weakly constrained and only weakly correlated (*Figure 6b*). However, this observation does not imply that the parameters of the network do not have to be finely tuned: pairwise marginals are averages over many network configurations, where all other parameters may take on diverse values, which could disguise that each individual configuration is finely tuned. Indeed, when we sampled parameters independently from their posterior histograms, the resulting circuit configurations rarely produced pyloric activity, indicating that parameters have to be tuned relative to each other (*Appendix 1—figure 14*). This analysis also illustrates that the (common) approach of independently setting parameters can be problematic: although each parameter individually is in a realistic range, the network as a whole is not (*Golowasch et al., 2002*). Finally, it shows the importance of identifying the full posterior distribution, which is far more informative than just finding individual parameters and assigning error bars.

In order to investigate the need for tuning between pairs of parameters, we held all but two parameters constant at a given consistent circuit configuration (sampled from the posterior), and observed the network activity across different values of the remaining pair of parameters. We can do so by calculating the conditional posterior distribution (details in Materials and methods), and do not have to generate additional simulations (as would be required by parameter search methods). Doing so has a simple interpretation: when all but two parameters are fixed, what values of the remaining two parameters can then lead to the desired network activity? We found that the desired pattern of pyloric activity can emerge only from narrowly tuned and often highly correlated combinations of the remaining two parameters, showing how these parameters can compensate for one another (*Figure 6c*). When repeating this analysis across multiple network configurations, we found that these 'conditional correlations' are often preserved (*Figure 6c*, left and right). This demonstrates that pairs of parameters can compensate for each other in a similar way, independently of the values taken by other parameters. This observation about compensation could be interpreted as an instance of modularity, a widespread underlying principle of biological robustness (*Kitano, 2004*).

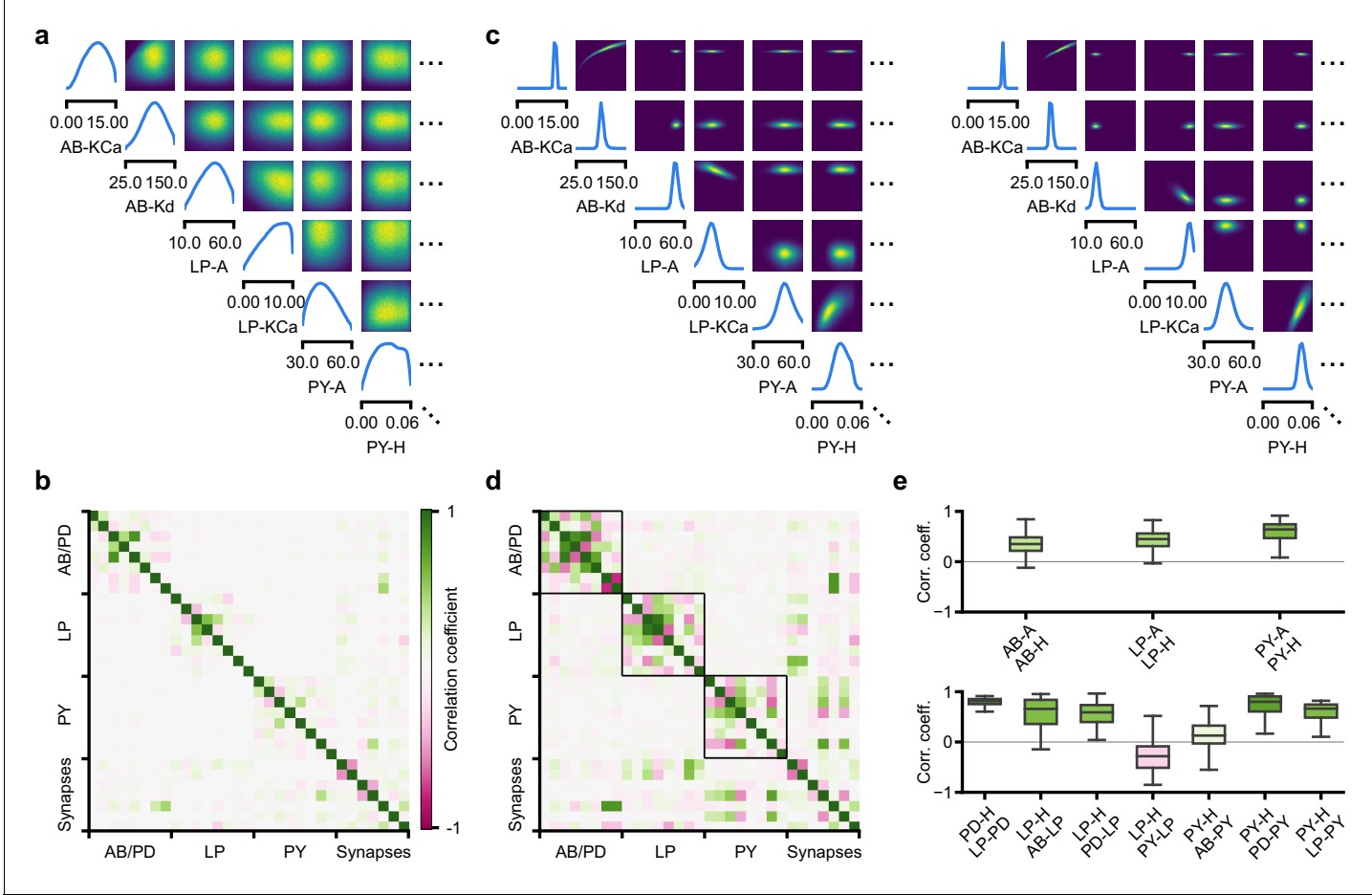

**Figure 6.** Predicting compensation mechanisms in the stomatogastric ganglion. (a) Inferred posterior. We show a subset of parameters which are weakly constrained (full posterior in *Appendix 1—figure 11*). Pyloric activity can emerge from a wide range of maximal membrane conductances, as the 1D and 2D posterior marginals cover almost the entire extent of the prior. (b) Correlation matrix, based on the samples shown in panel (a). Almost all correlations are weak. Ordering of membrane and synaptic conductances as in *Appendix 1—figure 11*. (c) Conditional distributions given a particular circuit configuration: for the plots on the diagonal, we keep all but one parameter fixed. For plots above the diagonal, we keep all but two parameters fixed. The remaining parameter(s) are narrowly tuned; tuning across parameters is often highly correlated. When conditioning on a different parameter setting (right plot), the conditional posteriors change, but correlations are often maintained. (d) Conditional correlation matrix, averaged over 500 conditional distributions like the ones shown in panel (c). Black squares highlight parameter-pairs within the same model neuron. (e) Consistency with experimental observations. Top: maximal conductance of the fast transient potassium current and the maximal conductance of the hyperpolarization current are positively correlated for all three neurons. This has also been experimentally observed in the PD and the LP neuron (*MacLean et al., 2005*). Bottom: the maximal conductance of the hyperpolarization current of the postsynaptic neuron can compensate the strength of the synaptic input, as experimentally observed in the PD and the LP neuron (*Grashow et al., 2010*; *Marder, 2011*). The boxplots indicate the maximum, 75% quantile, median, 25% quantile, and minimum across 500 conditional correlations for different parameter pairs. Face color indicates mean correlation using the colorbar shown in panel (b).

We calculated conditional correlations for each parameter pair using 500 different circuit configurations sampled from the posterior (*Figure 6d*). Compared to correlations based on the pairwise marginals (*Figure 6b*), these conditional correlations were substantially stronger. They were particularly strong across membrane conductances of the same neuron, but primarily weak across different neurons (black boxes in *Figure 6d*).

Finally, we tested whether the conditional correlations were in line with experimental observations. For the PD and the LP neuron, it has been reported that overexpression of the fast transient potassium current ($I_A$) leads to a compensating increase of the hyperpolarization current ($I_H$), suggesting a positive correlation between these two currents (*MacLean et al., 2003*; *MacLean et al., 2005*). These results are qualitatively consistent with the positive conditional correlations between the maximal conductances of $I_A$ and $I_H$ for all three model neurons (*Figure 6e* top). In addition,

using the dynamic clamp, it has been shown that diverse combinations of the synaptic input strength and the maximal conductance of $I_H$ lead to similar activity in the LP and the PD neuron (*Grashow et al., 2010*; *Marder, 2011*). Consistent with these findings, the non-zero conditional correlations reveal that there can indeed be compensation mechanisms between the synaptic strength and the maximal conductance of $I_H$ of the postsynaptic neuron (*Figure 6e* bottom).

Overall, we showed how SNPE can be used to study parameter dependencies, and how the posterior distribution can be used to efficiently explore potential compensation mechanisms. We found that our method can predict compensation mechanisms which are qualitatively consistent with experimental studies. We emphasize that these findings would not have been possible with a direct grid-search over all parameters: defining a grid in a 31-dimensional parameter space would require more than $2^{31} > 2$ billion simulations, even if one were to use the coarsest-possible grid with only two values per dimension.

## Discussion

How can we build models which give insights into the causal mechanisms underlying neural or behavioral dynamics? The cycle of building mechanistic models, generating predictions, comparing them to empirical data, and rejecting or refining models has been of crucial importance in the empirical sciences. However, a key challenge has been the difficulty of identifying mechanistic models which can quantitatively capture observed phenomena. We suggest that a generally applicable tool to constrain mechanistic models by data would expedite progress in neuroscience. While many considerations should go into designing a model that is appropriate for a given question and level of description (*Herz et al., 2006*; *Brette, 2015*; *Gerstner et al., 2012*; *O'Leary et al., 2015*), the question of whether and how one can perform statistical inference should not compromise model design. In our tool, SNPE, the process of model building and parameter inference are entirely decoupled. SNPE can be applied to *any* simulation-based model (requiring neither model nor summary features to be differentiable) and gives full flexibility on defining a prior. We illustrated the power of our approach on a diverse set of applications, highlighting the potential of SNPE to rapidly identify data-compatible mechanistic models, to investigate which data-features effectively constrain parameters, and to reveal shortcomings of candidate-models.

Finally, we used a model of the stomatogastric ganglion to show how SNPE can identify complex, high-dimensional parameter landscapes of neural systems. We analyzed the geometrical structure of the parameter landscape and confirmed that circuit configurations need to be finely tuned, even if individual parameters can take on a broad range of values. We showed that different configurations are connected in parameter space, and provided hypotheses for compensation mechanisms. These analyses were made possible by SNPE's ability to estimate full parameter posteriors, rather than just constraints on individual parameters, as is common in many statistical parameter-identification approaches.

### Related work

SNPE builds on recent advances in machine learning and in particular in density-estimation approaches to likelihood-free inference (*Papamakarios and Murray, 2016*; *Le et al., 2017a*; *Lueckmann et al., 2017*; *Chan et al., 2018*; *Greenberg et al., 2019*, reviewed in *Cranmer et al., 2020*). We here scaled these approaches to canonical mechanistic models of neural dynamics and provided methods and software-tools for inference, visualization, and analysis of the resulting posteriors (e.g. the high-probability paths and conditional correlations presented here).

The idea of learning inference networks on simulated data can be traced back to *regression-adjustment* methods in ABC (*Beaumont et al., 2002*; *Blum and François, 2010*). *Papamakarios and Murray, 2016* first proposed to use expressive conditional density estimators in the form of deep neural networks (*Bishop, 1994*; *Papamakarios et al., 2017*), and to optimize them sequentially over multiple rounds with cost-functions derived from Bayesian inference principles. Compared to commonly used rejection-based ABC methods (*Rubin, 1984*; *Pritchard et al., 1999*), such as MCMC-ABC (*Marjoram et al., 2003*), SMC-ABC (*Sisson et al., 2007*; *Liepe et al., 2014*), Bayesian-Optimization ABC (*Gutmann and Corander, 2016*), or ensemble methods (*Britton et al., 2013*; *Lawson et al., 2018*), SNPE approaches do not require one to define a distance function in data space. In addition, by leveraging the ability of neural networks to learn informative features, they

enable scaling to problems with high-dimensional observations, as are common in neuroscience and other fields in biology. We have illustrated this capability in the context of receptive field estimation, where a convolutional neural network extracts summary features from a 1681 dimensional spike-triggered average. Alternative likelihood-free approaches include *synthetic likelihood* methods (*Wood, 2010*; *Costa et al., 2013*; *Wilkinson, 2014*; *Meeds and Welling, 2014*; *Papamakarios et al., 2019a*; *Lueckmann et al., 2019*; *Durkan et al., 2018*), moment-based approximations of the posterior (*Barthelmé and Chopin, 2014*; *Schröder et al., 2019*), inference compilation (*Le et al., 2017b*; *Casado et al., 2017*), and density-ratio estimation (*Hermans et al., 2020*). For some mechanistic models in neuroscience (e.g. for integrate-and-fire neurons), likelihoods can be computed via stochastic numerical approximations (*Chen, 2003*; *Huys and Paninski, 2009*; *Meliza et al., 2014*) or model-specific analytical approaches (*Huys et al., 2006*; *Hertäg et al., 2012*; *Pozzorini et al., 2015*; *Ladenbauer et al., 2018*; *René et al., 2020*).

How big is the advance brought by SNPE relative to 'conventional' brute-force approaches that aim to exhaustively explore parameter space? A fundamental difference from grid search approaches that have been applied to neuroscientific models (*Prinz et al., 2003*; *Caplan et al., 2014*; *Stringer et al., 2016*) is that SNPE can perform Bayesian inference for *stochastic* models, whereas previous approaches identified parameters whose *deterministic* model-outputs were heuristically 'close' to empirical data. Depending on the goal of the analysis, either approach might be preferable. SNPE, and Bayesian inference more generally, is derived for stochastic models. SNPE can, in principle, also be applied to deterministic models, but a rigorous mathematical interpretation or empirical evaluation in this regime is beyond the scope of this study. SNPE also differs conceptually and quantitatively from rejection-ABC, in which random parameters are accepted or rejected based on a distance-criterion. SNPE uses *all* simulations during training instead of rejecting some, learns to identify data features informative about model parameters rather than relying on the user to choose the correct data features and distance metric, and performs considerably better than rejection-ABC, in particular for problems with high-dimensional observations (*Figure 2*). Another advantage over grid search and rejection-ABC is that SNPE can 'amortize' inference of parameter posteriors, so that one can quickly perform inference on new data, or explore compensation mechanisms, without having to carry out new simulations, or repeatedly search a simulation database. We should still note that SNPE can require the generation of large sets of simulations, which can be viewed as a brute-force step, emphasising that one of the main strengths of SNPE over conventional brute-force approaches relies on the processing of these simulations via deep neural density estimators.

Our approach is already finding its first applications in neuroscience–for example, *Oesterle et al., 2020* have used a variant of SNPE to constrain biophysical models of retinal neurons, with the goal of optimizing stimulation approaches for neuroprosthetics. Concurrently with our work, *Bittner et al., 2019* developed an alternative approach to parameter identification for mechanistic models and showed how it can be used to characterize neural population models which exhibit specific emergent computational properties. Both studies differ in their methodology and domain of applicability (see descriptions of underlying algorithms in our prior work [*Lueckmann et al., 2017*; *Greenberg et al., 2019*] and theirs [*Loaiza-Ganem et al., 2017*]), as well in the focus of their neuroscientific contributions. Both approaches share the overall goal of using deep probabilistic inference tools to build more interpretable models of neural data. These complementary and concurrent advances will expedite the cycle of building, adjusting and selecting mechanistic models in neuroscience.

Finally, a complementary approach to mechanistic modeling is to pursue purely phenomenological models, which are designed to have favorable statistical and computational properties: these data-driven models can be efficiently fit to neural data (*Brown et al., 1998*; *Truccolo et al., 2005*; *Pillow, 2007*; *Pillow et al., 2008*; *Schneidman et al., 2006*; *Macke et al., 2011*; *Yu et al., 2009*; *Pandarinath et al., 2018*; *Cunningham and Yu, 2014*) or to implement desired computations (*Sussillo and Abbott, 2009*). Although tremendously useful for a quantitative characterization of neural dynamics, these models typically have a large number of parameters, which rarely correspond to physically measurable or mechanistically interpretable quantities, and thus it can be challenging to derive mechanistic insights or causal hypotheses from them (but see e.g. *Mante et al., 2013*; *Sussillo and Barak, 2013*; *Maheswaranathan et al., 2019*).

## Use of summary features

When fitting mechanistic models to data, it is common to target summary features to isolate specific behaviors, rather than the full data. For example, the spike shape is known to constrain sodium and potassium conductances (*Druckmann et al., 2007*; *Pospischil et al., 2008*; *Hay et al., 2011*). When modeling population dynamics, it is often desirable to achieve realistic firing rates, rate-correlations and response nonlinearities (*Rubin et al., 2015*; *Bittner et al., 2019*), or specified oscillations (*Prinz et al., 2004*). In models of decision making, one is often interested in reproducing psychometric functions or reaction-time distributions (*Ratcliff and McKoon, 2008*). Choice of summary features might also be guided by known limitations of either the model or the measurement approach, or necessitated by the fact that published data are only available in summarized form. Several methods have been proposed to automatically construct informative summary features (*Blum et al., 2013*; *Jiang et al., 2017*; *Izbicki et al., 2019*). SNPE can be applied to, and might benefit from the use of summary features, but it also makes use of the ability of neural networks to automatically learn informative features in high-dimensional data. Thus, SNPE can also be applied directly to raw data (e.g. using recurrent neural networks [*Lueckmann et al., 2017*]), or to high-dimensional summary features which are challenging for ABC approaches (*Figure 2*). In all cases, care is needed when interpreting models fit to summary features, as choice of features can influence the results (*Blum et al., 2013*; *Jiang et al., 2017*; *Izbicki et al., 2019*).

## Applicability and limitations

A key advantage of SNPE is its general applicability: it can be applied whenever one has a simulator that allows to stochastically generate model outputs from specific parameters. Furthermore, it can be applied in a fully 'black-box manner', that is, does not require access to the internal workings of the simulator, its model equations, likelihoods or gradients. It does not impose any other limitations on the model or the summary features, and in particular does not require them to be differentiable. However, it also has limitations which we enumerate below.

First, current implementations of SNPE scale well to high-dimensional observations (~1000s of dimensions, also see *Greenberg et al., 2019*), but scaling SNPE to even higher-dimensional parameter spaces (above 30) is challenging (note that previous approaches were generally limited to less than 10 dimensions). Given that the difficulty of estimating full posteriors scales exponentially with dimensionality, this is an inherent challenge for all approaches that aim at full inference (in contrast to just identifying a single, or a few heuristically chosen parameter fits).

Second, while it is a long-term goal for these approaches to be made fully automatic, our current implementation still requires choices by the user: as described in Materials and methods, one needs to choose the type of the density estimation network, and specify settings related to network-optimization, and the number of simulations and inference rounds. These settings depend on the complexity of the relation between summary features and model parameters, and the number of simulations that can be afforded. In the documentation accompanying our code-package, we provide examples and guidance. For small-scale problems, we have found SNPE to be robust to these settings. However, for challenging, high-dimensional applications, SNPE might currently require substantial user interaction.

Third, the power of SNPE crucially rests on the ability of deep neural networks to perform density estimation. While deep nets have had ample empirical success, we still have an incomplete understanding of their limitations, in particular in cases where the mapping between data and parameters might not be smooth (e.g. near phase transitions).

Fourth, when applying SNPE (or any other model-identification approach), validation of the results is of crucial importance, both to assess the accuracy of the inference procedure, as well as to identify possible limitations of the mechanistic model itself. In the example applications, we used several procedures for assessing the quality of the inferred posteriors. One common ingredient of these approaches is to sample from the inferred model, and search for systematic differences between observed and simulated data, e.g. to perform *posterior predictive checks* (*Cook et al., 2006*; *Talts et al., 2018*; *Liepe et al., 2014*; *Lueckmann et al., 2017*; *Greenberg et al., 2019*; *Figure 2g*, *Figure 3f,g*, *Figure 4c*, and *Figure 5d*). These approaches allow one to detect 'failures' of SNPE, that is, cases in which samples from the posterior do not reproduce the data. However, when diagnosing any Bayesian inference approach, it is challenging to rigorously rule out the

possibility that additional parameter-settings (e.g. in an isolated 'island') would also explain the data. Thus, it is good practice to use multiple initializations of SNPE, and/or a large number of simulations in the initial round. There are challenges and opportunities ahead in further scaling and automating simulation-based inference approaches. However, in its current form, SNPE will be a powerful tool for quantitatively evaluating mechanistic hypotheses on neural data, and for designing better models of neural dynamics.

# Materials and methods

## Code availability

Code implementing SNPE based on Theano, is available at http://www.mackelab.org/delfi/. An extended toolbox based on PyTorch is available at http://www.mackelab.org/sbi/ (*Tejero-Cantero et al., 2020*).

## Simulation-based inference

To perform Bayesian parameter identification with SNPE, three types of input need to be specified:

1. A mechanistic model. The model only needs to be specified through a simulator, that is that one can generate a simulation result $\mathbf{x}$ for any parameters $\theta$. We do not assume access to the likelihood $p(\mathbf{x}|\theta)$ or the equations or internals of the code defining the model, nor do we require the model to be differentiable. This is in contrast to many alternative approaches (including *Bittner et al., 2019*), which require the model to be differentiable and to be implemented in a software code that is amenable to automatic differentiation packages. Finally, SNPE can both deal with inputs $\mathbf{x}$ which resemble 'raw' outputs of the model, or summary features calculated from data.
2. Observed data $\mathbf{x}_o$ of the same form as the results $\mathbf{x}$ produced by model simulations.
3. A prior distribution $p(\theta)$ describing the range of possible parameters. $p(\theta)$ could consist of upper and lower bounds for each parameter, or a more complex distribution incorporating mechanistic first principles or knowledge gained from previous inference procedures on other data. In our applications, we chose priors deemed reasonable or informed by previous studies (see Materials and methods), although setting such priors is an open problem in itself, and outside of the scope of this study.

For each problem, our goal was to estimate the posterior distribution $p(\theta|\mathbf{x}_o)$. To do this, we used SNPE (*Papamakarios and Murray, 2016*; *Lueckmann et al., 2017*; *Greenberg et al., 2019*). Setting up the inference procedure required three design choices:

1. A network architecture, including number of layers, units per layer, layer type (feedforward or convolutional), activation function and skip connections.
2. A parametric family of probability densities $q_\psi(\theta)$ to represent inferred posteriors, to be used as conditional density estimator. We used either a mixture of Gaussians (MoG) or a masked autoregressive flow (MAF) (*Papamakarios et al., 2017*). In the former case, the number of components $K$ must be specified; in the latter the number of *MADES* (Masked Autoencoder for Distribution Estimation) $n_{\mathrm{MADES}}$. Both choices are able to represent richly structured, and multimodal posterior distributions (more details on neural density estimation below).
3. A simulation budget, that is, number of rounds $R$ and simulations per round $N_r$. The required number of simulations depends on both the dimensionality and complexity of the function between summary statistics and model parameters. While the number of parameters and summary-features can easily be determined, it can be hard to determine how 'complex' (or nonlinear) this mapping is. This makes it difficult to give general guidelines on how many simulations will be required. A practical approach is to choose a simulation-budget based on the computational cost of the simulation, inspect the results (e.g. with posterior predictive checks), and add more simulations when it seems necessary.

We emphasize that SNPE is highly modular, that is, that the the inputs (data, the prior over parameter, the mechanistic model), and algorithmic components (network architecture, probability density, optimization approach) can all be modified and chosen independently. This allows neuroscientists to work with models which are designed with mechanistic principles—and not convenience of inference—in mind. Furthermore, it allows SNPE to benefit from advances in more flexible density estimators, more powerful network architectures, or optimization strategies.

With the problem and inference settings specified, SNPE adjusts the network weights $\phi$ based on simulation results, so that $p(\theta|\mathbf{x}) \approx q_{F(\mathbf{x},\phi)}(\theta)$ for any $\mathbf{x}$. In the first round of SNPE, simulation parameters are drawn from the prior $p(\theta)$. If a single round of inference is not sufficient, SNPE can be run in multiple rounds, in which samples are drawn from the version of $q_{F(\mathbf{x}_o,\phi)}(\theta)$ at the beginning of the round. After the last round, $q_{F(\mathbf{x}_o,\phi)}$ is returned as the inferred posterior on parameters $\theta$ given observed data $\mathbf{x}_o$. If SNPE is only run for a single round, then the generated samples only depend on the prior, but not on $\mathbf{x}_o$: in this case, the inference network is applicable to any data (covered by the prior ranges), and can be used for rapid amortized inference.

SNPE learns the correct network weights $\phi$ by minimizing the objective function $\sum_j \mathcal{L}(\theta_j, \mathbf{x}_j)$ where the simulation with parameters $\theta_j$ produced result $\mathbf{x}_j$. For the first round of SNPE $\mathcal{L}(\theta_j, \mathbf{x}_j) = -\log q_{F(\mathbf{x}_j,\phi)}$, while in subsequent rounds a different loss function accounts for the fact that simulation parameters were not sampled from the prior. Different choices of the loss function for later rounds result in SNPE-A (*Papamakarios and Murray, 2016*), SNPE-B (*Lueckmann et al., 2017*) or SNPE-C algorithm (*Greenberg et al., 2019*). To optimize the networks, we used ADAM with default settings (*Kingma and Ba, 2014*).

The details of the algorithm are below:

---

**Algorithm 1: SNPE**

---

**Input**: simulator with (implicit) density $p(\mathbf{x}|\theta)$, observed data $\mathbf{x}_o$, prior $p(\theta)$, density family $q_\psi$, neural network $F(\mathbf{x}, \phi)$, number of rounds , simulation count for each round $N_r$
randomly initialize $\phi$
$\tilde{p}_1(\theta) := p(\theta)$
$N := 0$
**for** $r = 1$ to $R$ **do**
 **for** $i = 1 \ldots N_r$ **do**
 sample $\theta_{N+i} \sim \tilde{p}_r(\theta)$
 simulate $\mathbf{x}_{N+i} \sim p(\mathbf{x}|\theta_{N+i})$
 $N \leftarrow N + N_r$
 train $\phi \leftarrow_\phi \sum_{j=1}^{N} \mathcal{L}(\theta_j, \mathbf{x}_j)$
 $\tilde{p}_r(\theta) := q_{F(\mathbf{x}_o,\phi)}(\theta)$
**return** $q_{F(\mathbf{x}_o,\phi)}(\theta)$

---

## Bayesian inference without likelihood-evaluations with SNPE

In *Papamakarios and Murray, 2016*, it was shown that the procedure described above (i.e. sample from the prior, train a flexible density estimator by minimizing the log-loss $\mathcal{L}(\theta_j, \mathbf{x}_j) = -\sum_j \log q_{F(\mathbf{x}_j,\phi)}(\theta_j)$) can be used to perform Bayesian inference without likelihood evaluations.

For the multi-round case, in which samples are no longer drawn from the prior, but adaptively generated from a (generally more focused) proposal distribution, the loss function needs to be modified. Different variants of SNPE differ in how exactly this is done:

- SNPE-A minimizes the same loss function as in the first round, but applies a post-hoc analytical correction (*Papamakarios and Murray, 2016*)
- SNPE-B minimizes an importance-weighted loss function, directly approximating the posterior and therefore not requiring a post-hoc correction (*Lueckmann et al., 2017*)
- SNPE-C avoids importance weights (which can have high variance), by either calculating normalization constants in closed-form or using a classifier-based loss (*Greenberg et al., 2019*)

## Neural density estimation

As described above, SNPE approximates the posterior distribution with flexible neural density estimators: either a mixture density network (MDN) or a masked autoregressive flow (MAF). Below, we provide a few more details about these density estimators, how we chose their respective architectures, and when to choose one or the other.

The MDN outputs the parameters of a mixture of Gaussians (i.e. mixture weights, and for each component of the mixture, the mean vector and covariance entries). Thus, for an MDN composed of

$K$ components, we chose an architecture with at least as many units per layer as $K(1 + N_\theta + N_\theta(N_\theta + 1)/2) - 1$, where $N_\theta$ is the number of parameters to infer, to ensure enough flexibility to approximate well the parameters of the mixture of Gaussians. For example, when inferring the parameters of the Hodgkin-Huxley model given in vitro recordings from mouse cortex (Allen Cell Types Database, https://celltypes.brain-map.org/data), we infer the posterior over eight parameters with a mixture of two Gaussians, and the MDN needs at least 89 units per layer. Across applications, we found two layers to be sufficient to appropriately approximate the posterior distribution.

MAF is a specific type of normalizing flow, which is a highly flexible density estimator (*Rezende and Mohamed, 2015*; *Papamakarios et al., 2017*; *Papamakarios et al., 2019b*). Normalizing flows consist of a stack of bijections which transform a simple distribution (usually a multivariate Gaussian distribution) into the target distribution. Each bijection is parameterized by a specific type of neural network (for MAF: a Masked Autoencoder for Distribution Estimation, or MADE). In our experiments, five stacked bijections are enough to approximate even complex posterior distributions. Depending on the size of the parameter and data space, each neural network had between [50,50] and [100,100,100] hidden units.

When using SNPE in a single-round, we generally found superior performance for MAFs as compared to MDNs. When running inference across multiple rounds, training MAFs leads to additional challenges which might impede the quality of inference (*Greenberg et al., 2019*; *Durkan et al., 2020*).

## Linear-nonlinear encoding models

We used a Linear-Nonlinear (LN) encoding model (a special case of a generalized linear model, GLM, [*Brown et al., 1998*; *Paninski, 2004*; *Truccolo et al., 2005*; *Pillow, 2007*; *Pillow et al., 2008*; *Gerwinn et al., 2010*]) to simulate the activity of a neuron in response to a univariate time-varying stimulus. Neural activity $z_i$ was subdivided in $T = 100$ bins and, within each bin $i$, spikes were generated according to a Bernoulli observation model,

$$z_i \sim \text{Bern}(\eta(\mathbf{v}_i^\top \boldsymbol{f} + \beta)),$$

where $\mathbf{v}_i$ is a vector of white noise inputs between time bins $i - 8$ and $i$, $\boldsymbol{f}$ a length-9 linear filter, $\beta$ is the bias, and $\eta(\cdot) = \exp(\cdot)/(1 + \exp(\cdot))$ is the canonical inverse link function for a Bernoulli GLM. As summary features, we used the total number of spikes $N$ and the spike-triggered average $\frac{1}{N}\mathbf{V}\mathbf{z}$, where $\mathbf{V} = [v_1, v_2, \ldots, v_T]$ is the so-called design matrix of size $9 \times T$. We note that the spike-triggered sum $\mathbf{V}\mathbf{z}$ constitutes sufficient statistics for this GLM, that is that selecting the STA and $N$ together as summary features does not lead to loss of model relevant information over the full input-output dataset $\{\mathbf{V}, \mathbf{z}\}$. We used a Gaussian prior with zero mean and covariance matrix $\boldsymbol{\Sigma} = \sigma^2(\mathbf{F}^\top\mathbf{F})^{-1}$, where $\mathbf{F}$ encourages smoothness by penalizing the second-order differences in the vector of parameters (*De Nicolao et al., 1997*).

For inference, we used a single round of 10,000 simulations, and the posterior was approximated with a Gaussian distribution ($\theta \in \mathbb{R}^{10}, \mathbf{x} \in \mathbb{R}^{10}$). We used a feedforward neural network with two hidden layers of 50 units each. We used a Polya Gamma Markov Chain Monte Carlo sampling scheme (*Polson et al., 2013*) to estimate a reference posterior.

In *Figure 2d*, we compare the performance of SNPE with two classical ABC algorithms, rejection ABC and Sequential Monte Carlo ABC as a function of the number of simulations. We report the relative error in Kullback-Leibler divergence, which is defined as:

$$\frac{D_{\text{KL}}(p_{MCMC}(\theta|\mathbf{x}) \,||\, \hat{p}(\theta|\mathbf{x}))}{D_{\text{KL}}(p_{MCMC}(\theta|\mathbf{x}) \,||\, p(\theta))}, \tag{1}$$

and which ranges between 0 (perfect recovery of the posterior) and 1 (estimated posterior no better than the prior). Here, $p_{MCMC}(\theta|\mathbf{x})$ is the ground-truth posterior estimated via Markov Chain Monte Carlo sampling, $\hat{p}(\theta|\mathbf{x})$ is the estimated posterior via SNPE, rejection ABC or Sequential Monte Carlo ABC, and $p(\theta)$ is the prior.

For the spatial receptive field model of a cell in primary visual cortex, we simulated the activity of a neuron depending on an image-valued stimulus. Neural activity was subdivided in bins of length $\Delta t = 0.025s$ and within each bin $i$, spikes were generated according to a Poisson observation model,

$$z_i \sim \mathrm{Poiss}(\eta(\mathbf{v}_i^\top \boldsymbol{h} + \beta)),$$

where $\mathbf{v}_i$ is the vectorized white noise stimulus at time bin $i$, $\boldsymbol{h}$ a $41 \times 41$ linear filter, $\beta$ is the bias, and $\eta(\cdot) = \exp(\cdot)$ is the canonical inverse link function for a Poisson GLM. The receptive field $\boldsymbol{h}$ is constrained to be a Gabor filter:

$$
\begin{aligned}
h(g_x, g_y) &= g \exp\left(-\frac{x'^2 + r^2 y'^2}{2\sigma^2}\right) \cos(2\pi f x' - \phi) \\
x' &= (g_x - x)\cos\psi - (g_y - y)\sin\psi \\
y' &= (g_x - x)\sin\psi + (g_y - y)\cos\psi \\
\sigma &= \frac{\sqrt{2}\log 2}{2\pi f}\frac{2^w + 1}{2^w - 1},
\end{aligned}
$$

where $(g_x, g_y)$ is a regular grid of $41 \times 41$ positions spanning the 2D image-valued stimulus. The parameters of the Gabor are gain $g$, spatial frequency $f$, aspect-ratio $r$, width $w$, phase $\phi$ (between 0 and $\pi$), angle $\psi$ (between 0 and $2\pi$) and location $x, y$ (assumed within the stimulated area, scaled to be between $-1$ and 1). Bounded parameters were transformed with a log-, or logit-transform, to yield unconstrained parameters. After applying SNPE, we back-transformed both the parameters and the estimated posteriors in closed form, as shown in *Figure 2*. We did not transform the bias $\beta$.

We used a factorizing Gaussian prior for the vector of transformed Gabor parameters

$$[\, \log g, \, \log f, \, \log r, \, \log w, \, l_{0,\pi}(\phi), \, l_{0,2\pi}(\psi), \, l_{-1,1}(x), \, l_{-1,1}(y) \,],$$

where transforms $l_{0,\pi}(X) = \log(X/(2\pi - X))$, $l_{0,2\pi}(X) = \log(X/(\pi - X))$, $l_{-1,1}(X) = \log((X+1)/(1-X))$ ensured the assumed ranges for the Gabor parameters $\phi, \psi, x, y$. Our Gaussian prior had zero mean and standard deviations $[0.5, 0.5, 0.5, 0.5, 1.9, 1.78, 1.78, 1.78]$. We note that a Gaussian prior on a logit-transformed random variable $\mathrm{logit} X$ with zero mean and standard deviation around 1.78 is close to a uniform prior over the original variable $X$. For the bias $\beta$, we used a Gaussian prior with mean $-0.57$ and variance 1.63, which approximately corresponds to an exponential prior $exp(\beta) \sim Exp(\lambda)$ with rate $\lambda = 1$ on the baseline firing rate $\exp(\beta)$ in absence of any stimulus.

The ground-truth parameters for the demonstration in *Figure 2* were chosen to give an asymptotic firing rate of 1 Hz for 5 min stimulation, resulting in 299 spikes, and a signal-to-noise ratio of $-12$dB.

As summary features, we used the total number of spikes $N$ and the spike-triggered average $\frac{1}{N}\mathbf{V}\mathbf{z}$, where $\mathbf{V} = [v_1, v_2, \ldots, v_T]$ is the stimulation video of length $T = 300/\Delta t = 12000$. As for the GLM with a temporal filter, the spike-triggered sum $\mathbf{V}\mathbf{z}$ constitutes sufficient statistics for this GLM.

For inference, we applied SNPE-A with in total two rounds: an initial round serves to first roughly identify the relevant region of parameter space. Here we used a Gaussian distribution to approximate the posterior from 100,000 simulations. A second round then used a mixture of eight Gaussian components to estimate the exact shape of the posterior from another 100,000 simulations ($\theta \in R^9, \mathbf{x} \in R^{1682}$). We used a convolutional network with five convolutional layers with 16 to 32 convolutional filters followed by two fully connected layers with 50 units each. The total number of spikes $N$ within a simulated experiment was passed as an additional input directly to the fully-connected layers of the network. Similar to the previous GLM, this model has a tractable likelihood, so we use MCMC to obtain a reference posterior.

We applied this approach to extracellular recordings from primary visual cortex of alert mice obtained using silicon microelectrodes in response to colored-noise visual stimulation. Experimental methods are described in *Dyballa et al., 2018*.

## Comparison with Sequential Monte Carlo (SMC) ABC

In order to illustrate the competitive performance of SNPE, we obtained a posterior estimate with a classical ABC method, Sequential Monte Carlo (SMC) ABC (*Sisson et al., 2007*; *Beaumont et al., 2009*). Likelihood-free inference methods from the ABC family require a distance function $d(\mathbf{x}_o, \mathbf{x})$ between observed data $\mathbf{x}_o$ and possible simulation outputs $\mathbf{x}$ to characterize dissimilarity between simulations and data. A common choice is the (scaled) Euclidean distance $d(\mathbf{x}_o, \mathbf{x}) = ||\mathbf{x} - \mathbf{x}_o||_2$. The Euclidean distance here was computed over 1681 summary features given by the spike-triggered average (one per pixel) and a single summary feature given by the 'spike count'. To ensure that the

distance measure was sensitive to differences in both STA and spike count, we scaled the summary feature 'spike count' to account for about 20% of the average total distance (other values did not yield better results). The other 80% were computed from the remaining 1681 summary features given by spike-triggered averages.

To showcase how this situation is challenging for ABC approaches, we generated 10,000 input-output pairs $(\theta_i, \mathbf{x}_i) \sim p(\mathbf{x}|\theta)p(\theta)$ with the prior and simulator used above, and illustrate the 10 STAs and spike counts with closest $d(\mathbf{x}_o, \mathbf{x}_i)$ in *Appendix 1—figure 5a*. Spike counts were comparable to the observed data (299 spikes), but STAs were noise-dominated and the 10 'closest' underlying receptive fields (orange contours) showed substantial variability in location and shape of the receptive field. If even the 'closest' samples do not show any visible receptive field, then there is little hope that even an appropriately chosen acceptance threshold will yield a good approximation to the posterior. These findings were also reflected in the results from SMC-ABC with a total simulation budget of $10^6$ simulations (*Appendix 1—figure 5b*). The estimated posterior marginals for 'bias' and 'gain' parameters show that the parameters related to the firing rate were constrained by the data $\mathbf{x}_o$, but marginals of parameters related to shape and location of the receptive field did not differ from the prior, highlighting that SMC-ABC was not able to identify the posterior distribution. The low correlations between the ground-truth receptive field and receptive fields sampled from SMC-ABC posterior further highlight the failure of SMC-ABC to infer the ground-truth posterior (*Appendix 1—figure 5c*). Further comparisons of neural-density estimation approaches with ABC-methods can be found in the studies describing the underlying machine-learning methodologies (*Papamakarios and Murray, 2016*; *Lueckmann et al., 2019*; *Greenberg et al., 2019*).

## Ion channel models

We simulated non-inactivating potassium channel currents subject to voltage-clamp protocols as:

$$I_K = \bar{g}_\mathrm{K} m (V - E_\mathrm{K}),$$

where $V$ is the membrane potential, $\bar{g}_\mathrm{K}$ is the density of potassium channels, $E_\mathrm{K}$ is the reversal potential of potassium, and $m$ is the gating variable for potassium channel activation. $m$ is modeled according to the first-order kinetic equation

$$\frac{dm}{dt} = \frac{m_\infty(V) - m}{\tau_m(V)},$$

where $m_\infty(V)$ is the steady-state activation, and $\tau_m(V)$ the respective time constant. We used a general formulation of $m_\infty(V)$ and $\tau_m(V)$ (*Destexhe and Huguenard, 2000*), where the steady-state activation curve has two parameters (slope and offset) and the time constant curve has six parameters, amounting to a total of 8 parameters ($\theta_1$ to $\theta_8$):

$$m_\infty(V) = \frac{1}{1 + e^{-\theta_1 V + \theta_2}}$$
$$\tau_m(V) = \frac{\theta_4}{e^{-[\theta_5(V-\theta_3) + \theta_6(V-\theta_3)^2]} + e^{[\theta_7(V-\theta_3) + \theta_8(V-\theta_3)^2]}}.$$

Since this model can be used to describe the dynamics of a wide variety of channel models, we refer to it as *Omnimodel*.

We modeled responses of the Omnimodel to a set of five noisy voltage-clamp protocols (*Podlaski et al., 2017*): as described in *Podlaski et al., 2017*, the original voltage-clamp protocols correspond to standard protocols of activation, inactivation, deactivation, ramp and action potential, to which we added Gaussian noise with zero mean and standard deviation 0.5 mV. Current responses were reduced to 55 summary features (11 per protocol). Summary features were coefficients to basis functions derived via Principal Components Analysis (PCA) (10 per protocol) plus a linear offset (one per protocol) found via least-squares fitting. PCA basis functions were found by simulating responses of the non-inactivating potassium channel models to the five voltage-clamp protocols and reducing responses to each protocol to 10 dimensions (explaining 99.9% of the variance).

To amortize inference on the model, we specified a wide uniform prior over the parameters: $\theta_1 \in \mathcal{U}(0,1), \theta_2 \in \mathcal{U}(-10., 10.),$ $\theta_3 \in \mathcal{U}(-120., 120.), \theta_4 \in \mathcal{U}(0., 2000.),$ $\theta_5 \in \mathcal{U}(0., 0.5), \theta_6 \in \mathcal{U}(0, 0.05),$ $\theta_7 \in \mathcal{U}(0., 0.5), \theta_8 \in \mathcal{U}(0, 0.05).$

For inference, we trained a shared inference network in a single round of $10^6$ simulations generated by sampling from the prior ($\theta \in R^8, \mathbf{x} \in R^{55}$). The density estimator was a masked autoregressive flow (MAF) (*Papamakarios et al., 2017*) with five MADES with [250,250] hidden units each.

We evaluated performance on 350 non-inactivating potassium ion channels selected from Ion-ChannelGenealogy (ICG) by calculating the correlation coefficient between traces generated by the original model and traces from the Omnimodel using the posterior mode (*Appendix 1—figure 7*).

## Single-compartment Hodgkin–Huxley neurons

We simulated a single-compartment Hodgkin–Huxley type neuron with channel kinetics as in *Pospischil et al., 2008*,

$$C_m \frac{dV}{dt} = g_l(E_l - V) + \bar{g}_{Na}m^3h(E_{Na} - V) + \bar{g}_K n^4(E_K - V) + \bar{g}_M p(E_K - V) + I_{inj} + \sigma\eta(t)$$
$$\frac{dq}{dt} = \frac{q_\infty(V) - q}{\tau_q(V)}, q \in \{m, h, n, p\},$$

where $V$ is the membrane potential, $C_m$ is the membrane capacitance, $g_l$ is the leak conductance, $E_l$ is the membrane reversal potential, $\bar{g}_c$ is the density of channels of type $c$ (Na$^+$, K$^+$, M), $E_c$ is the reversal potential of $c$, ($m$, $h$, $n$, $p$) are the respective channel gating kinetic variables, and $\sigma\eta(t)$ is the intrinsic neural Gaussian noise. The right hand side of the voltage dynamics is composed of a leak current, a voltage-dependent Na$^+$ current, a delayed-rectifier K$^+$ current, a slow voltage-dependent K$^+$ current responsible for spike-frequency adaptation, and an injected current $I_{inj}$. Channel gating variables $q$ have dynamics fully characterized by the neuron membrane potential $V$, given the respective steady-state $q_\infty(V)$ and time constant $\tau_q(V)$ (details in *Pospischil et al., 2008*). Two additional parameters are implicit in the functions $q_\infty(V)$ and $\tau_q(V)$: $V_T$ adjusts the spike threshold through $m_\infty$, $h_\infty$, $n_\infty$, $\tau_m$, $\tau_h$ and $\tau_n$; $\tau_{max}$ scales the time constant of adaptation through $\tau_p(V)$ (details in *Pospischil et al., 2008*). We set $E_{Na} = 53$ mV and $E_K = -107$ mV, similar to the values used for simulations in Allen Cell Types Database (http://help.brain-map.org/download/attachments/8323525/BiophysModelPeri.pdf).

We applied SNPE to infer the posterior over eight parameters ($\bar{g}_{Na}$, $\bar{g}_K$, $g_l$, $\bar{g}_M$, $\tau_{max}$, $V_T$, $\sigma$, $E_l$), given seven voltage features (number of spikes, mean resting potential, standard deviation of the resting potential, and the first four voltage moments, mean, standard deviation, skewness and kurtosis).

The prior distribution over the parameters was uniform,

$$\theta \sim \mathcal{U}(p_{low}, p_{high}),$$

where $p_{low} = [0.5, 10^{-4}, 10^{-4}, 10^{-4}, 50, 40, 10^{-4}, 35]$ and $p_{high} = [80, 15, 0.6, 0.6, 3000, 90, 0.15, 100]$. These ranges are similar to the ones obtained in *Pospischil et al., 2008*, when fitting the above model to a set of electrophysiological recordings.

For inference in simulated data, we used a single round of 100,000 simulations ($\theta \in R^8, \mathbf{x} \in R^7$). The density estimator was a masked autoregressive flow (MAF) (*Papamakarios et al., 2017*) with five MADES with [50,50] hidden units each.

For the inference on in vitro recordings from mouse cortex (Allen Cell Types Database, https://celltypes.brain-map.org/data), we selected eight recordings corresponding to spiny neurons with at least 10 spikes during the current-clamp stimulation. The respective cell identities and sweeps are: (518290966,57), (509881736,39), (566517779,46), (567399060,38), (569469018,44), (532571720,42), (555060623,34), (534524026,29). For each recording, SNPE-B was run for two rounds with 125,000 Hodgkin–Huxley simulations each, and the posterior was approximated by a mixture of two Gaussians. In this case, the density estimator was composed of two fully connected layers of 100 units each.

## Comparison with genetic algorithm

We compared SNPE posterior with a state-of-the-art genetic algorithm (Indicator Based Evolutionary Algorithm IBEA, [*Bleuler et al., 2003*; *Zitzler and Künzli, 2004*] from the BluePyOpt package [*Van Geit et al., 2016*]), in the context of the Hodgkin-Huxley model with 8 parameters and seven features (*Appendix 1—figure 10*). For each Hodgkin-Huxley model simulation $i$ and summary feature $j$, we used the following objective score:

$$\epsilon_{ij} = \left| \frac{x_{ij} - x_{oj}}{\sigma_j} \right|, j = 1, ..., 7,$$

where $x_{ij}$ is the value of summary feature $j$ for simulation $i$, $x_{oj}$ is the observed summary feature $j$, and $\sigma_j$ is the standard deviation of the summary feature $j$ computed across 1000 previously simulated datasets. IBEA outputs the hall-of-fame, which corresponds to the 10 parameter sets with the lowest sum of objectives $\sum_j^7 \epsilon_{ij}$. We ran IBEA with 100 generations and an offspring size of 1000 individuals, corresponding to a total of 100,000 simulations.

## Circuit model of the crustacean stomatogastric ganglion

We used extracellular nerve recordings made from the stomatogastric motor neurons that principally comprise the triphasic pyloric rhythm in the crab *Cancer borealis* (*Haddad and Marder, 2018*). The preparations were decentralized, that is, the axons of the descending modulatory inputs were severed. The data was recorded at a temperature of 11°C. See *Haddad and Marder, 2018* for full experimental details.

We simulated the circuit model of the crustacean stomatogastric ganglion by adapting a model described in *Prinz et al., 2004*. The model is composed of three single-compartment neurons, AB/PD, LP, and PD, where the electrically coupled AB and PD neurons are modeled as a single neuron. Each of the model neurons contains eight currents, a Na$^+$ current $I_{Na}$, a fast and a slow transient Ca$^{2+}$ current $I_{CaT}$ and $I_{CaS}$, a transient K$^+$ current $I_A$, a Ca$^{2+}$-dependent K$^+$ current $I_{KCa}$, a delayed rectifier K$^+$ current $I_{Kd}$, a hyperpolarization-activated inward current $I_H$, and a leak current $I_{leak}$. In addition, the model contains seven synapses. As in *Prinz et al., 2004*, these synapses were simulated using a standard model of synaptic dynamics (*Abbott and Marder, 1998*). The synaptic input current into the neurons is given by $I_s = g_s s (V_{post} - E_s)$, where $g_s$ is the maximal synapse conductance, $V_{post}$ the membrane potential of the postsynaptic neuron, and $E_s$ the reversal potential of the synapse. The evolution of the activation variable $s$ is given by

$$\frac{ds}{dt} = \frac{\bar{s}(V_{pre}) - s}{\tau_s}$$

with

$$\bar{s}(V_{pre}) = \frac{1}{1 + \exp((V_{th} - V_{pre})/\delta)} \quad \text{and} \quad \tau_s = \frac{1 - \bar{s}(V_{pre})}{k_-}.$$

Here, $V_{pre}$ is the membrane potential of the presynaptic neuron, $V_{th}$ is the half-activation voltage of the synapse, $\delta$ sets the slope of the activation curve, and $k_-$ is the rate constant for transmitter-receptor dissociation rate.

As in *Prinz et al., 2004*, two types of synapses were modeled since AB, LP, and PY are glutamatergic neurons whereas PD is cholinergic. We set $E_s = -70$ mV and $k_- = 1/40$ ms for all glutamatergic synapses and $E_s = -80$ mV and $k_- = 1/100$ ms for all cholinergic synapses. For both synapse types, we set $V_{th} = -35$ mV and $\delta = 5$ mV.

For each set of membrane and synaptic conductances, we numerically simulated the rhythm for 10 s with a step size of 0.025 ms. At each time step, each neuron received Gaussian noise with mean zero and standard deviation 0.001 mV.ms$^{-0.5}$.

We applied SNPE to infer the posterior over 24 membrane parameters and 7 synaptic parameters, that is, 31 parameters in total. The seven synaptic parameters were the maximal conductances $g_s$ of all synapses in the circuit, each of which was varied uniformly in logarithmic domain from $0.01\,\text{nS}$ to $1000\,\text{nS}$, with the exception of the synapse from AB to LP, which was varied uniformly in logarithmic domain from $0.01\,\text{nS}$ to $10000\,\text{nS}$. The membrane parameters were the maximal membrane

conductances for each of the neurons. The membrane conductances were varied over an extended range of previously reported values (*Prinz et al., 2004*), which led us to the uniform prior bounds $p_{\text{low}} = [0, 0, 0, 0, 0, 25, 0, 0]\text{mScm}^{-2}$ and $p_{\text{high}} = [500, 7.5, 8, 60, 15, 150, 0.2, 0.01]\text{mScm}^{-2}$ for the maximal membrane conductances of the AB neuron, $p_{\text{low}} = [0, 0, 2, 10, 0, 0, 0, 0.01]\text{mScm}^{-2}$ and $p_{\text{high}} = [200, 2.5, 12, 60, 10, 125, 0.06, 0.04]\text{mScm}^{-2}$ for the maximal membrane conductances of the LP neuron, and $p_{\text{low}} = [0, 0, 0, 30, 0, 50, 0, 0]\text{mScm}^{-2}$ and $p_{\text{high}} = [600, 12.5, 4, 60, 5, 150, 0.06, 0.04]\text{mScm}^{-2}$ for the maximal membrane conductances of the PY neuron. The order of the membrane currents was: [Na, CaT, CaS, A, KCa, Kd, H, leak].

We used the 15 summary features proposed by *Prinz et al., 2004*, and extended them by three additional features. The features proposed by *Prinz et al., 2004* are 15 salient features of the pyloric rhythm, namely: cycle period $T$ (s), AB/PD burst duration $d_{\text{AB}}^{\text{b}}$ (s), LP burst duration $d_{\text{LP}}^{\text{b}}$ (s), PY burst duration $d_{\text{PY}}^{\text{b}}$ (s), gap AB/PD end to LP start $\Delta t_{\text{AB}-\text{LP}}^{\text{es}}$ (s), gap LP end to PY start $\Delta t_{\text{LP}-\text{PY}}^{\text{es}}$ (s), delay AB/PD start to LP start $\Delta t_{\text{AB}-\text{LP}}^{\text{ss}}$ (s), delay LP start to PY start $\Delta t_{\text{LP}-\text{PY}}^{\text{ss}}$ (s), AB/PD duty cycle $d_{\text{AB}}$, LP duty cycle $d_{\text{LP}}$, PY duty cycle $d_{\text{PY}}$, phase gap AB/PD end to LP start $\Delta\phi_{\text{AB}-\text{LP}}$, phase gap LP end to PY start $\Delta\phi_{\text{LP}-\text{PY}}$, LP start phase $\phi_{\text{LP}}$, and PY start phase $\phi_{\text{PY}}$. Note that several of these values are only defined if each neuron produces rhythmic bursting behavior. In addition, for each of the three neurons, we used one feature that describes the maximal duration of its voltage being above $-30$ mV. We did this as we observed plateaus at around $-10$ mV during the onset of bursts, and wanted to distinguish such traces from others. If the maximal duration was below 5 ms, we set this feature to 5 ms. To extract the summary features from the observed experimental data, we first found spikes by searching for local maxima above a hand-picked voltage threshold, and then extracted the 15 above described features. We set the additional 3 features to 5 ms.

We used SNPE to infer the posterior distribution over the 18 summary features from experimental data. For inference, we used a single round with 18.5 million samples, out of which 174,000 samples contain bursts in all neurons. We therefore used these 174,000 samples with well defined summary features for training the inference network ($\theta \in R^{31}, \mathbf{x} \in R^{18}$). The density estimator was a masked autoregressive flow (MAF) (*Papamakarios et al., 2017*) with five MADES with [100,100,100] hidden units each. The synaptic conductances were transformed into logarithmic space before training and for the entire analysis.

Previous approaches for fitting the STG circuit (*Prinz et al., 2004*) first fit individual neuron features and reduce the number of possible neuron models (*Prinz et al., 2003*), and then fit the whole circuit model. While powerful, this approach both requires the availability of single-neuron data, and cannot give access to potential compensation mechanisms between single-neuron and synaptic parameters. Unlike *Prinz et al., 2004*, we apply SNPE to directly identify the full 31 dimensional parameter space without requiring experimental measurements of each individual neuron in the circuit. Despite the high-dimensional parameter space, SNPE can identify the posterior distribution using 18 million samples, whereas a direct application of a full-grid method would require $4.65 \cdot 10^{21}$ samples to fill the 31 dimensional parameter space on a grid with five values per dimension.

## Finding paths in the posterior

In order to find directions of robust network output, we searched for a path of high posterior probability. First, as in *Prinz et al., 2004*, we aimed to find two similar model outputs with disparate parameters. To do so, we sampled from the posterior and searched for two parameter sets whose summary features were within 0.1 standard deviations of all 174,000 samples from the observed experimental data, but that had strongly disparate parameters from each other. In the following, we denote the obtained parameter sets by $\theta_s$ and $\theta_g$.

Second, in order to identify whether network output can be maintained along a continuous path between these two samples, we searched for a connection in parameter space lying in regions of high posterior probability. To do so, we considered the connection between the samples as a path and minimized the following path integral:

$$\mathcal{L}(\gamma) = \int_0^1 -\log(p_{\theta|\mathbf{x}}(\gamma(s)|\mathbf{x_o}))\|\dot{\gamma}(s)\|ds. \tag{2}$$

To minimize this term, we parameterized the path $\gamma(s)$ using sinusoidal basis-functions with coefficients $\alpha_{n,k}$:

$$\gamma(s) = \begin{bmatrix} \sum_{k=1}^{K} \alpha_{1,k} \cdot \sin(\pi k s) \\ \vdots \\ \sum_{k=1}^{K} \alpha_{N,k} \cdot \sin(\pi k s) \end{bmatrix} + \begin{bmatrix} \sum_{k=K+1}^{2K} \alpha_{1,k} \cdot \sin^2(\pi k s) \\ \vdots \\ \sum_{k=K+1}^{2K} \alpha_{N,k} \cdot \sin^2(\pi k s) \end{bmatrix} + (1-s) \cdot \theta_s + s\theta_g$$

These basis functions are defined such that, for any coefficients $\alpha_{n,k}$, the start and end points of the path are exactly the two parameter sets defined above:

$$\gamma(0) = \theta_s \qquad \gamma(1) = \theta_g$$

With this formulation, we have framed the problem of finding the path as an unconstrained optimization problem over the parameters $\alpha_{n,k}$. We can therefore minimize the path integral $\mathcal{L}$ using gradient descent over $\alpha_{n,k}$. For numerical simulations, we approximated the integral in *Equation 2* as a sum over 80 points along the path and use two basis functions for each of the 31 dimensions, that is, $K = 2$.

In order to demonstrate the sensitivity of the pyloric network, we aimed to find a path along which the circuit output quickly breaks down. For this, we picked a starting point along the high-probability path and then minimized the posterior probability. In addition, we enforced that the orthogonal path lies within an orthogonal disk to the high-probability path, leading to the following constrained optimization problem:

$$\min_{\theta} \log(p(\theta|\mathbf{x})) \qquad \text{s.t.} \quad n^T \Delta\theta = 0$$

where $n$ is the tangent vector along the path of high probability. This optimization problem can be solved using the gradient projection method (*Rosen, 1960*):

$$\Delta\theta = -\frac{P(\nabla \log(p(\theta|\mathbf{x})))}{\sqrt{(\nabla \log(p(\theta|\mathbf{x})))^T P(\nabla \log(p(\theta|\mathbf{x})))}}$$

with projection matrix $P = \mathbb{1} - \frac{1}{n^T n} n n^T$ and $\mathbb{1}$ indicating the identity matrix. Each gradient update is a step along the orthogonal path. We let the optimization run until the distance along the path is 1/27 of the distance along the high-probability path.

## Identifying conditional correlations

In order to investigate compensation mechanisms in the STG, we compared marginal and conditional correlations. For the marginal correlation matrix in *Figure 6b*, we calculated the Pearson correlation coefficient based on 1.26 million samples from the posterior distribution $p(\theta|\mathbf{x})$. To find the two-dimensional conditional distribution for any pair of parameters, we fixed all other parameters to values taken from an arbitrary posterior sample, and varied the remaining two on an evenly spaced grid with 50 points along each dimension, covering the entire prior space. We evaluated the posterior distribution at every value on this grid. We then calculated the conditional correlation as the Pearson correlation coefficient over this distribution. For the 1-dimensional conditional distribution, we varied only one parameter and kept all others fixed. Lastly, in *Figure 6d*, we sampled 500 parameter sets from the posterior, computed the respective conditional posteriors and conditional correlation matrices, and took the average over the conditional correlation matrices.

## Acknowledgements

We thank Mahmood S Hoseini and Michael Stryker for sharing their data for *Figure 2*, and Philipp Berens, Sean Bittner, Jan Boelts, John Cunningham, Richard Gao, Scott Linderman, Eve Marder, Iain Murray, George Papamakarios, Astrid Prinz, Auguste Schulz and Srinivas Turaga for discussions and/or comments on the manuscript. This work was supported by the German Research Foundation (DFG) through SFB 1233 'Robust Vision', (276693517), SFB 1089 'Synaptic Microcircuits', SPP 2041 'Computational Connectomics' and Germany's Excellence Strategy – EXC-Number 2064/1 – Project number 390727645 and the German Federal Ministry of Education and Research (BMBF, project

'ADIMEM', FKZ 01IS18052 A-D) to JHM, a Sir Henry Dale Fellowship by the Wellcome Trust and the Royal Society (WT100000; WFP and TPV), a Wellcome Trust Senior Research Fellowship (214316/Z/18/Z; TPV), a ERC Consolidator Grant (SYNAPSEEK; WPF and CC), and a UK Research and Innovation, Biotechnology and Biological Sciences Research Council (CC, UKRI-BBSRC BB/N019512/1). We gratefully acknowledge the Leibniz Supercomputing Centre for funding this project by providing computing time on its Linux-Cluster.

## Additional information

### Funding

| Funder | Grant reference number | Author |
| --- | --- | --- |
| Deutsche Forschungsgemeinschaft | SFB 1233 | Jan-Matthis Lueckmann<br>Marcel Nonnenmacher<br>Giacomo Bassetto<br>Jakob H Macke |
| Deutsche Forschungsgemeinschaft | SFB 1089 | Pedro J Gonçalves<br>Jakob H Macke |
| Deutsche Forschungsgemeinschaft | SPP 2041 | Jakob H Macke |
| Bundesministerium für Bildung und Forschung | 01IS18052 A-D | Michael Deistler<br>Marcel Nonnenmacher<br>Jakob H Macke |
| H2020 European Research Council | SYNAPSEEK | Chaitanya Chintaluri<br>William F Podlaski<br>Tim P Vogels |
| Wellcome Trust Senior Research Fellowship | 214316/Z/18/Z | Tim P Vogels |
| UK Research and Innovation | UKRI-BBSRC BB/N019512/1 | Chaitanya Chintaluri |
| Deutsche Forschungsgemeinschaft | Germany's Excellence Strategy - EXC-Number 2064/1 610 - Project number 390727645 | Jakob H Macke |
| Wellcome Trust | Sir Henry Dale Fellowship WT100000 | William F Podlaski<br>Tim P Vogels |
| Royal Society | Sir Henry Dale Fellowship WT100000 | William F Podlaski<br>Tim P Vogels |

The funders had no role in study design, data collection and interpretation, or the decision to submit the work for publication.

### Author contributions

Pedro J Gonçalves, Conceptualization, Data curation, Software, Formal analysis, Supervision, Validation, Investigation, Visualization, Methodology, Writing - original draft, Project administration, Writing - review and editing; Jan-Matthis Lueckmann, Michael Deistler, Conceptualization, Data curation, Software, Formal analysis, Validation, Investigation, Visualization, Methodology, Writing - original draft, Writing - review and editing; Marcel Nonnenmacher, Data curation, Software, Formal analysis, Validation, Investigation, Visualization, Methodology, Writing - review and editing; Kaan Öcal, Software, Investigation, Visualization, Methodology, Writing - review and editing; Giacomo Bassetto, Software, Visualization, Methodology, Writing - review and editing; Chaitanya Chintaluri, William F Podlaski, Sara A Haddad, Resources, Data curation, Writing - review and editing; Tim P Vogels, Conceptualization, Resources, Supervision, Funding acquisition, Project administration, Writing - review and editing; David S Greenberg, Data curation, Software, Formal analysis, Supervision, Validation, Investigation, Visualization, Methodology, Writing - original draft, Writing - review and editing; Jakob H Macke, Conceptualization, Supervision, Funding acquisition, Validation, Visualization, Methodology, Writing - original draft, Project administration, Writing - review and editing

## Author ORCIDs

Pedro J Gonçalves (ID) https://orcid.org/0000-0002-6987-4836
Jan-Matthis Lueckmann (ID) https://orcid.org/0000-0003-4320-4663
Michael Deistler (ID) https://orcid.org/0000-0002-3573-0404
Kaan Öcal (ID) http://orcid.org/0000-0002-8528-6858
Chaitanya Chintaluri (ID) http://orcid.org/0000-0003-4252-1608
William F Podlaski (ID) http://orcid.org/0000-0001-6619-7502
Sara A Haddad (ID) https://orcid.org/0000-0003-0807-0823
Jakob H Macke (ID) https://orcid.org/0000-0001-5154-8912

## Decision letter and Author response

Decision letter https://doi.org/10.7554/eLife.56261.sa1
Author response https://doi.org/10.7554/eLife.56261.sa2

# Additional files

## Supplementary files

• Transparent reporting form

## Data availability

The contributions of the work are primarily the development and application of computational models, no new data has been obtained or is being published. All code and associated data are available at https://github.com/mackelab/delfi/ (archived at https://archive.softwareheritage.org/swh:1:rev: 62a99a879145bdc675917fc33eed69293b964048;origin=https://github.com/mackelab/delfi/;visit= swh:1:snp:95a69f58c195feb5660760cd4ab833de366f5441/) and https://github.com/mackelab/Iden- tifyMechanisticModels_2020 (archived at https://archive.softwareheritage.org/swh:1:rev: b93c90ec6156ae5f8afee6aaac7317373e9caf5e;origin=https://github.com/mackelab/IdentifyMecha- nisticModels_2020;visit=swh:1:snp:20bfde9fbbb134657b75bc29ab4905a2ef3d5b17/).

The following datasets were generated:

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

## Appendix 1

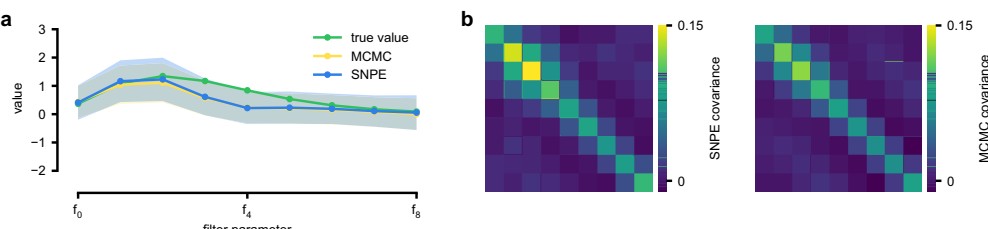

**Appendix 1—figure 1.** Comparison between SNPE-estimated posterior and reference posterior (obtained via MCMC) on LN model. (**a**) Posterior mean ± one standard deviation of temporal filter (receptive field) from SNPE posterior (SNPE, blue) and reference posterior (MCMC, yellow). (**b**) Full covariance matrices from SNPE posterior (left) and reference (MCMC, right).

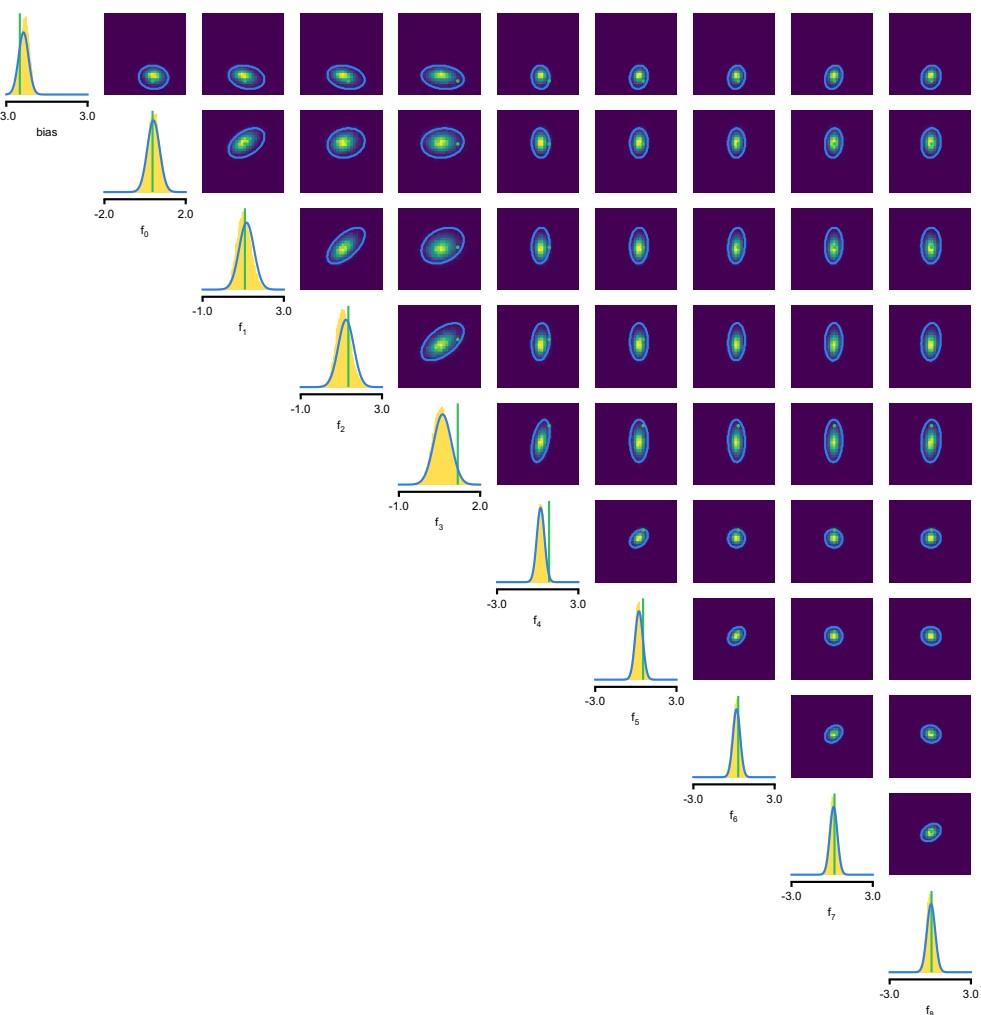

**Appendix 1—figure 2.** Full posterior for LN model. In green, ground-truth parameters. Marginals (blue lines) and 2D marginals for SNPE (contour lines correspond to 95% of the mass) and MCMC (yellow histograms).

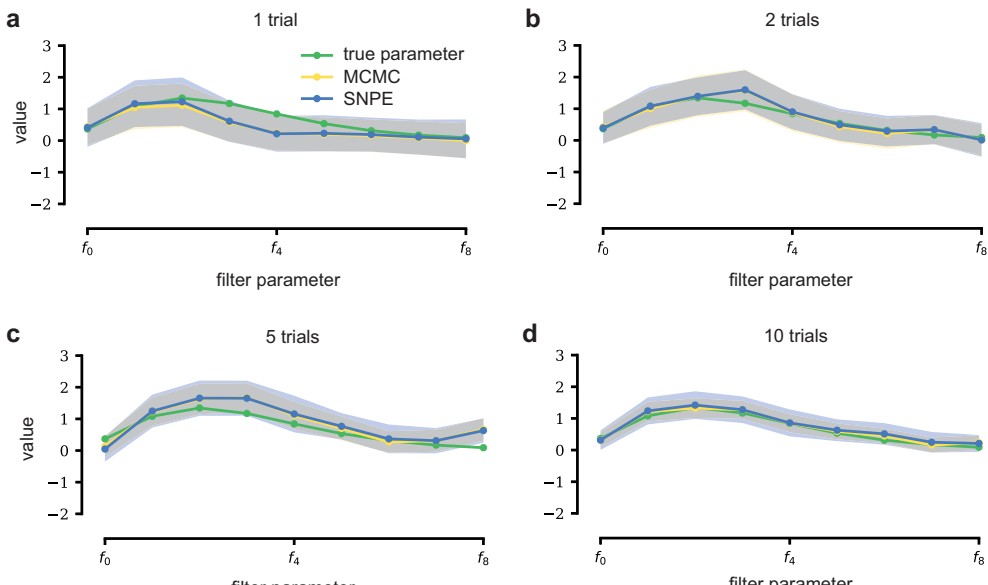

**Appendix 1—figure 3.** LN model with additional data. With additional data, posterior samples cluster more tightly around the true filter. From left to right and top to bottom, SNPE (blue) and MCMC (yellow, for reference) are applied to observations with more independent Bernoulli trials, leading to progressively tighter posteriors and posterior samples closer to the true filter (which is the same across panels). Mean ± one standard deviation is shown. Note that SNPE closely agrees with the MCMC reference solution in all cases (**a-d**).

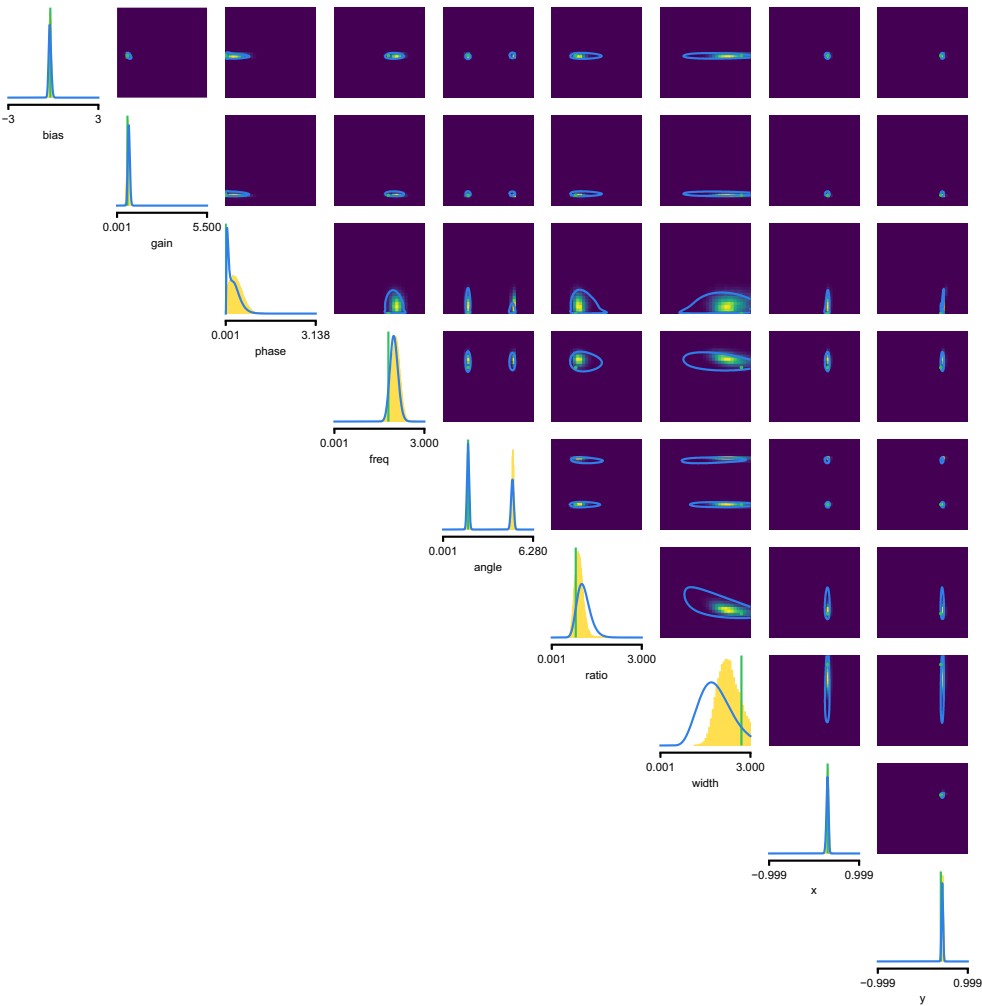

**Appendix 1—figure 4.** Full posterior for Gabor GLM receptive field model. SNPE posterior estimate (blue lines) compared to reference posterior (MCMC, histograms). Ground-truth parameters used to simulate the data in green. We depict the distributions over the original receptive field parameters, whereas we estimate the posterior as a Gaussian mixture over transformed parameters, see Materials and methods for details. We find that a (back-transformed) Gaussian mixture with four components approximates the posterior well in this case.

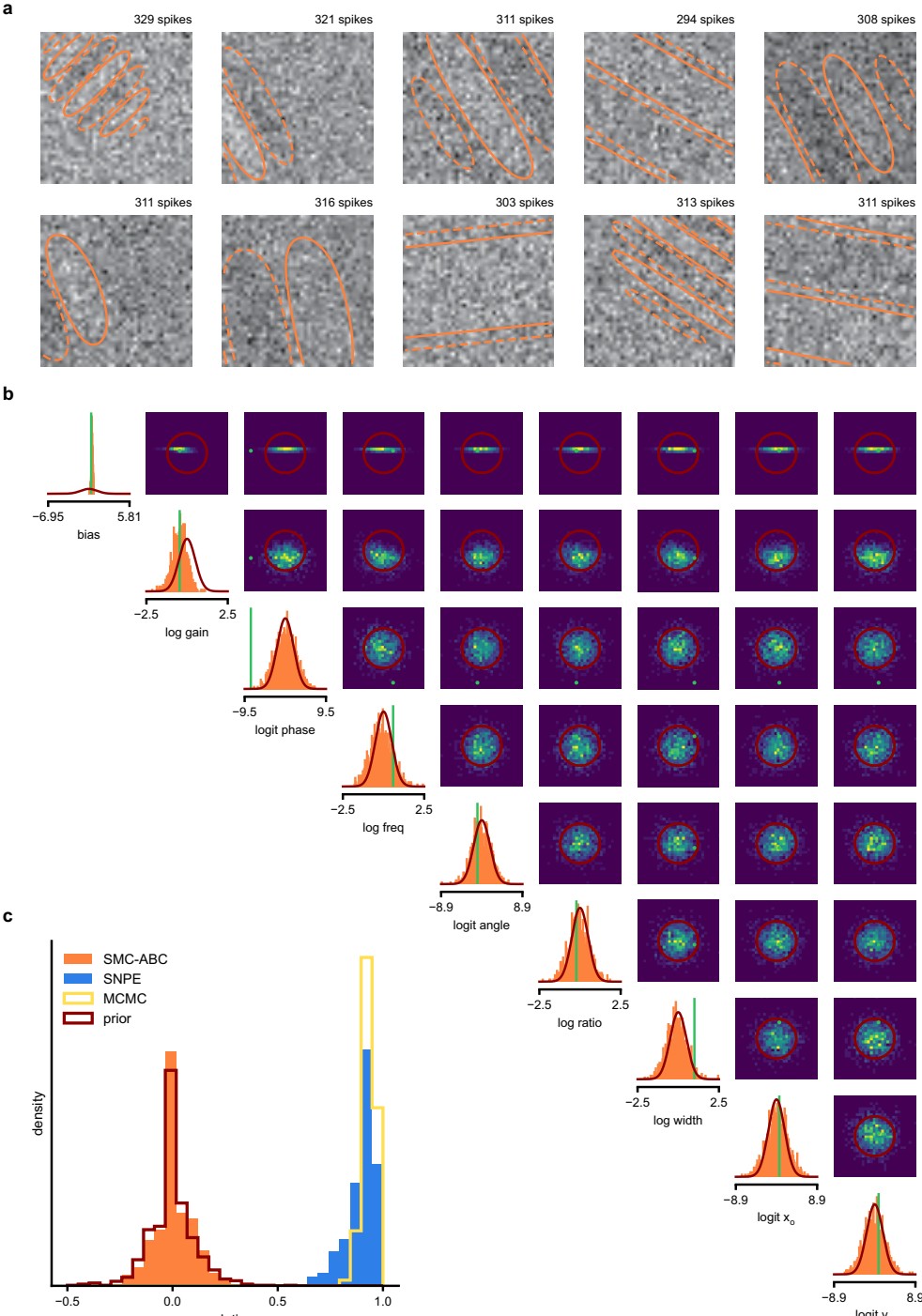

**Appendix 1—figure 5.** SMC-ABC posterior estimate for Gabor GLM receptive field model. (**a**) Spike-triggered averages (STAs) and spike counts with closest distance to the observed data out of 10000 simulations with parameters sampled from the prior. Spike counts are comparable to the observed data (299 spikes), but receptive fields (contours) are not well constrained. (**b**) Results for SMC-ABC with a total of $10^6$ simulations. Histograms of 1000 particles (orange) returned in the final iteration of SMC-ABC, compared to prior (red contour lines) and ground-truth parameters (green). Distributions over (log-/logit-)transformed parameters, axis limits scaled to mean ± 3 standard deviations of the prior. (**c**) Correlations between ground-truth receptive field and receptive fields sampled from SMC-ABC posterior (orange), SNPE posterior (blue), reference MCMC posterior

(yellow) and prior (red). The SNPE-estimated receptive fields are almost as good as those of the reference posterior, the SMC-ABC estimated ones no better than the prior.

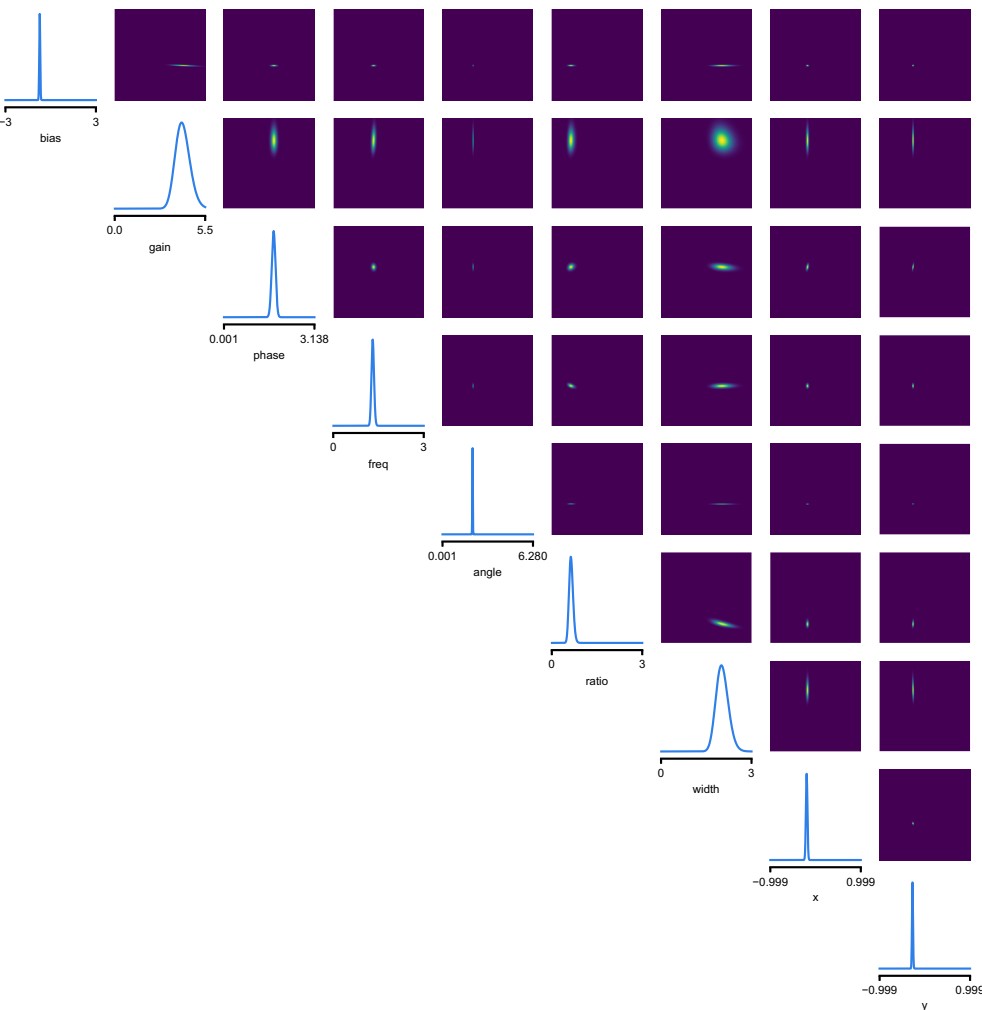

**Appendix 1—figure 6.** Full posterior for Gabor LN receptive field model on V1 recordings. We depict the distributions over the receptive field parameters, derived from the Gaussian mixture over transformed-parameters (see Materials and methods for details).

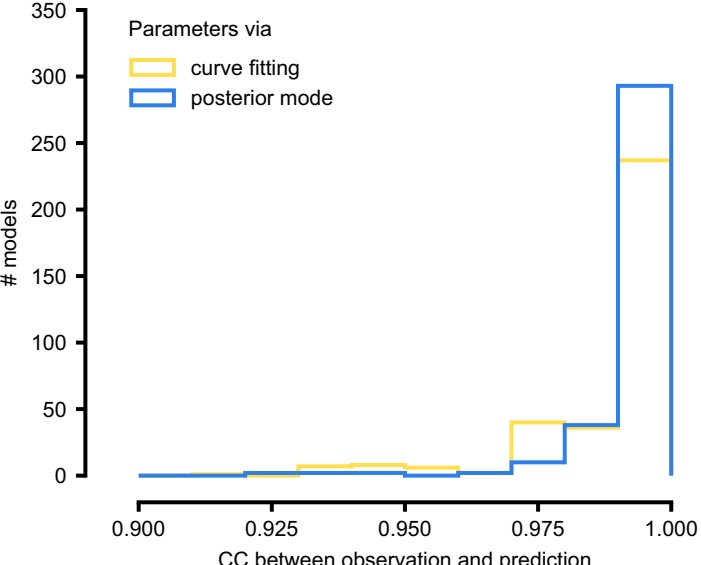

**Appendix 1—figure 7.** Summary results on 350 ICG channel models, and comparison with direct fits. We generate predictions either with the posterior mode (blue) or with parameters obtained by directly fitting steady-state activation and time-constant curves (yellow). We calculate the correlation coefficient (CC) between observation and prediction. The distribution of CCs is similar for both approaches.

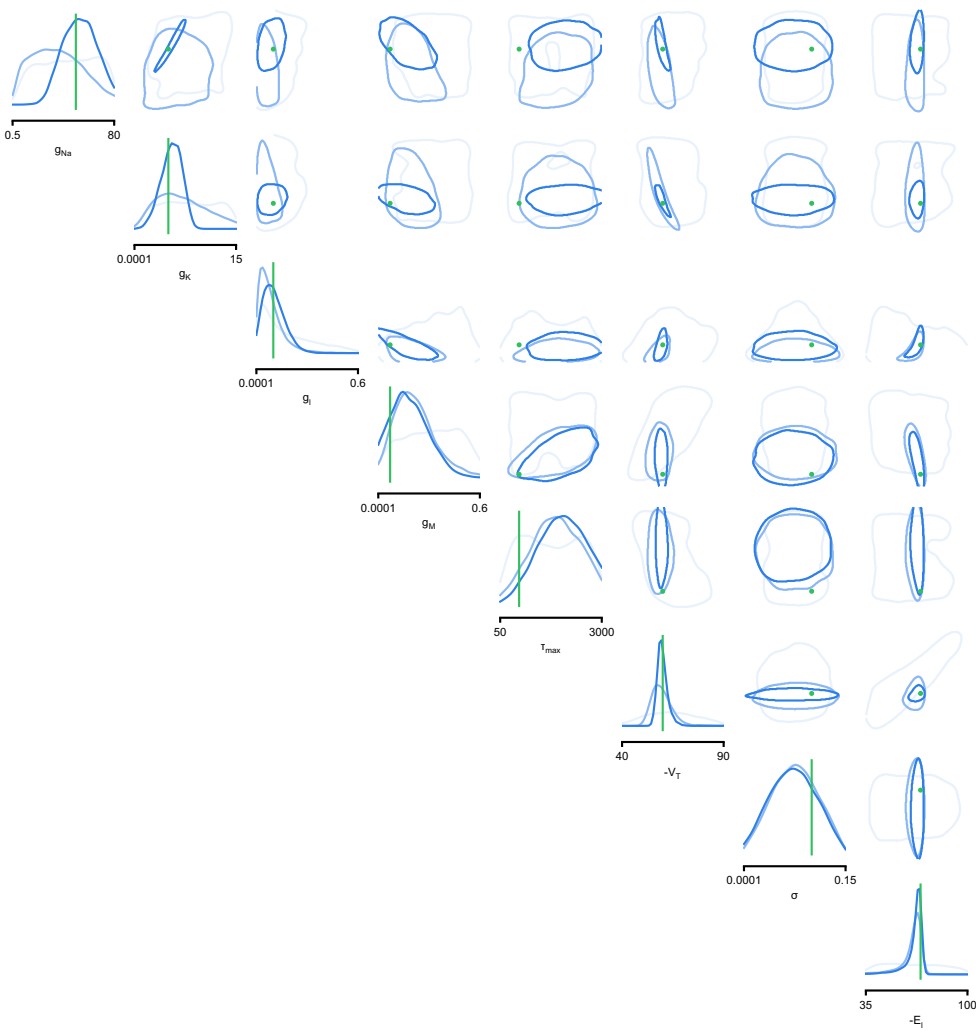

**Appendix 1—figure 8.** Full posteriors for Hodgkin-Huxley model for 1, 4, and 7 features. Images show the pairwise marginals for 7 features. Each contour line corresponds to 68% density mass for a different inferred posterior. Light blue corresponds to 1 feature and dark blue to 7 features. Ground truth parameters in green.

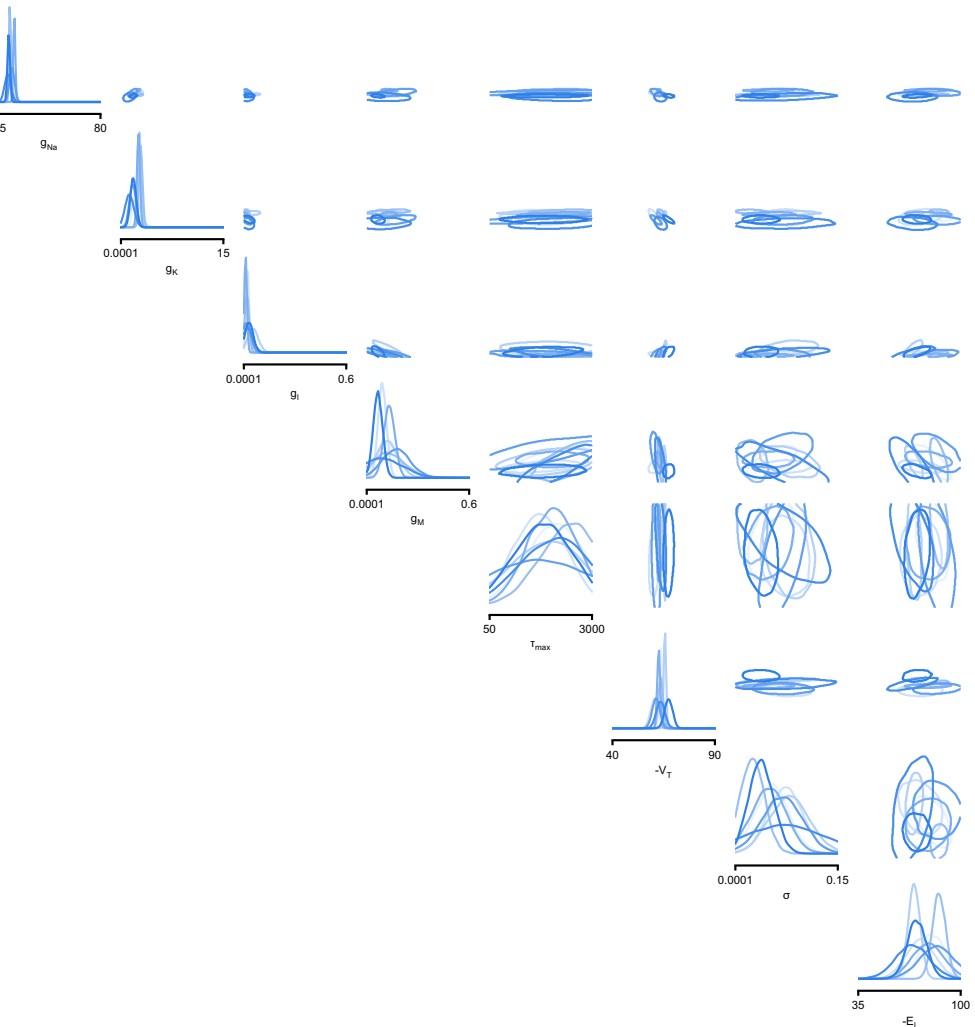

**Appendix 1—figure 9.** Full posteriors for Hodgkin-Huxley model on 8 different recordings from Allen Cell Type Database. Images show the pairwise marginals for 7 features. Each contour line corresponds to 68% density mass for a different inferred posterior.

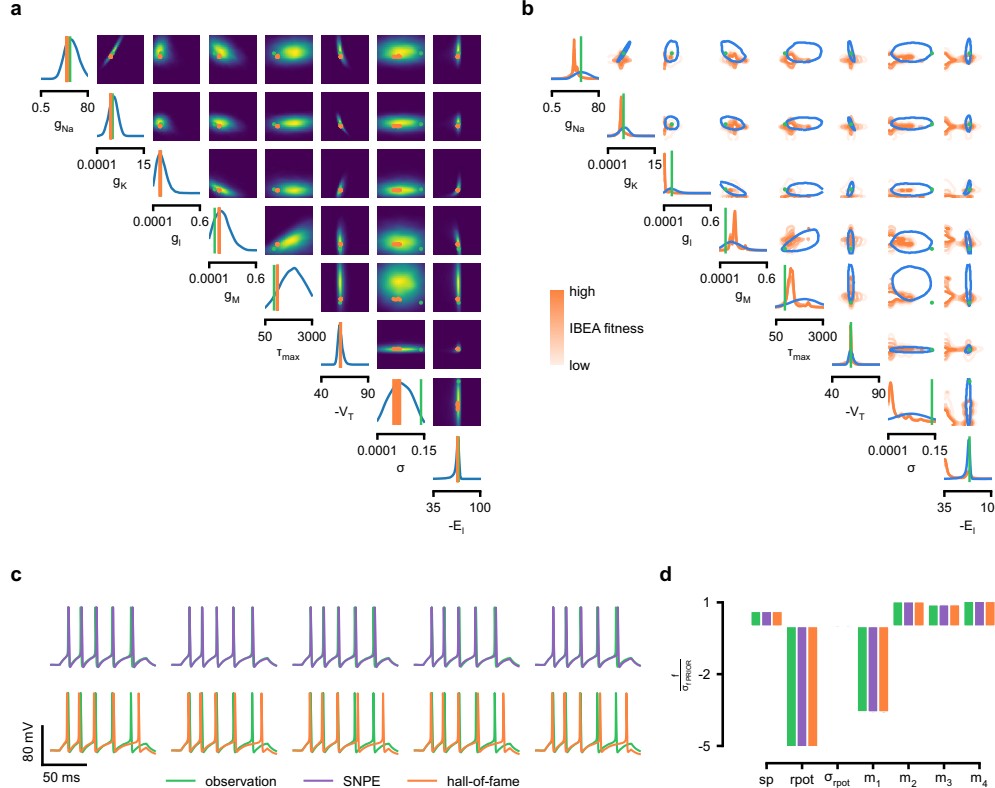

**Appendix 1—figure 10.** Comparison between SNPE posterior and IBEA samples for Hodgkin-Huxley model with 8 parameters and 7 features. (**a**) Full SNPE posterior distribution. Ground truth parameters in green and IBEA 10 parameters with highest fitness ('hall-of-fame') in orange. (**b**) Blue contour line corresponds to 68% density mass for SNPE posterior. Light orange corresponds to IBEA sampled parameters with lowest IBEA fitness and dark orange to IBEA sampled parameters with highest IBEA fitness. This plot shows that, in general, SNPE and IBEA can return very different answers– this is not surprising, as both algorithms have different objectives, but this highlights that genetic algorithms do not in general perform statistical inference. (**c**) Traces for samples with high probability under SNPE posterior (purple), and for samples with high fitness under IBEA objective (hall-of-fame; orange traces). (**d**) Features for the desired output (observation), the mode of the inferred posterior (purple) and the best sample under IBEA objective (orange). Each voltage feature is normalized by $\sigma_{\mathbb{PRIOR}}$, the standard deviation of the respective feature of simulations sampled from the prior.

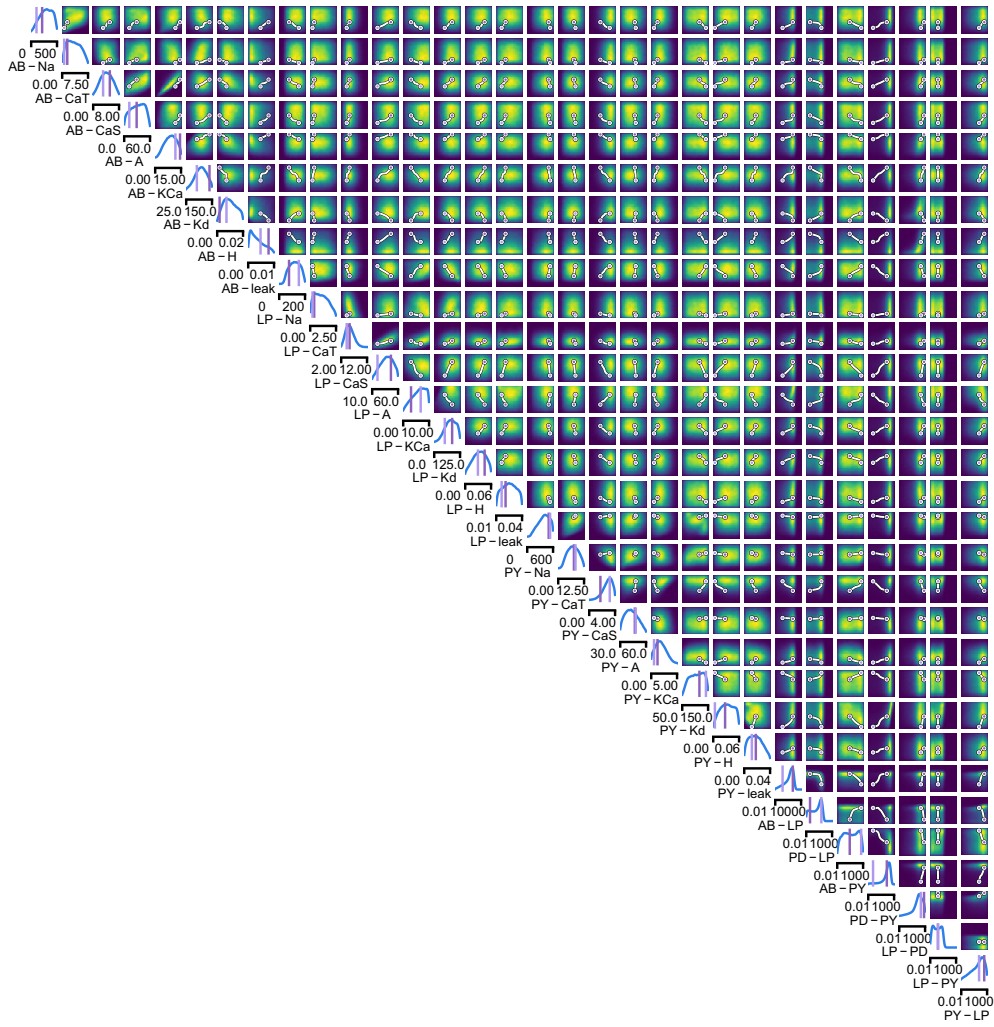

**Appendix 1—figure 11.** Full posterior for the stomatogastric ganglion over 24 membrane and 7 synaptic conductances. The first 24 dimensions depict membrane conductances (top left), the last 7 depict synaptic conductances (bottom right). All synaptic conductances are logarithmically spaced. Between two samples from the posterior with high posterior probability (purple dots), there is a path of high posterior probability (white).

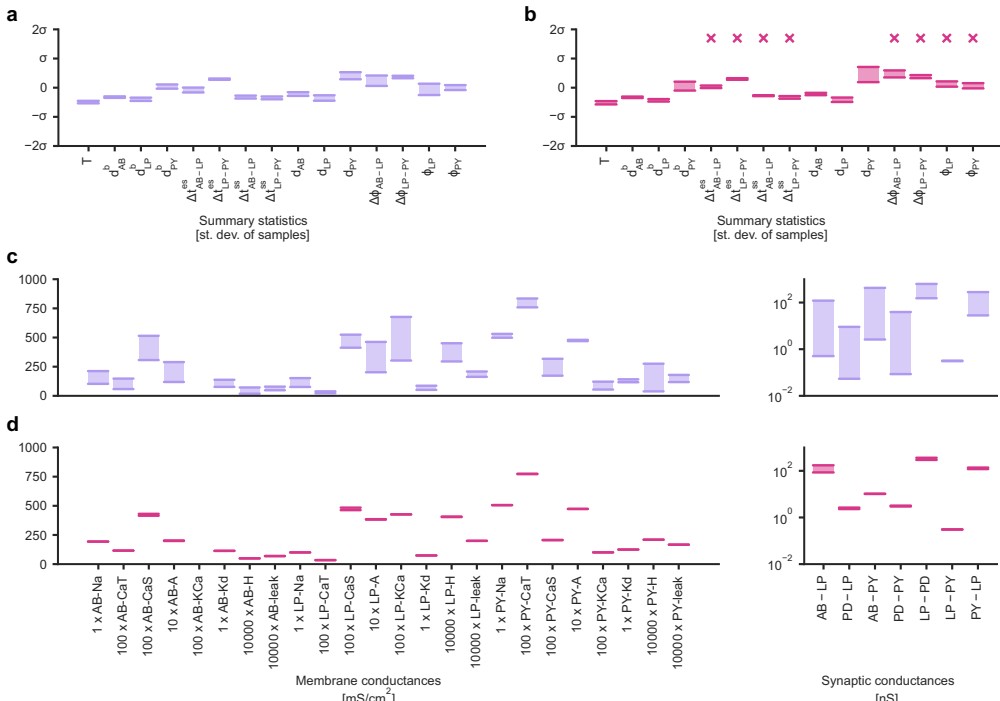

**Appendix 1—figure 12.** Identifying directions of sloppiness and stiffness in the pyloric network of the crustacean stomatogastric ganglion. (**a**) Minimal and maximal values of all summary statistics along the path lying in regions of high posterior probability, sampled at 20 evenly spaced points. Summary statistics change only little. The summary statistics are z-scored with the mean and standard deviation of 170,000 bursting samples in the created dataset. (**b**) Summary statistics sampled at 20 evenly spaced points along the orthogonal path. The summary statistics show stronger changes than in panel a and, in particular, often could not be defined because neurons bursted irregularly, as indicated by an 'x' above barplots. (**c**) Minimal and maximal values of the circuit parameters along the path lying in regions of high posterior probability. Both membrane conductances (left) and synaptic conductances (right) vary over large ranges. Axes as in panel (**d**). (**d**) Circuit parameters along the orthogonal path. The difference between the minimal and maximal value is much smaller than in panel (**c**).

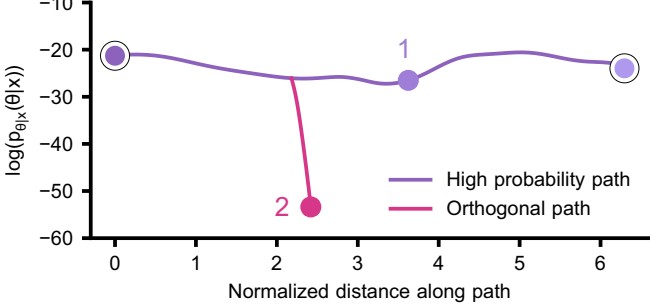

**Appendix 1—figure 13.** Posterior probability along high probability and orthogonal path. Along the path that was optimized to lie in regions of high posterior probability (purple), the posterior probability remains relatively constant. Along the orthogonal path (pink), optimized to quickly reduce posterior probability, the probability quickly drops. The start and end points as well as the points labeled 1 and 2 correspond to the points shown in *Figure 5c*.

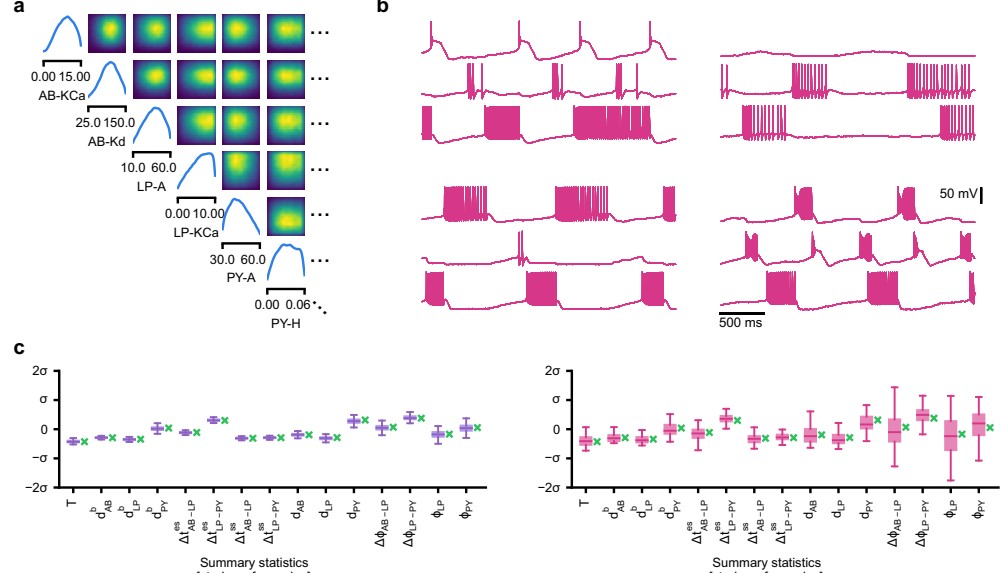

**Appendix 1—figure 14.** Evaluating circuit configurations in which parameters have been sampled independently. (**a**) Factorized posterior, that is, posterior obtained by sampling each parameter independently from the associated marginals. Many of the pairwise marginals look similar to the full posterior shown in *Appendix 1—figure 11*, as the posterior correlations are low. (**b**) Samples from the factorized posterior– only a minority of these samples produce pyloric activity, highlighting the significance of the posterior correlations between parameters. (**c**) Left: summary features for 500 samples from the posterior. Boxplot for samples where all summary features are well-defined (80% of all samples). Right: summary features for 500 samples from the factorized posterior. Only 23% of these samples have well-defined summary features. The summary features from the factorized posterior have higher variation than the posterior ones. Summary features are z-scored using the mean and standard deviation of all samples in our training dataset obtained from prior samples. The boxplots indicate the maximum, 75% quantile, median, 25% quantile, and minimum. The green 'x' indicates the value of the experimental data (the observation, shown in *Figure 5b*).

