## [Decision Letter]

**Acceptance summary:**

The problem of relating dynamical models to data pervades science and is particularly challenging in neuroscience due to high uncertainty in model parameters, model structure and inherent biological variability. This study presents an approach to performing model parameter inference based on large scale simulation and deep neural networks that permits efficient querying of model behaviour and underlying parameter distribution once simulations have been performed. The usefulness is demonstrated on a number of challenging and relevant models.

**Decision letter after peer review:**

Thank you for submitting your article "Training deep neural density estimators to identify mechanistic models of neural dynamics" for consideration by *eLife*. Your article has been reviewed by three peer reviewers, and the evaluation has been overseen by a Reviewing Editor and John Huguenard as the Senior Editor. The following individual involved in review of your submission has agreed to reveal their identity: Mark S. Goldman (Reviewer #3).

The reviewers have discussed the reviews with one another and the Reviewing Editor has drafted this decision to help you prepare a revised submission.

This is a generally well written paper showcasing methodology for inferring model parameter distributions that are compatible with specified behaviours. The methodology uses recent advances in machine learning to approximate a likelihood function over model parameters that can be reused to perform inference. The method is applied to a number of informative examples. Benchmarks provide a favourable comparison against some existing methods (approximate bayesian computation/sampling) and the required computational resources are described.

Some of the reviewers found the methodology hard to follow in places; certain details were lacking. Although the authors pointed out limitations, there were also queries about potential failure modes of this methodology and whether there were methodological heuristics that are non-obvious but important, such as a means of assessing how many samples might be required for a particular model.

The reviewers agreed that a brief revision addressing specific points on methodological clarity (outlined in the reviewer's comments) and commenting on these other points would be worthwhile.

Reviewer #1:

General Assessment: This paper introduces a machine learning technique which generalizes Bayesian approaches to mechanistic models which are complex, multiparameter, and which do not necessarily have probability distributions as their key outputs. This is an interesting and likely productive direction to investigate. They claim their method is scalable and adaptable and the paper applies it to several problems of increasing complexity. However, I am unable to judge their method, because almost no details are given. This is the case even for very general questions such as what they use in place of the likelihood function for Bayes rule, and what they mean by a trained density estimator. The method in this paper may or may not be worthy of publication, but – since it is a method paper – I think the paper would need to be rewritten in a manner that readers can understand both in generality and in detail what their method accomplishes.

Review of Goncalves et al.:

The authors introduce a machine learning technique for finding ensembles of parameter values of mechanistic models which are consistent with experimental data.

The core of their method is the “Sequential Neural Posterior Estimation” (SNPE) which takes in a mechanistic model, a (thing which resembles a) prior and summary data, does some mysterious machine-learning processing and spits out something like a posterior probability distribution.

I agree with the authors that large unwieldy mechanistic models likely require machine learning to find parameter ensembles. I also agree that it should be possible to develop Bayesian like approaches for models which are not naturally probabilistic, a problem they pose nicely, and which I have come across but not quite realized was of such importance. I also find their results on the gastric circuit interesting- particularly the finding that the many disparate parameter values that can all fit data seem to lie on a single plateau, rather than on distinct minima

However, I don't really know where to begin in evaluating SNPE. It feels as if there is a section missing- the Introduction talks in generalities about large mechanistic models and the need to address their difficulties, and then it jumps to discussing the particular applications of SNPE.

I don't know what the authors mean by posterior and prior, nor what they mean when they say their method works without a likelihood.

There is muddling of what their output is and what they do to achieve it, and I don't find either sufficiently explained. For example, machine learning could be used to approximate a posterior in the usual Bayesian sense. Still I would want to know what prior they are using what their likelihood function is and what posterior they are approximating, even if there is a black box in the middle. I don't understand what their p(θ|x) is an approximation to.

Reviewer #2:

General assessment

This is a really clear and well written paper with practical solutions for an important problem in theoretical neuroscience. In addition, the authors provide high quality code and documentation making it fairly straightforward for other groups to make use of these tools. I would expect that these tools would become widely adopted in the community. Indeed, I would love to have had this for many previous modelling studies and will certainly use it in the future.

Major comments

1) My main issue is that applying this method depends on how well the deep network does its job of estimating the density (which depends partly on the user specifying the correct architecture and parameters). The authors do discuss this under limitations, but it might be interesting to see what a failure would look like here and what are the scientific consequences of such a failure. I suspect the answer might partly be that you should always verify the resulting posterior distributions, but how practical is this in a 31-dimensional setting, for example? When the authors were working on the examples for this manuscript, how much work did it take to make sure that there were no failures (assuming they verified this) and how did they detect them? (For example, in subsection “Functional diversity of ion channels: efficient high-throughput inference” it's implied that the only failures in this example were because the Omnimodel wasn't able to fit some data.)

2) Would it be possible to use SNPE to automatically suggest the optimal experiment that would give the narrowest posterior distribution over parameters? This might already be what was suggested when citing papers that "automatically construct informative summary features", I wasn't entirely sure.

Reviewer #3:

This paper addresses a major issue in fitting neuroscience models: how to identify the often degenerate, or nearly degenerate, set of parameters that can underlie a set of experimental observations. Whereas previous techniques often depended upon brute force explorations or special parametric forms or local linearizations to generate sets of parameters consistent with measured properties, the authors take advantage of advances in deep networks for use in estimating probability densities to address this problem. Overall, I think this paper and the complementary submission have the potential to be transformative contributions to model fitting efforts in neuroscience, and the ability of this work to handle cases that are not continuous is a plus. However, I think the paper needs to clarify/quantify how much of an advance this is over brute force methods (plus simple post-hoc analyses on the simulated database), given the heavy brute force aspect of generating the training set, and also whether there are biases in the method. Also, the writing could benefit from more focus on methodology (rather than on examples) to ease adoption of the method.

Substantive concerns:

1) In the example of Figure 1, the Appendix 1—figure 1A (which should appear in the main figure rather than the supplement) appears to show that the SNPE's mean temporal filter is considerably off from the true temporal value. Why is this the case? More generally, this suggests that more “ground truth” examples with direct side-by-side comparisons should be provided to assure that the estimates provided by this method are not significantly biased. For example, Figure 5 is wonderful in showing a sloppy/insensitive direction in parameter space that has sharp curvature. Could one do something similar to this and directly compare (for an example with smaller number of parameters, e.g. 4 or 5) the estimated probability distribution to a “ground truth” generated through a sufficiently dense grid search, for different numbers of training samples.

2) The authors need to clarify, quantitatively, how much of a benefit this is over simple post-hoc analysis of results of brute force methods that just randomly (uniformly) sample parameters, especially given that there is a lot of brute force work in this method as well in generating the samples for the deep network training. For such brute force methods (e.g. rather than a grid, some studies just uniformly randomly generate parameters to use in the simulations), one can also just post-hoc fit the results to a mixture of Gaussians or similar classical density estimator so that other values can be interpolated from the obtained data. How much is the deep network approach needed? My guess is that it does help, potentially a huge amount, but the authors need to show this in a carefully controlled comparison.

3) Related to #1, given that this paper is primarily a methodological advance, I think the authors could spend more time laying out the essence of the basic algorithm in the beginning of the Results. For example, it's not clear in Figure 1 exactly what is being input and output from the neural density estimator deep network. More generally, it's worth describing more about the algorithm, perhaps in less technical language, in the main results.

Within the Materials and methods, since the authors indicated some playing around with parameters was needed, it would help a lot for the adoption of this method by others to have more explanation of different features of the method and more intuition behind choices. For example, what is the loss function used after the initial round? Are there any basic constraints on what shape the deep network should have (e.g. should it fan out in the middle layers, fan in, or be approximately uniform in width)? What is the basic idea behind a masked autoregressive flow (MAF) and when should that be chosen versus the mixture of Gaussians model? The algorithm's description could also be expanded (or a short version could be in the main and expanded version in the Materials and methods).

4) Is there some way to ballpark estimate, or post-hoc characterize, how many samples may be needed? Seemingly, the fact that this works with less than one sample per dimension says something about the geometry of the solution space and its effective dimensionality. (If this seems too hard, the authors can punt on it).

[Editors' note: further revisions were suggested prior to acceptance, as described below.]

Thank you for resubmitting your article "Training deep neural density estimators to identify mechanistic models of neural dynamics" for consideration by *eLife*. Your revised article has been reviewed by two peer reviewers, and the evaluation has been overseen by a Reviewing Editor and John Huguenard as the Senior Editor. The following individual involved in review of your submission has agreed to reveal their identity: Mark S Goldman (Reviewer #3).

The reviewers have discussed the reviews with one another and the Reviewing Editor has drafted this decision to help you prepare a revised submission.

Summary

This is methodology is of potentially wide interest, and although the authors have made efforts to explain the interpretation, applicability and caveats of the methodology, one reviewer feels that the manuscript could benefit from further clarification on:

1) How one should interpret “posterior probability density” in cases where there is a deterministic relationship between parameters and behaviour, e.g. in extreme cases where there might be a single solution.

2) The fairness of comparisons with ABC/sampling – are these alternatives being compared in an even-handed way?

3) It may be helpful to have a very explicit example of the results SNPE returns on a problem where “ground truth” of feasible parameters are known. This could be a simple model dynamical system, used for illustration.

Full reviewer comments are included below, for reference.

Reviewer #2:

I'm very happy with the reviewers response to my previous review, and congratulate them on a great paper!

Reviewer #3:

The authors have clarified the methods in a way that I think should be helpful, and I remain optimistic that this will be a very valuable technique for the field. However, I think some of the other concerns raised about rigorously demonstrating the technique, and clarifying the method conceptually were not addressed. I will try to re-state these concerns in a broader scope that I hope strengthens the paper, clarifies potential issues, and gets at some key conceptual and rigor issues that need to be addressed

1) Conceptual question: The manuscript and reply heavily emphasize that SNPE performs Bayesian inference, as compared to approaches that only are applicable to deterministic modeling (like classic brute force approaches). However, outside of the very simple illustrative example of Figure 2, it appears that all of the remaining examples are done on deterministic models. This leads to a fundamental conceptual question: how should one interpret the resulting posteriors resulting from SNPE?

To be concrete, if one used the precise output of a simulation (rather than summary statistics), then the actual posterior probability over parameter values is a δ function (assuming the model if identifiable). If there are nearby solutions, this wouldn't change the true posterior probability, i.e. despite being nearby, they would have posterior probability of zero. Thus, while the authors criticize other methods for using “user defined” criteria of being close to a solution, this example illustrates that if one found the exact posterior probability (as the authors state in their reply that the method should ideally do if it works correctly), all information about the existence of nearby (i.e. almost, but not entirely, degenerate) solutions would be lost. I'm guessing the density estimator will somehow tradeoff “posterior probability large” for “close to the correct solution”, but this then requires some interpretation of how “closeness to the correct simulation output” is translated to “posterior probability”. Put in this light, at least a user defined criterion of closeness for a deterministic model, as done in traditional fitting methods, is explicit and easy to interpret. Can the authors justify or provide users with an explanation for how to interpret SNPE when applied to deterministic simulations-my guess is that, when SNPE is working well, it's some sort of smoothed version of the true, very spiky posterior, but is there a way for users to interpret the solution, e.g. how would a user know if there are nearby solutions if the SNPE did a great job of finding the true posterior and therefore didn't indicate these nearby solutions? And how should the user interpret “probabilities” in a deterministic model where there may only be 1 true solution?

Note: when summary statistics are used rather than exact simulation output, this will likely create some degeneracies. However, this still may end up creating a razor thin slice in parameter space that corresponds to truly, exactly matching the summary statistics, which would then be represented by the true posterior. This scenario then returns to the same fundamental questions of: a) how to interpret the posterior from SNPE, and b) how to interpret a simulation output that is very close to the summary statistics but not equal to them and therefore perhaps lost if SNPE is finding the true posterior of the deterministic model.

2) On the one example that is stochastic (Figure 2), I am still confused by the results shown. Specifically, in Figure 2C, the posterior samples consistently and systematically undershoot the true receptive field for intermediate parameter values. This is a simple LN model for which classic maximum likelihood models usually doing an excellent job finding the true receptive field, at least with enough data. If one ran standard maximum likelihood estimation on this, would it miss? If so, why? Is this reflecting that the authors are using too broad a prior and it is not converging to the true posterior or that too few samples are used (the number of samples is not indicated)-if the latter, can the authors run with more samples and show that their method then converges to the correct answer? Or is it something else (e.g. see next paragraph)?

In addition, the authors state that their method works because it gets the same answer as MCMC. However, I'm wondering if both might be making errors due to the use of the Polya-Γ sampling scheme since, at least in the Pillow and Scott article cited, this was used not for a standard LN model but rather to model more complex models in which the variance was greater than for a Poisson spike count. In any case, it seems worth explaining where the systematic error in the posterior samples is coming from, given that the LN model is such a simple classic model fit whose parameters have been successfully identified by simpler maximum likelihood methods.

3) Related to the Figure 2 example, I was struck by the huge errors and nearly zero correlation for the ABC methods. Is this fundamental to ABC or is this because the authors are hamstringing the ABC method by applying it in a very naïve manner that isn't how someone using ABC would apply it. Specifically, from the methods, it looks like the authors are considering every parameter to be independent, with no smoothing or regularization. I would think that someone using ABC methods would apply regularizations or the like that might dramatically reduce the effective dimensionality of the problem, so it's not clear if the comparison shown in the manuscript is fair. This isn't to say that SNPE won't still work much better than ABC methods, but it's worth checking to make sure the ABC methods are being applied in a realistic manner if the comparison is to be made.

4) Ground truthing. The authors suggest that they can't ground truth the technique due, in part, to the stochastic nature of it. While I appreciate there may be some challenges, a lot could be done, especially since most of the manuscript focuses on deterministic examples that are easier to check. For example, in Figures 3-5, parameter estimation results that may or may not be correct are shown and the reader is left to assume they are correct by looking at a single or very small number of examples. While I fully expect that it does work reliably, it would be nice to actually run a large set of simulations for posterior checks (even if, e.g., only the mode of the distribution was checked) and then report some statistics.

5) Related to ground truthing, in Figure 5D and the highlighted 2-D plot of Figure 5C, I am confused as to whether the example shows the success or partial failure of the method. Specifically, the point labeled "2" in the Figures 5C and 5D appears to be in a bright (and therefore presumably high probability?) region of the parameter space, both in an absolute sense and possibly (it's hard to tell) compared to the light purple circle simulation (and also relative to going further down below this light purple trace). However, the authors use point 2 to make the point that the simulation looks very far off from the reference trace. This suggests, possibly, that similar probabilities being output by SNPE is on the one hand (values close to the light purple circle) leading to a very good fit and on the other hand (red point 2) leading to a very bad fit – thus, this suggests that one perhaps can't trust “bright” regions in the SNPE posterior to correspond to “good” fits. Can the authors check this and address the more general concern. I'm guessing this may relate to my larger conceptual question of interpreting the “probabilities” for SNPE in my reviewer comment #1 above.

6) Difference from “brute force methods”. I thank the authors for trying to generate a comparison to brute force methods by showing a rejection method. However, I don't think this is necessary and I think it's confusing as I don't think “brute force” and “rejection” are the same (i.e. I'd remove this from the paper as being more confusing than helpful). To me at least, “brute force” methods simply generate many example simulations into a database. This is separable from the problem of “what does one then do with this database afterwards”. My point was simply that I think SNPE can be interpreted as being either within the class of brute force searches in generating a very large database of simulations (e.g. ~20 million for Figure 5), or if one focuses on the density estimation component, as being a method that is complementary to the brute force database generation step by addressing the “what does one then do with this database afterwards” component of the problem. For the “what does one do with this database” question, one can certainly imagine many things-one can simply visualize the parameter space through various slices, effectively marginalizing over certain variables and illustrating for example where bifurcations occur that might be difficult to identify mathematically, or can (as the authors highlight has often been done previously…but this should not be equated with parameter search methods, as it's just one possible post-search thing that can be done) implement some sort of selection/rejection criterion. One might also want to interpolate/extrapolate, asking for an estimate of what a new set of parameters would lead to, for which a density estimator such as SNPE would excel.

This is less a criticism of the approach and more of a clarification for placing the method within existing work. I agree with the authors that current use of brute force generation of databases has not taken advantage of better methods of post-database-generation analysis, as SNPE offers, leading to the likely (and potentially dramatically, as demonstrated in Figure 5) incorrect implicit assumption that one needs to generate many samples per dimension to well-characterize the space. I just wanted to separate the notion of generating a database of many examples (which doesn't need to be done on a grid), from the notion of how one post-hoc analyzes it. And overall, this focus on the power of post-database-generation analysis is why I still think SNPE, and the more general focus on use of density estimators, is a potentially transformative contribution for the practice of model building (in addition to SNPE's applicability to stochastic problems that is already well emphasized in the manuscript).

---

## [Author Response]

Reviewer #1:[…] I agree with the authors that large unwieldy mechanistic models likely require machine learning to find parameter ensembles. I also agree that it should be possible to develop Bayesian like approaches for models which are not naturally probabilistic, a problem they pose nicely, and which I have come across but not quite realized was of such importance. I also find their results on the gastric circuit interesting- particularly the finding that the many disparate parameter values that can all fit data seem to lie on a single plateau, rather than on distinct minimaHowever, I don't really know where to begin in evaluating SNPE. It feels as if there is a section missing- the Introduction talks in generalities about large mechanistic models and the need to address their difficulties, and then it jumps to discussing the particular applications of SNPE.I don't know what the authors mean by posterior and prior, nor what they mean when they say their method works without a likelihood.There is muddling of what their output is and what they do to achieve it, and I don't find either sufficiently explained. For example, machine learning could be used to approximate a posterior in the usual Bayesian sense. Still I would want to know what prior they are using what their likelihood function is and what posterior they are approximating, even if there is a black box in the middle. I don't understand what their p(θ|x) is an approximation to.

We thank reviewer 1 for the detailed feedback, which has led us to clarify the descriptions of our techniques in both the Results and Materials and methods sections, putting emphasis on (1) the methodological foundation and inner workings of the algorithm and (2) how prior distributions on model parameters are chosen.

To clarify: SNPE performs Bayesian inference “in the usual sense” of calculating the posterior distribution, that is, the probability distribution of model parameters given observed data. However, in contrast to other approaches based on explicit application of Bayes’ rule, SNPE calculates the posterior distribution without ever having to numerically evaluate or approximate the likelihood of parameters given data. Instead, it trains a neural network using an objective function and learning procedure that have been proved capable, in both theory and practice, of recovering the posterior from model simulations (see first section of Results, and Materials and methods).

Thus, our definitions and usage of the concepts of “prior” and “posterior” are the standard versions used in statistical data analysis, the only (but important!) difference is that we can approximate the posterior without having to evaluate the likelihood function.

In order to emphasize and clarify these points, we now start the Results section with a description of SNPE:

“SNPE performs Bayesian inference on mechanistic models using only model-simulations, without requiring likelihood evaluations. […] SNPE's efficiency can be further improved by using the running estimate of the posterior distribution to guide further simulations towards data-compatible regions of the parameter space (Papamakarios and Murray 2016, Lueckmann et al., 2017, Greenberg et al., 2019).”

In the Materials and methods, we now have a section (“Bayesian inference without likelihood-evaluations with SNPE”) describing in more detail the loss function and the diverse versions of SNPE algorithms.

Reviewer #2:General assessmentThis is a really clear and well written paper with practical solutions for an important problem in theoretical neuroscience. In addition, the authors provide high quality code and documentation making it fairly straightforward for other groups to make use of these tools. I would expect that these tools would become widely adopted in the community. Indeed, I would love to have had this for many previous modelling studies and will certainly use it in the future.

We thank reviewer 2 for evaluating our paper and are pleased to know that the reviewer considers the paper to be “well-written” and the tools to have the potential to be “widely adopted in the community.”

Major comments1) My main issue is that applying this method depends on how well the deep network does its job of estimating the density (which depends partly on the user specifying the correct architecture and parameters). The authors do discuss this under limitations, but it might be interesting to see what a failure would look like here and what are the scientific consequences of such a failure. I suspect the answer might partly be that you should always verify the resulting posterior distributions, but how practical is this in a 31-dimensional setting, for example? When the authors were working on the examples for this manuscript, how much work did it take to make sure that there were no failures (assuming they verified this) and how did they detect them? (For example, in subsection “Functional diversity of ion channels: efficient high-throughput inference” it's implied that the only failures in this example were because the Omnimodel wasn't able to fit some data.)

The reviewer is correct to point out that this methodology has limitations, and this becomes more evident in high-dimensional parameter spaces. In the previous version of the manuscript, we included a Discussion section on limitations, and mentioned the use of posterior predictive checks to diagnose inference-failures:

“One common ingredient of these approaches is to sample from the inferred model, and search for systematic differences between observed and simulated data, e.g. to perform posterior predictive checks (Cook et al., 2006,Talts et al., 2018, Liepe et al., 2014, Lueckmann et al., 2017, Greenberg et al., 2019) (Figure 2G, Figure 3F,G, Figure 4C, and Figure 5D).”

In this approach, one samples from the inferred posterior given the observed data, simulates for these sampled parameters, and compares the simulations with the observed data. While “posterior predictive checks” are scalable to higher-dimensional parameter spaces, since one can sample efficiently with SNPE, this approach only allows one to detect whether a mode of the posterior has been wrongly identified, but fails at detecting when there is an “undiscovered” island of the posterior. Such issues are currently unsolved in the field of simulation-based inference and in Bayesian inference in general. For all applications, and especially in high-dimensional problems, it is good practice to resort to multiple initialisations of these algorithms in order to minimise the failure to detect parameter islands.

We have expanded the Discussion about limitations:

“These approaches allow one to detect “failures” of SNPE, i.e. cases in which samples from the posterior do not reproduce the data. However, when diagnosing any Bayesian inference approach, it is challenging to rigorously rule out the possibility that additional parameter-settings (e.g. in an isolated “island”) would also explain the data. Thus, it is good practice to use multiple initialisations of SNPE, and/or a large number of simulations in the initial round.”

We should emphasise that in the manuscript, we analyse a few test cases (Figure 2) where the posterior is known or calculable, cases that have similar dimensionality as other applications where the posterior is not known (the ion-channel model and the Hodgkin-Huxley model). This gives us confidence in the quality of the inferred posteriors.

2) Would it be possible to use SNPE to automatically suggest the optimal experiment that would give the narrowest posterior distribution over parameters? This might already be what was suggested when citing papers that "automatically construct informative summary features", I wasn't entirely sure.

We thank the reviewer for the excellent point. SNPE could indeed be extended for Bayesian experimental design (see e.g. Kleinegesse and Gutmann, 2019 for a related approach), and we regard this as an exciting avenue for future work. However, we do think that this would constitute a major new research project that would be substantially beyond the scope of this study. We now cite this study when describing the possibilities brought by amortised inference:

“These results show how SNPE allows fast and accurate identification of biophysical model parameters on new data, and how SNPE can be deployed for applications requiring rapid automated inference, such as high-throughput screening-assays, closed-loop paradigms (e.g. for adaptive experimental manipulations or stimulus-selection (Kleinegesse and Gutmann, 2019)), or interactive software tools.”

Reviewer #3:This paper addresses a major issue in fitting neuroscience models: how to identify the often degenerate, or nearly degenerate, set of parameters that can underlie a set of experimental observations. Whereas previous techniques often depended upon brute force explorations or special parametric forms or local linearizations to generate sets of parameters consistent with measured properties, the authors take advantage of advances in deep networks for use in estimating probability densities to address this problem. Overall, I think this paper and the complementary submission have the potential to be transformative contributions to model fitting efforts in neuroscience, and the ability of this work to handle cases that are not continuous is a plus. However, I think the paper needs to clarify/quantify how much of an advance this is over brute force methods (plus simple post-hoc analyses on the simulated database), given the heavy brute force aspect of generating the training set, and also whether there are biases in the method. Also, the writing could benefit from more focus on methodology (rather than on examples) to ease adoption of the method.

We thank reviewer 3 for evaluating our paper and are pleased to know that the reviewer considers the methodology presented to have “the potential to be transformative”. We address the reviewer's specific comments below.

Substantive concerns:1) In the example of Figure 1, the Appendix 1—figure 1A (which should appear in the main figure rather than the Appendix) appears to show that the SNPE's mean temporal filter is considerably off from the true temporal value. Why is this the case? More generally, this suggests that more “ground truth” examples with direct side-by-side comparisons should be provided to assure that the estimates provided by this method are not significantly biased. For example, Figure 5 is wonderful in showing a sloppy/insensitive direction in parameter space that has sharp curvature. Could one do something similar to this and directly compare (for an example with smaller number of parameters, e.g. 4 or 5) the estimated probability distribution to a “ground truth” generated through a sufficiently dense grid search, for different numbers of training samples.

The goal of SNPE is to perform Bayesian inference. Thus, if SNPE works correctly, the posterior distribution inferred by SNPE should be the same as the “true” posterior. In particular, the true SNPE-mean filter should be the same as the “true-posterior” mean filter (here computed with MCMC), which is the case in Appendix 1—figure 1A. So, Appendix 1—figure 1A shows that SNPE is working as intended.

For finite and noisy data, the posterior mean is not necessarily the same as the true parameters. In this setting, it is fundamentally impossible for any estimation procedure to recover the true parameters, and Bayesian inference therefore uses a compromise between the data and the prior. This has advantageous features on average, but for a single example, the posterior mean might deviate from the true parameters. These issues are well understood and widely appreciated in statistics, but we realize that we have not sufficiently made this clear in this figure.

(As a side note, this issue implies that verifying the correctness of SNPE is not as simple as “comparing true and recovered parameters” is the reason for why we have included extensive analyses on problems with known ground-truth posteriors and additional posterior predictive checks, see e.g. Figure 2C,D,F,G,H and Appendix 1—figures 1,2,4).

In order to clarify that perfect inference does not necessarily mean recovering the true underlying parameters, we now write in the respective Results sub-section:

“If SNPE works correctly, its posterior mean filter will match that of the reference posterior, however, it is not to be expected that either of them precisely matches the ground-truth filter (Figure 2c and Appendix 1—figure 1): In the presence of finite sampling and stochasticity, multiple different filters could have plausibly given rise to the observed data. A properly inferred posterior will reflect this uncertainty, and include the true filters as one of many plausible explanations of the data (but not necessarily as the “mean” of all plausible explanations) (Appendix 1—figure 2).”

Regarding moving Appendix 1—figure 1A to Figure 2, we think that the posterior samples plotted in Figure 2C alongside the true temporal filter, already allow one to appreciate the respective similarities and differences.

Regarding the comparison of our approach on a (simpler) STG model to a dense grid-search, we would like to point out that the grid search would require the probabilistic mapping of the parameters to the summary statistics, i.e. the likelihood function, which is intractable in the STG case and is therefore not possible to do at this point. Thus, in any problems with intractable likelihoods, it is intractable to compute the “true posterior” to “prove” correctness, but we can do a range of posterior predictive checks (as shown in the manuscript), e.g. showing that for high-probability under the STG posterior, the simulated traces are similar to the observed data (Figure 5C,D, and Appendix 1—figure 12), whereas orthogonal paths lead very quickly to simulated traces different from the observed data. This suggests that the inferred STG posterior is capturing correctly at least one region of high probability.

2) The authors need to clarify, quantitatively, how much of a benefit this is over simple post-hoc analysis of results of brute force methods that just randomly (uniformly) sample parameters, especially given that there is a lot of brute force work in this method as well in generating the samples for the deep network training. For such brute force methods (e.g. rather than a grid, some studies just uniformly randomly generate parameters to use in the simulations), one can also just post-hoc fit the results to a mixture of Gaussians or similar classical density estimator so that other values can be interpolated from the obtained data. How much is the deep network approach needed? My guess is that it does help, potentially a huge amount, but the authors need to show this in a carefully controlled comparison.

We thank the reviewer for the excellent comment. We would like to emphasise that SNPE performs Bayesian inference on stochastic models, whereas “brute-force” approaches are generally only applied to deterministic models, and based on heuristically chosen distance measures. If applied to stochastic models, such brute-force approaches can be interpreted as rejection-ABC, which rejects simulations according to the chosen distance measure, and which we show in Figure 2 to perform poorly compared to SNPE. This suggests that the big advantage of SNPE arises from the fact that (1) it uses all simulations independently of the similarity to the empirical data, and (2) it learns to identify the features most informative about the parameters.

We now have an extensive Discussion on these aspects:

“How big is the advance brought by SNPE relative to “conventional” brute-force approaches that aim to exhaustively explore parameter space? […] Finally, we showed above that, on the STG circuit, SNPE also yielded solutions more similar to experimental data than a brute-force approach (based on fitting a mixture-model to accepted simulations).”

Furthermore, we provide a comparison between SNPE and an approach where we combined rejection ABC with mixture-of-Gaussians density estimation. To do so, we used the dataset of simulations obtained for the crustacean stomatogastric ganglion by sampling from the prior and picked the 10000 parameter sets that produced the closest summary features to the observed data. As a distance metric, we used the Euclidean distance over summary features after having standardized them. We then fitted a mixture of Gaussians with 20 components to these 10000 samples using the Python machine learning library scikit-learn. We found that samples from the SNPE posterior have summary statistics that are substantially closer to the observed data than samples from the mixture-of-Gaussians approach. We report these new results as Appendix 1—figure 14 and a respective paragraph in the Results:

“How much does the conditional density estimation network contribute to posterior estimation? […] In the next section, we will furthermore show how such a powerful posterior representation can be used to explore the properties of neural dynamics.”

3) Related to #1, given that this paper is primarily a methodological advance, I think the authors could spend more time laying out the essence of the basic algorithm in the beginning of the Results. For example, it's not clear in Figure 1 exactly what is being input and output from the neural density estimator deep network. More generally, it's worth describing more about the algorithm, perhaps in less technical language, in the main results.

As replied to reviewer 1’s comments, we have clarified the methodology in Results. We now start the Results section with a description of SNPE:

“SNPE performs Bayesian inference on mechanistic models using only model-simulations, without requiring likelihood evaluations. […] SNPE's efficiency can be further improved by using the running estimate of the posterior distribution to guide further simulations towards data-compatible regions of the parameter space (Papamakarios and Murray, 2016, Lueckmann et al., 2017, Greenberg et al., 2019).”

Within the Materials and methods, since the authors indicated some playing around with parameters was needed, it would help a lot for the adoption of this method by others to have more explanation of different features of the method and more intuition behind choices. For example, what is the loss function used after the initial round? Are there any basic constraints on what shape the deep network should have (e.g. should it fan out in the middle layers, fan in, or be approximately uniform in width)? What is the basic idea behind a masked autoregressive flow (MAF) and when should that be chosen versus the mixture of Gaussians model? The algorithm's description could also be expanded (or a short version could be in the main and expanded version in the Materials and methods).

We have extended the Materials and methods section to address reviewer’s questions:

1) We now provide more details on the loss functions for SNPE-A/B/C, and point to previous literature showing how minimising the loss function can be used to perform Bayesian inference without likelihood evaluations;

2) We include a new section about “Neural density estimation”, where we discuss the constraints on the shape of the mixture density networks:

“The MDN outputs the parameters of a mixture of Gaussians (i.e. mixture weights, and for each component of the mixture, the mean vector and covariance entries). Thus, for an MDN composed of K components, we chose an architecture with at least as many units per layer as K( 1 + dim(θ) + dim(θ)(dim(θ)+1)/2 ) – 1, to ensure enough flexibility to approximate well the parameters of the mixture of Gaussians. For example, when inferring the parameters of the Hodgkin-Huxley model given in vitro recordings from mouse cortex (Allen Cell Types Database, https://celltypes.brain-map.org/data), we infer the posterior over 8 parameters with a mixture of two Gaussians, and the MDN needs at least 89 units per layer. Across applications, we found 2 layers to be sufficient to appropriately approximate the posterior distribution.”

3) We also describe the basic idea of MAFs and the respective constraints:

“MAF is a specific type of normalizing flow, which is a highly flexible density estimator (Rezende et al., 2015; Papamakarios et al., 2017, Papamakarios et al., 2019). Normalizing flows consist of a stack of bijections which transform a simple distribution (usually a multivariate Gaussian distribution) into the target distribution. Each bijection is parameterized by a specific type of neural network (for MAF: a Masked Autoencoder for Distribution Estimation, or MADE). In our experiments, 5 stacked bijections are enough to approximate even complex posterior distributions. Depending on the size of the parameter and data space, each neural network had between (50,50) and (200,400) hidden units.”

4) Regarding the choice between MDNs and MAFs, we write:

“When using SNPE in a single-round, we generally found superior performance for MAFs as compared to MDNs. When running inference across multiple rounds, training MAFs leads to additional challenges which might impede the quality of inference (Greenberg et al., 2019; Durkan et al., 2020)”

4) Is there some way to ballpark estimate, or post-hoc characterize, how many samples may be needed? Seemingly, the fact that this works with less than one sample per dimension says something about the geometry of the solution space and its effective dimensionality. (If this seems too hard, the authors can punt on it).

The reviewer is raising an important point. SNPE is using neural density estimators to approximate the function between summary statistics and a distribution over parameters. Thus, in general, the higher the dimensionality and complexity of this function, the more flexible the neural density estimator should be, and therefore the higher the required number of simulations to appropriately fit such a function. Given the unavailability of such a function a priori, it is in general hard to provide a definite answer. In our workflow, we have a few heuristics, which we now clarify in the manuscript, in the Materials and methods, section “Simulation-based inference”:

“The required number of simulations depends on both the dimensionality and complexity of the function between summary statistics and model parameters. While the number of parameters and summary-features can easily be determined, it can be hard to determine how “complex” (or nonlinear) this mapping is, this makes it difficult to give general guidelines on how many simulations will be required. A practical approach is to choose a simulation-budget based on the computational cost of the simulation, inspect the results (e.g. with posterior predictive checks), and add more simulations when it seems necessary.”

[Editors' note: further revisions were suggested prior to acceptance, as described below.]

SummaryThis is methodology is of potentially wide interest, and although the authors have made efforts to explain the interpretation, applicability and caveats of the methodology, one reviewer feels that the manuscript could benefit from further clarification on:

We thank the editors and reviewers for their constructive comments. We are glad that reviewer 2 was “very happy” with our response and hope to address any remaining concerns by reviewer 3 both in this response and by making the corresponding changes in the manuscript.

Some main concerns seem to be based on the (incorrect) assumption that we deal with deterministic models—this is not the case, all of our models are stochastic. They are only “likelihood-free” in the sense that the likelihood cannot be evaluated (but mathematically it exists). This is addressed in detail in our response below. In addition, we made sure this is now even more clear from the manuscript. We hope to clear up the confusion about the example with ground truth, showing that SNPE indeed agrees with reference solutions and posterior uncertainty shrinks in light of more data, and we clarify fairness of comparisons with ABC/sampling.

1) How one should interpret “posterior probability density” in cases where there is a deterministic relationship between parameters and behaviour, e.g. in extreme cases where there might be a single solution.

SNPE is designed to perform Bayesian inference, i.e. to calculate the posterior distribution on stochastic models. While, in principle, one can also apply SNPE to deterministic models, we believe that a full investigation of how it behaves under these circumstances, and how to interpret its outputs, is outside of the scope of this paper. Furthermore, we would claim that having some model of variability is paramount for any data analysis, as there can never be a perfect match between model and data.

2) The fairness of comparisons with ABC/sampling – are these alternatives being compared in an even-handed way?

We believe they are—see detailed explanation below. We also remark that the superiority of SNPE-algorithms over rejection ABC, albeit in problems outside of neuroscience, has been demonstrated both by us (Lueckmann, Goncalves et al., 2017, Greenberg et al., 2019) and others (Papamakarios and Murray, 2016).

3) It may be helpful to have a very explicit example of the results SNPE returns on a problem where “ground truth” of feasible parameters are known. This could be a simple model dynamical system, used for illustration.

The manuscript contains two examples (a LN GLM and a Gabor GLM) where the likelihood is known, and as we explain below (see response to reviewer), we can obtain the respective ground-truth posteriors using standard inference algorithms (MCMC). We show that SNPE recovers the correct posterior in both cases. In our detailed response below, we also include additional analyses on the LN GLM model, and clear out concerns about the correctness of the returned posteriors (e.g., the STG).

We therefore respectfully disagree that the addition of yet another example would strengthen the manuscript. In case the editor still finds it necessary after our detailed response, we would like to clearly understand what precisely the additional value provided by adding one more example application would be.

Reviewer #3:The authors have clarified the methods in a way that I think should be helpful, and I remain optimistic that this will be a very valuable technique for the field. However, I think some of the other concerns raised about rigorously demonstrating the technique, and clarifying the method conceptually were not addressed. I will try to re-state these concerns in a broader scope that I hope strengthens the paper, clarifies potential issues, and gets at some key conceptual and rigor issues that need to be addressed1) Conceptual question: The manuscript and reply heavily emphasize that SNPE performs Bayesian inference, as compared to approaches that only are applicable to deterministic modeling (like classic brute force approaches). However, outside of the very simple illustrative example of Figure 2, it appears that all of the remaining examples are done on deterministic models. This leads to a fundamental conceptual question: how should one interpret the resulting posteriors resulting from SNPE?To be concrete, if one used the precise output of a simulation (rather than summary statistics), then the actual posterior probability over parameter values is a δ function (assuming the model if identifiable). […] And how should the user interpret “probabilities” in a deterministic model where there may only be 1 true solution?Note: when summary statistics are used rather than exact simulation output, this will likely create some degeneracies. However, this still may end up creating a razor thin slice in parameter space that corresponds to truly, exactly matching the summary statistics, which would then be represented by the true posterior. This scenario then returns to the same fundamental questions of: a) how to interpret the posterior from SNPE, and b) how to interpret a simulation output that is very close to the summary statistics but not equal to them and therefore perhaps lost if SNPE is finding the true posterior of the deterministic model.

All the applications in the manuscript are stochastic, as described in the original version of the manuscript: we write both in Results and Materials and methods that the Hodgkin-Huxley model includes "intrinsic neural noise"; regarding the STG circuit, we write in Materials and methods that "we added Gaussian noise with a standard deviation of 0.001 mV to the input of each neuron". We note that the information of the stochastic nature of the ion channel model was omitted by mistake in the Materials and methods, and this is now corrected.

We do, however, realize that this point was not sufficiently clear, so we emphasized it further:

1) In the ion channel application, we now explicitly write in the Results where the stochasticity comes in:

“We aimed to identify these ion channel parameters θ for each ICG model, based on 11 features of the model's response to a sequence of 5 noisy voltage clamp protocols, resulting in a total of 55 different characteristic features per model (Figure 3B, see Materials and methods for details).”

And in the Materials and methods:

“We modeled responses of the Omnimodel to a set of five noisy voltage-clamp protocols (Podlaski et al., 2017): as described in Podlaski et al., the original voltage-clamp protocols correspond to standard protocols of activation, inactivation, deactivation, ramp and action potential, to which we added Gaussian noise with zero mean and standard deviation 0.5 mV.”

2) In the Hodgkin-Huxley application:

“We considered the problem of inferring 8 biophysical parameters in a HH single-compartment model, describing voltage-dependent sodium and potassium conductances and other intrinsic membrane properties, including neural noise, making the model stochastic by nature (Figure 4A, left).”

3) In the STG application:

“This model has been used to demonstrate that virtually indistinguishable activity can arise from vastly different membrane and synaptic conductances in the STG (Alonso and Marder, 2019; Prinz et al., 2004). Here, we build on these studies and extend the model to include intrinsic neural noise on each neuron (see Materials and methods).”

More generally, we hold the view that modelling and explicitly treating *some* form of noise (either in the underlying system and/or the measurement process) is of critical importance for data analysis in neuroscience. This is (arguably) under-appreciated in computational neuroscience, which often concentrates on reproducing averaged neural activity without taking into account the associated variability or uncertainty. So, we maintain that the restriction of stochastic models is, in our view, not a serious restriction.

As the reviewer writes, SNPE could also be applied to deterministic models and will also return posteriors in this case. However, a rigorous interpretation of the posteriors in that case is mathematically subtle, and a mathematically rigorous treatment of this case is therefore, in our view, beyond the scope of the manuscript. Somewhat speculatively, we agree that the posterior would likely be a “smoothed out” version of the true (δ-shape) posterior, where the smoothing is dependent on the capacity of the neural network used for inference. We note that there is some recent work in machine learning to deal with such cases rigorously, e.g. by resorting to spread divergences (https://arxiv.org/abs/1811.08968; didactic blog post by David Barber at https://aiucl.github.io/spread-divergence /).

We have added a small Discussion about deterministic models:

“SNPE, and Bayesian inference more generally, is derived for stochastic models. SNPE can, in principle, also be applied to deterministic models, but a rigorous mathematical interpretation or empirical evaluation in this regime is beyond the scope of this study.”

2) On the one example that is stochastic (Figure 2), I am still confused by the results shown. Specifically, in Figure 2C, the posterior samples consistently and systematically undershoot the true receptive field for intermediate parameter values. This is a simple LN model for which classic maximum likelihood models usually doing an excellent job finding the true receptive field, at least with enough data. If one ran standard maximum likelihood estimation on this, would it miss? If so, why? Is this reflecting that the authors are using too broad a prior and it is not converging to the true posterior or that too few samples are used (the number of samples is not indicated)-if the latter, can the authors run with more samples and show that their method then converges to the correct answer? Or is it something else (e.g. see next paragraph)?In addition, the authors state that their method works because it gets the same answer as MCMC. However, I'm wondering if both might be making errors due to the use of the Polya-Γ sampling scheme since, at least in the Pillow and Scott article cited, this was used not for a standard LN model but rather to model more complex models in which the variance was greater than for a Poisson spike count. In any case, it seems worth explaining where the systematic error in the posterior samples is coming from, given that the LN model is such a simple classic model fit whose parameters have been successfully identified by simpler maximum likelihood methods.

This comment contains multiple sub-questions which we would like to answer separately:

1) Can Polya-Γ MCMC (PG-MCMC) be used to infer a reference posterior for the LN model?

For the LN model, we use PG-MCMC (a likelihood-based method) to obtain a reference posterior to compare SNPE against. As shown in Polson, Scott and Windle, 2012, (cited in the manuscript), PG-MCMC is applicable to Bernoulli GLMs, which the LN model is one instance of. It is not the case that PG-MCMC is only applicable to overdispersed models.

2) Is the result obtained by SNPE correct?

The goal of SNPE is to perform Bayesian inference. Thus, to evaluate whether SNPE “works”, one must not compare its posterior to the ground-truth parameters, but rather to the ground-truth posterior, or at least a proxy thereof. We here used the posterior obtained via PG-MCMC as a reference posterior, which SNPE correctly recovers. We do not have doubts about the correctness of PG-MCMC on this example, in particular as the posterior agrees with summary checks, and this model has a unimodal posterior for which MCMC sampling is well understood and works reliably. (If the reference posterior was wrong, it would also be a bit of a coincidence that SNPE gets the same, but wrong posterior…). The only reason why the reviewer seems to question the correctness of the posterior is that he is surprised by the large posterior variance, see below.

3) Does posterior uncertainty shrink with more data?

The posteriors inferred with the PG-MCMC scheme and SNPE are not tightly constrained around the true filters (see Appendix 1—figure 1, showing posteriors means +/- 1 standard deviation). This “systematic undershooting” does not indicate a failure of the PG-MCMC scheme, but is simply a consequence of the fact that we simulated a short experiment consisting of 100 Bernoulli samples (which we call “single-trial” simulation). For short experiments, posterior uncertainty will be large, and a mismatch between true parameters and posterior mean is expected. As expected, increasing the number of single-trials in one observation leads to progressively tighter posteriors, with posterior samples closer to the true filter. We now describe this result in the Results section and Appendix 1—figure 3:

“Increasing the number of Bernoulli samples in the observed data leads to progressively tighter posteriors, with posterior samples closer to the true filter (Appendix 1—figure 3). Furthermore, SNPE closely agrees with the MCMC reference solution in all these cases, further emphasizing the correctness of the posteriors inferred with SNPE.”

4) Why are we not as “successful as simple MLE methods”

We note that the results by Pillow et al., which the reviewer likely refers to, have been obtained in extremely long recordings, which lead to very tightly constrained posteriors: in this regime, both maximum likelihood estimation and Bayesian inference return the same answer, but this also makes it hard to assess whether the posterior distribution, i.e. the estimation uncertainty, is correctly recovered. The fact that our posterior mean, with more samples, converges to the true parameters shows that, as the data increases, all these methods will converge to the same answer, namely the true parameters. In neuroscience, however, it is very common that one is not in this data-regime, making it important that methods perform correct inference even in the small-data regime.

3) Related to the Figure 2 example, I was struck by the huge errors and nearly zero correlation for the ABC methods. Is this fundamental to ABC or is this because the authors are hamstringing the ABC method by applying it in a very naïve manner that isn't how someone using ABC would apply it. Specifically, from the Materials and methods, it looks like the authors are considering every parameter to be independent, with no smoothing or regularization. I would think that someone using ABC methods would apply regularizations or the like that might dramatically reduce the effective dimensionality of the problem, so it's not clear if the comparison shown in the manuscript is fair. This isn't to say that SNPE won't still work much better than ABC methods, but it's worth checking to make sure the ABC methods are being applied in a realistic manner if the comparison is to be made.

Note that we compare SNPE against SMC-ABC in Figure 2 on two different examples: We show a comparison on the LN model example (Figure 2D) as well as a comparison on the Gabor model (Figure 2G). SNPE outperforms rejection methods in both cases, a result which is consistent with previous work (e.g., Lueckmann, Goncalves et al., 2017; Greenberg et al., 2019). Benchmarking against other methods is not the primary focus of this manuscript.

The LN model comparison is performed on low dimensional summary statistics (10d) rather than raw data traces, so it does not put SMC-ABC to a disadvantage. In the Gabor example, SMC-ABC fails because it is not able to deal with the high-dimensionality of the data. This illustrates an important drawback of SMC-ABC, which would need to be combined with other methods/pre-processing steps, if one wanted to use it for this problem. SNPE allows automatic learning of summary statistics, so that inference on high-dimensional data is possible. We maintain that this is a good illustrative comparison to include.

We also note that the suggestion to smooth the problem would, in effect, just reduce this high-dimensional estimation to an (effectively) lower-dimensional, and substantially easier estimation problem. Accordingly, all estimation approaches would work better, and the differences between methods would be smaller.

4) Ground truthing. The authors suggest that they can't ground truth the technique due, in part, to the stochastic nature of it. While I appreciate there may be some challenges, a lot could be done, especially since most of the manuscript focuses on deterministic examples that are easier to check. For example, in Figures 3-5, parameter estimation results that may or may not be correct are shown and the reader is left to assume they are correct by looking at a single or very small number of examples. While I fully expect that it does work reliably, it would be nice to actually run a large set of simulations for posterior checks (even if, e.g., only the mode of the distribution was checked) and then report some statistics.

Regarding the LN and Gabor models, and as discussed above, we include comparisons of SNPE posteriors with reference posteriors. Furthermore, we now also have a new supplementary figure for the LN model showing consistency with increases in data (see above).

All the other applications covered in the manuscript have intractable likelihoods, so ground-truthing is challenging. Therefore, we opted for posterior predictive checks on the data, rather than ground-truthing on the parameters. In addition, we do point out that the manuscript does, in effect, contain multiple quantitative evaluations:

1) In the ion channel model application, Appendix 1—figure 7 shows the correlation between the observation and the mode of the respective posterior for 350 different observations. We believe this figure demonstrates the reliable quality of the posteriors for this application.

2) In the Hodgkin-Huxley model applications, Figure 4A-E shows the application of SNPE to a synthetic dataset, Figure 4F shows the application of SNPE to 8 different datasets, and Appendix 1—figure 10C,D show multiple samples from the posterior for the synthetic dataset. We believe this is strong evidence of the success of SNPE on this application.

3) In the STG application, Figure 5D in combination with Appendix 1—figure 12A, and Appendix 1—figure 14C show that the summary statistics of a large number of samples from the posterior vary little around the observed summary statistics. This is again strong evidence that SNPE performed well in this application.

5) Related to ground truthing, in Figure 5D and the highlighted 2-D plot of Figure 5C, I am confused as to whether the example shows the success or partial failure of the method. Specifically, the point labeled "2" in the Figures 5C and 5D appears to be in a bright (and therefore presumably high probability?) region of the parameter space, both in an absolute sense and possibly (it's hard to tell) compared to the light purple circle simulation (and also relative to going further down below this light purple trace). However, the authors use point 2 to make the point that the simulation looks very far off from the reference trace. This suggests, possibly, that similar probabilities being output by SNPE is on the one hand (values close to the light purple circle) leading to a very good fit and on the other hand (red point 2) leading to a very bad fit – thus, this suggests that one perhaps can't trust “bright” regions in the SNPE posterior to correspond to “good” fits. Can the authors check this and address the more general concern. I'm guessing this may relate to my larger conceptual question of interpreting the “probabilities” for SNPE in my reviewer comment #1 above.

We thank the reviewer for raising this potentially confusing point. As the reviewer points out, when inspecting the 2D-marginal probability distribution, the point labeled “2” seems to lie in regions of high posterior probability. However, this does not imply that the point actually lies in a region of high probability in the full 31-dimensional posterior space (its 2-d project lies in a high probability region, but the full high-d version does not), and thus the visual inspection of 2D-marginals can be misleading. To demonstrate this point, we have added a supplementary figure that shows the posterior probability along both paths, see Appendix 1—figure 13 and refer to this figure in the main text as follows:

“Note that, while parameter set 2 seems to lie in regions of high probability when inspecting pairwise marginals, it in fact has low probability under the full posterior distribution (see Appendix 1—figure 13).”

As can be seen, the posterior probability along the orthogonal path drops off very quickly, which matches the off-looking activity and further demonstrates the success of the method.

6) Difference from “brute force methods”. I thank the authors for trying to generate a comparison to brute force methods by showing a rejection method. However, I don't think this is necessary and I think it's confusing as I don't think “brute force” and “rejection” are the same (i.e. I'd remove this from the paper as being more confusing than helpful […] And overall, this focus on the power of post-database-generation analysis is why I still think SNPE, and the more general focus on use of density estimators, is a potentially transformative contribution for the practice of model building (in addition to SNPE's applicability to stochastic problems that is already well emphasized in the manuscript).

We thank the reviewer for the suggestion regarding the clarification of the method in the context of previous approaches. We agree that the process of generating a large set of simulations and analysing it can be decoupled, at least for the case of one round SNPE—we note that in multi-round SNPE, the sampling of parameters is adaptively guided by the posterior from the previous round, and thus this decoupling of sampling and analysis is somewhat debatable. We have added a clarification regarding the brute-force aspect of SNPE in the Discussion subsection “Related Work”:

“We should still note that SNPE can require the generation of large sets of simulations, which can be viewed as a brute-force step, emphasising that one of the main strengths of SNPE over conventional brute-force approaches relies on the processing of these simulations via deep neural density estimators.”

We have now removed the analysis we had made and included (at the reviewers request, we were under the impression we were following his instructions) in the previous round. If the reviewer or the editor wants it included again, we are happy to put it back in.